# Reconstruction of the cell pseudo-space from single-cell RNA sequencing data with scSpace

Jingyang Qian[1,2,3,12], Jie Liao [1,2,3,12] ✉, Ziqi Liu[1,12], Ying Chi [4,12], Yin Fang [5], Yanrong Zheng[1,6], Xin Shao [1,2,3,7], Bingqi Liu[8], Yongjin Cui[1,2,3], Wenbo Guo[1,2,3], Yining Hu[1,3], Hudong Bao[1], Penghui Yang[1,2,3], Qian Chen[2,3], Mingxiao Li [9], Bing Zhang [4,10,11] ✉ & Xiaohui Fan [1,2,3,10] ✉

Tissues are highly complicated with spatial heterogeneity in gene expression. However, the cutting-edge single-cell RNA-seq technology eliminates the spatial information of individual cells, which contributes to the characterization of cell identities. Herein, we propose **s**ingle-**c**ell **s**patial **p**osition **a**ssociated **c**o-**e**mbeddings (scSpace), an integrative method to identify spatially variable cell subpopulations by reconstructing cells onto a pseudo-space with spatial transcriptome references (Visium, STARmap, Slide-seq, etc.). We benchmark scSpace with both simulated and biological datasets, and demonstrate that scSpace can accurately and robustly identify spatially variated cell subpopulations. When employed to reconstruct the spatial architectures of complex tissue such as the brain cortex, the small intestinal villus, the liver lobule, the kidney, the embryonic heart, and others, scSpace shows promising performance on revealing the pairwise cellular spatial association within single-cell data. The application of scSpace in melanoma and COVID-19 exhibits a broad prospect in the discovery of spatial therapeutic markers.

Uncovering the organization of cells in a tissue and how this organization affects function is a fundamental pursuit of life science research[1,2]. Spatial characteristic plays a key role, sometimes even as a determinant in the identity of a single cell in specific complex tissues such as brain regions and tumor microenvironments (TME)[2–5] since cell subpopulations show more heterogeneity in space than transcriptomes. While single-cell RNA sequencing (scRNA-seq) technologies have greatly expanded our understanding of the comprehensive characterization of cells from complex tissues, classic protocols of scRNA-seq[6–9] are to digest tissues into single-cell suspension, which leads to the loss of spatial information of cells. Meanwhile, recently developed spatially resolved transcriptomics technologies can overcome the limitations of scRNA-seq by profiling the gene expression with spatial information preserved across tissue sections[10–15], though, they fail to provide unbiased transcriptomes of individual cells.

[1]College of Pharmaceutical Sciences, Zhejiang University, 310058 Hangzhou, China. [2]Future Health Laboratory, Innovation Center of Yangtze River Delta, Zhejiang University, 314102 Jiaxing, China. [3]National Key Laboratory of Modern Chinese Medicine Innovation and Manufacturing, 310058 Hangzhou, China. [4]DAMO Academy, Alibaba group, 310052 Hangzhou, China. [5]College of Computer Science and Technology, Zhejiang University, 310013 Hangzhou, China. [6]Key Laboratory of Neuropharmacology and Translational Medicine of Zhejiang Province, School of Pharmaceutical Sciences, Zhejiang Chinese Medical University, 310053 Hangzhou, China. [7]Key Laboratory of Integrated Oncology and Intelligent Medicine of Zhejiang Province, Affiliated Hangzhou First People's Hospital, Zhejiang University School of Medicine, 310006 Hangzhou, China. [8]School of Mathematical Sciences, Zhejiang University, 310058 Hangzhou, China. [9]Institute of Microelectronics of the Chinese Academy of Sciences, 100029 Beijing, China. [10]iMedicine Lab, Alibaba-Zhejiang University Joint Research Center for Future Digital Healthcare, 310058 Hangzhou, China. [11]Alibaba Cloud, Alibaba Group, 310052 Hangzhou, China. [12]These authors contributed equally: Jingyang Qian, Jie Liao, Ziqi Liu, Ying Chi. ✉e-mail: liaojie@zju.edu.cn; zb224035@alibaba-inc.com; fanxh@zju.edu.cn

It is recommended to conduct both state-of-the-art technologies to decipher the cellular and spatial heterogeneity within the complex tissue across multiple conditions simultaneously. However, this is expensive and there may not be enough available samples. Therefore, computational methods are urgently needed to integrate single-cell and spatial transcriptomics (ST) data from different experiments and patients. Recently, in silico methods are developed to improve the quality of ST data by integrating single-cell and ST data, including spot deconvolution[16–21], spatial mapping[22–24], and gene imputation[25–27]. However, there are few approaches reported for reconstructing the spatial association of cells and identifying spatially heterogeneous subpopulations from single-cell data using ST data as references.

To this end, we introduce scSpace, an integrative method that uses ST data as a spatial reference to reconstruct the pseudo-space of scRNA-seq data. Subsequently, a space-informed clustering is conducted to identify spatially variable cell subpopulations within the

scRNA-seq data (Fig. 1a). Specifically, using a transfer learning model, termed transfer component analysis (TCA), which was originally used in domain adaptation to solve a learning problem in a target domain by utilizing the training data in a different but related source domain[28], scSpace enables eliminating the batch effect between single-cell and ST data and extracting the shared latent feature across these two types of data. By creatively integrating pseudo-space reconstruction and space-informed clustering, scSpace significantly outperforms other methods on simulated datasets. Moreover, both existing ST data and single-cell transcriptomics data are utilized to validate the performance of scSpace. Finally, we apply scSpace to existing datasets from the embryonic heart[29], human middle temporal gyrus (MTG)[30], human melanoma[31], and human lung under normal and COVID-19 states[32] to discover significant cell subpopulations with spatial heterogeneity as well as transcriptional specificities in various circumstances.

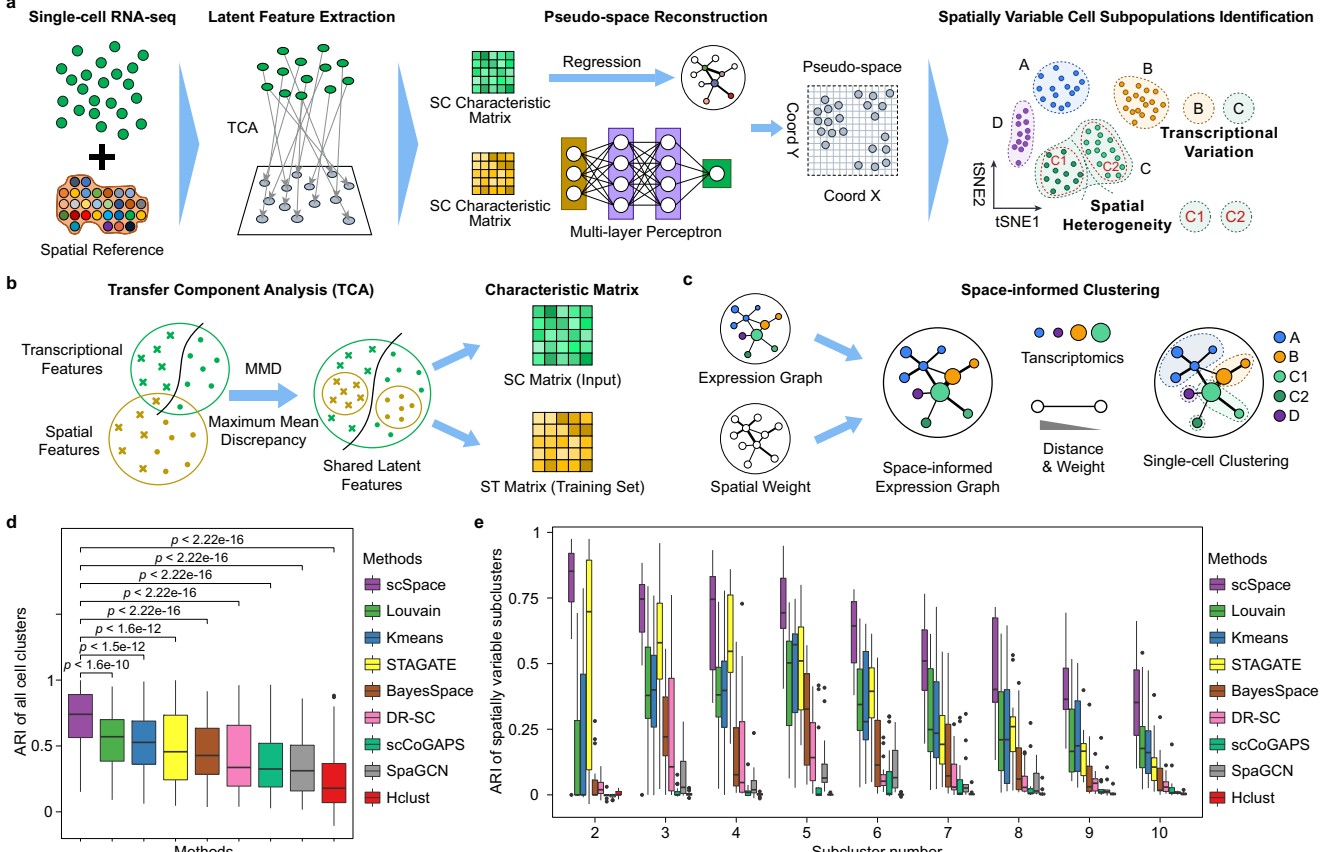

**Fig. 1 | Schematic workflow of scSpace and performance evaluations on simulated data. a** Overview of the design concept of scSpace. Given the scRNA-seq data (SC) and spatial transcriptomics reference (ST), scSpace co-embeds these two types of data into a shared latent space and extracts the shared latent features. Using the characteristic matrix from ST data, scSpace trains a multi-layer perceptron model with spatial coordinates as the outcome and latent features as the predictors. The trained model is then applied to the characteristic matrix from SC data for pseudo-space reconstruction. Based on the gene expression profiles as well as the pseudo-space information, scSpace identifies the spatially variable cell subpopulations from scRNA-seq data. **b** Conceptual framework of latent feature extraction with scSpace. A transfer learning method termed transfer component analysis (TCA) is applied to extract the shared latent feature representation across scRNA-seq and spatial transcriptomics data. TCA first projects the scRNA-seq and spatial transcriptomics data into a Reproducing Kernel Hilbert Space (RKHS), and then reduce the difference in the distribution of transformed two domain data by minimizing the maximum mean discrepancy (MMD) between them. The shared

latent feature representation across two domain data is then extracted for the next pseudo-space reconstruction step. **c** Conceptual framework of space-informed clustering with scSpace. The gene expression graph is first constructed on the reduced principal components derived from normalized gene expression profiles of single cells using the *k*-nearest neighbor (KNN) algorithm. For each edge in the gene expression graph, a spatial weight is introduced based on the distances between cells in the pseudo-space. Then, scSpace performs the unsupervised clustering step on the space-informed gene expression graph to identify spatially variable cell subpopulations from scRNA-seq data. **d** and **e** Comparison of scSpace with other existing clustering methods in identifying all cell clusters (**d**) and only spatially heterogeneous subclusters (**e**) on 140 simulated datasets. Data are presented as boxplots (minima, 25th percentile, median, 75th percentile, and maxima). *P*-value is calculated with the two-sided Wilcoxon rank-sum test (the exact *P*-values from left to right are 1.6e−10, 1.5e−12, 1.6e−12, 3.1e−18, 1.1e−17, 4.7e−28, 2.2e−30, and 4.0e−35, respectively).

# Result

## Design concept of scSpace and performance evaluation on simulated data

We hypothesize that single-cell analysis, which typically considers only transcriptome information, can be enhanced by recovering the spatial arrangement of cells. Therefore, scSpace is developed to reconstruct the spatial architectures of cells and thus identify cell subpopulations with spatial heterogeneity from single-cell data. As illustrated in Fig. 1a, the workflow of scSpace comprises three main components: (1) extract the shared latent biological feature representation across scRNA-seq and ST data, (2) reconstruct the spatial association of cells from scRNA-seq data, and (3) identify spatially heterogeneous cell subpopulations from the reconstructed spatial architecture of single-cell (optional). Briefly, scSpace first applies a transfer learning model, termed TCA, to eliminate the batch effect as well as extract shared biological characteristics between scRNA-seq and ST data (Fig. 1b). Subsequently, scSpace trains the shared latent features extracted from ST data with a multi-layer perceptron to learn the relationship between characteristics and spatial coordinates. Next, the trained model is employed to generate spatial coordinates for single cells with the extracted characteristic matrix from scRNA-seq data via feature representation. The resulting spatial arrangement of single cells is termed the "pseudo-space". Additionally, scSpace can further perform space-informed clustering to identify spatially heterogeneous cell subpopulations in scRNA-seq data. Based on the Leiden algorithm[33], a classical clustering method widely used in single-cell data analysis, we extend its applicability by introducing the spatial weight of edges in the gene expression graph constructed from gene expression profiles. Therefore, scSpace allows for both gene expression and pseudo-space information of cells in the clustering process (Fig. 1c).

We first evaluate the performances of scSpace on a series of simulated datasets to test whether scSpace can reconstruct the spatial arrangement of single cells. To achieve this, we employ Splatter[34] to simulate paired scRNA-seq data and ST data with varying numbers of spatially heterogeneous cell populations (Supplementary Fig. 1 and Methods). Using ST as the spatial reference, scSpace exhibits outstanding performance on the spatial reconstruction of scRNA-seq data (Supplementary Fig. 2a) with Pearson correlation coefficient (PCC) of pairwise distances between cells in the pseudo-space and original space above 0.9 (Supplementary Fig. 2b and Supplementary Data 3).

We next conduct scSpace to identify spatially heterogeneous cell subpopulations and then compare its performance with other methods using 140 simulated data, which constitute different spatial distribution patterns including several cell clusters and 2–10 refined cell subclusters. Specifically, the performance of scSpace is compared with three classical clustering algorithms that are applied to scRNA-seq data, Louvain, K-means and Hierarchical clustering (Hclust), four recently published spatial domain identification methods, SpaGCN[35], STAGATE[36], BayesSpace[37], and DR-SC[38], and one latent space learning method for gene expression data using transfer learning algorithms, scCoGAPS[39]. The clustering accuracy is evaluated by the adjusted Rand index (ARI) of cell-type assignments. As shown, scSpace significantly performed superior to other methods (Fig. 1d), especially in spatially variable cell subpopulation identification (Fig. 1e). The ARI of scSpace is consistently the highest as the number of subclusters increases. Besides, the results also illustrate that the performance of scSpace remains relatively stable when the number of subclusters increases to 8, whereas other methods continue to decrease (Fig. 1e). Notably, all four spatial domain identification methods perform worse than clustering methods designed for scRNA-seq data over the simulated data, which may due to the difference in application purposes and scopes between them. scSpace, as well as other single-cell clustering methods, is designed for scRNA-seq data analysis. Moreover, scSpace further focuses on distinguishing spatially heterogeneous cell subpopulations. On the contrary, spatial domain identification methods are generally

more suitable for ST data rather than identifying specific subpopulations in single-cell data.

For spatial weight construction, scSpace is also suitable for the framework that works with distances, which has been employed in previous methods such as SpatialDE[40]. Since the hyperparameter $l$, also known as the characteristic length scale, determines how rapidly the weight decays as a function of distance and will influence the performance of scSpace partly, we next discuss the effect of $l$ on the performance of scSpace. The results show that scSpace performed best on 140 simulated data when $l$ is set to 20 (Supplementary Fig. 3a–d). Furthermore, the space-informed clustering is compared between the two strategies of calculating the spatial weight $w$ in scSpace, and the results indicate no significant difference between them when $l = 20$ (Supplementary Fig. 3e).

In addition, we also evaluate the computation time and scalability of scSpace by increasing the cell number from 500 to 50,000. As shown in Supplementary Fig. 4, with a 24 GB NVIDIA GeForce RTX 3090 GPU, scSpace could process 50,000 single cells within 20 min, and the clustering accuracy remains stable at a high level as the number of cells increases.

## Reconstruction of the hierarchical structure of human and mouse cortex using existing ST data by scSpace

ST data are better examples than simulated data for evaluating the performance of scSpace on spatial reconstruction because cell or spot coordinates in spatial data are biologically meaningful and objectively present. Therefore, two ST datasets of the human dorsolateral prefrontal cortex (DLPFC)[41] and mouse primary visual cortex (V1)[42], which are profiled by 10X Visium and STARmap, respectively, are collected to evaluate the performance of scSpace in reconstructing the hierarchical structures of highly organized cortex tissues. Specifically, scSpace uses the ST data of one tissue slice as the spatial reference and reconstructs the pseudo-space of spots in another tissue slice with coordinates removed in advance.

As illustrated in Fig. 2a and Supplementary Figs. 5 and 6, scSpace can successfully reconstruct the hierarchical structure of different human DLPFC layers in the pseudo-space, with the relative position between the layers as well as the neighbor system of spots preserved (Fig. 2b). Further analysis demonstrates that the pairwise distances between spots in the pseudo-space and original space are highly correlated (Fig. 2c). Moreover, we have examined the spatial distribution of differential expression genes for each layer identified by Seurat[43] and find that the spatial expression patterns of these genes in the pseudo-space and original space exhibit consistent distributions (Fig. 2d).

Similar results are reproduced when we estimate scSpace using another spatially resolved single-cell transcriptomics data with 1020 targeted genes from mouse V1 neocortex by STARmap (Fig. 2e). Interestingly, for the STARmap data, scSpace can successfully restore the spatial arrangement of cells along the X-axis, with a Pearson's correlation coefficient (PCC) of 0.782 (Fig. 2f), yet fails along the Y-axis (PCC = 0.112) (Supplementary Fig. 7a). This difference may cause by the specific shape of the tissue slices used for RNA imaging. In the 1.4 mm by 0.3 mm slice of the mouse V1 cortex, the X-axis (1.4 mm) corresponds to the layer-axis, showing striking heterogeneity in both cell type and gene expression and the opposite is true for the Y-axis (0.3 mm) (Supplementary Fig. 7b, c). We further investigate the expression trend of marker genes of each layer along the X-axis in the pseudo-space and original space and find that the pseudo-space constructed by scSpace well preserves the expression pattern of genes along the layer-axis in the original space (Fig. 2g).

## Evaluating the performance of scSpace on tissues with more complex structures

When more complex conditions are encountered, such as cancers, the molecular signature of the tissues is highly heterogeneous among

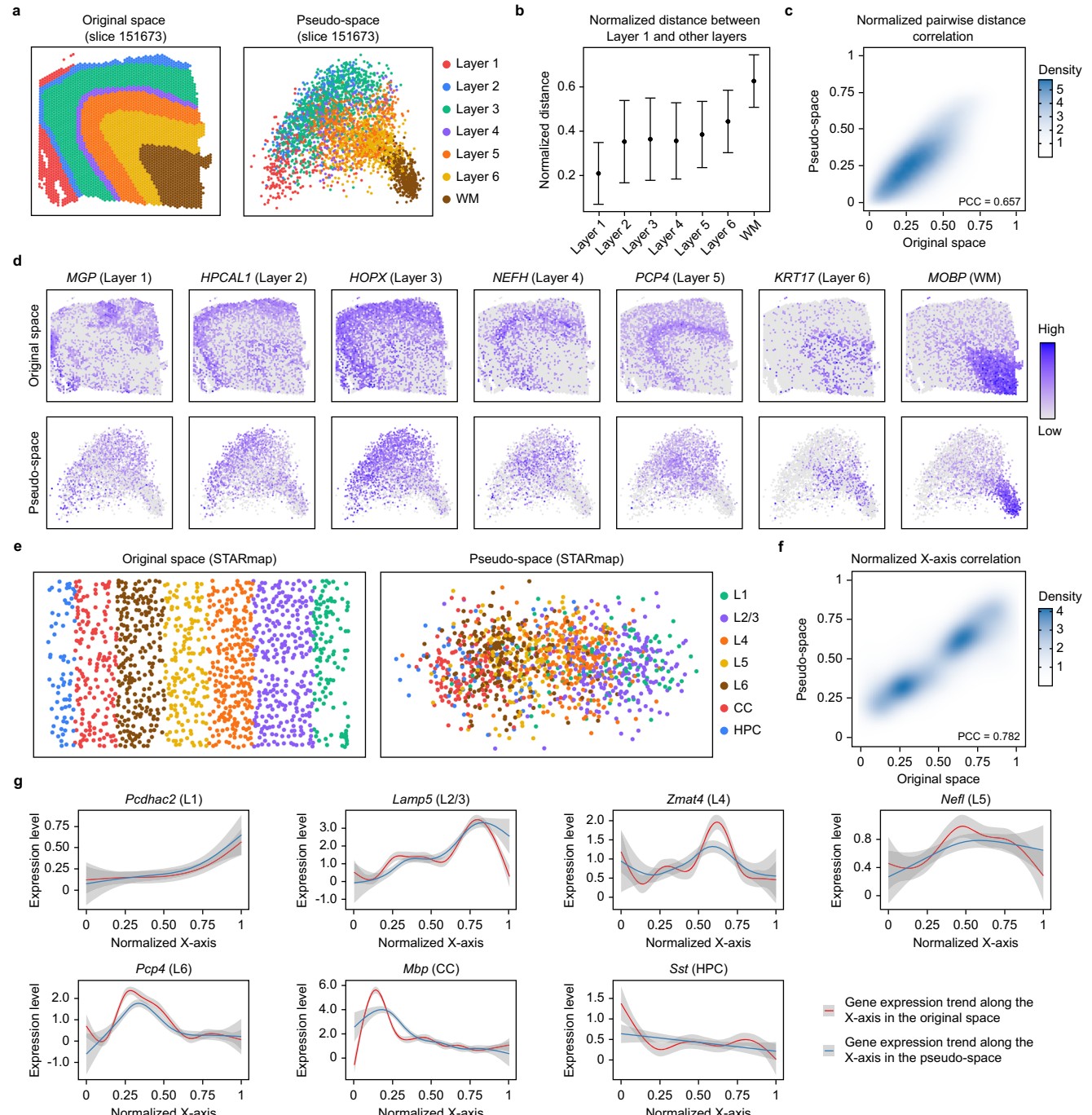

**Fig. 2 | Reconstructing the hierarchical structure of brain cortex using ST data.** **a** The original space (left) and reconstructed pseudo-space (right) of human DLPFC layers of slice 151673. **b** Normalized pairwise distance between spots from different layers to Layer 1. Data are presented as the median ± SD. The number of data points for the error bars from left to right are 37,401, 69,069, 269,997, 59,514, 183,729, 188,916, and 140,049, respectively. **c** Density plot of the correlation of pairwise distances between spots in the original-space and pseudo-space. **d** The spatial expression patterns of marker genes of each layer in the original-space and pseudo-space. **e** The original-space (left) and reconstructed pseudo-space (right) of mouse primary visual cortex (V1) layers. **f** Density plot of the correlation of the spatial arrangement of cells along the *X*-axis in the original-space and pseudo-space. **g** The expression trend of marker genes of each layer along the *X*-axis in the original space and the pseudo-space. The confidence interval (CI) is 0.95.

different patients and even within tumors. Thus, scSpace is applied to ST data from the human skin squamous cell carcinoma (SCC)[44] and human HER2 breast cancer (BC)[45], to further evaluate the performance of scSpace on the reconstruction of the pseudo-space. Notably, the four patients of SCC exhibit substantial variability in the correlation of pairwise distances between spots in the pseudo-space and original space. In detail, for Patient 2, the mean PCC value of pairwise distances between spots in the pseudo-space and original space across three slices is 0.63, however, for Patients 5, 9, and 10, the mean PCC values are around 0.2–0.4 (Supplementary Fig. 8). Similar results are revealed on the eight patients of BC data, compared with Patients B, F, G, and H, Patients A, C, D, and E show higher correlations of pairwise distances between spots in the pseudo-space and original space (Supplementary Figs. 9, 10, and 11). This difference may cause by the intrinsic difference in the spatial homogeneity of tumors[45]. Compared with tissue slices from other donors, the biological replicates from Patients 5, 9, and 10

of the SCC and Patients B, F, G, and H of the BC may share lower spatial coherences between each other, leading to a poor result of the pseudo-space reconstruction by scSpace. Moreover, as shown in Supplementary Fig. 12a, when scSpace is used in tissue sections with inherent spatial patterns, such as cortical layers of brains whose structure is conservative even between multiple individuals, the resulting PCC scores remain at a high level robustly across many consecutive or discrete slices. In contrast, limitation regarding variable morphology of the tissue microenvironment, the heterogeneity across multiple donors, and the intrinsic intra-tumor difference exists in distinct tissue sections from the SCC (Supplementary Fig. 12b) and BC (Supplementary Fig. 12c) patients occasionally.

## Regional reconstruction of scRNA-seq data in different circumstances with scSpace

To investigate the ability of scSpace in restoring the relative spatial associations among cells, we focus on real scRNA-seq data that are obtained from different tissues, including the small intestinal villus[46], liver lobule[47], V1 neocortex[48], and kidney[49]. After the cells are allocated to the pseudo-space, their spatial distributions vary from region to region (Supplementary Figs. 13–17), exhibiting a diverse functioning zonation in the tissue microenvironment. As shown, the "cortex−outer medulla−inner medulla" three-layer zonal distribution pattern of cells in the kidney is restored by scSpace (Supplementary Fig. 13c−e). Furthermore, scSpace also accurately reconstructs the more refined spatial architectures of the various cell populations of the thin limb of Loop of Henle (tl-LoH) as well as the principal cells (PCs) of the ureteric epithelium, whose diversities relate at least in part to position along the cortical−medullary axis[49]. As illustrated in Supplementary Fig. 13f−h and Supplementary Fig. 13i−k, the cell populations of tl-LoH and the PCs exhibit sequential orders along the cortical−medullary axis in the pseudo-space, which are consistent with their anatomies. Similar results are reproduced when reconstructing the pseudo-space of scRNA-seq data with another spatial reference from kidney Slide-seq V2 ST data (Supplementary Fig. 14). In addition, the regional reconstruction of the small intestinal villus (Supplementary Fig. 15), liver lobule (Supplementary Fig. 16), and V1 neocortex (Supplementary Fig. 17), further demonstrates the general applicability of scSpace in the spatial reconstruction of scRNA-seq data from different tissues.

## Regional reconstruction of scRNA-seq data in the embryonic human heart using scSpace

To further demonstrate the high flexibility of scSpace in spatial reconstruction and spatial heterogeneous subpopulations identification in single-cell data, we employ the ST data of embryonic heart generated by Asp et al.[29] as the spatial reference, to reconstruct the pseudo-space of the paired scRNA-seq data from the same experiment (Fig. 3). Here, we focus on the cardiomyocytes, whose transcriptional as well as spatial heterogeneity reflects developmental origins and differences in electrophysiological, contractile, and secretory processes[50].

The pseudo-space constructed by scSpace accurately restores the molecular architecture and spatial relationship between different cardiomyocytes[29]. Specifically, the atrial and ventricular cardiomyocytes are separated from each other in the pseudo-space, in agreement with their true spatial localization in the atria and ventricles, respectively. Moreover, the *Myoz2*-enriched cardiomyocytes, expressing *MYOZ2* and *FABP3*, are localized between the atrial cardiomyocytes and ventricular cardiomyocytes in pseudo-space, which is also consistent with its true localization in both the atria and ventricles (Supplementary Fig. 18a, b).

Subsequently, the scRNA-seq data are classified into 14 clusters by scSpace and Seurat, respectively, according to the number of cell types in the original annotations (Fig. 3a). As illustrated in Fig. 3b, atrial (C9), ventricular (C4), and *Myoz2*-enriched (C12) cardiomyocytes can be precisely distinguished by scSpace (Left). However, Seurat fails to identify the spatial variation within single-cell data at the same clustering resolution (Fig. 3b, c). The ARI and normalized mutual information (NMI) scores for cardiomyocytes of scSpace, representing the clustering accuracy, are 0.91 and 0.85, respectively, which are significantly higher than that of Seurat (0.16 and 0.18), as shown in Fig. 3d. Besides, the spatial localization in the pseudo-space as well as the spatial expression patterns of relevant marker genes of these three clusters identified by scSpace (Fig. 3e-g) are also in line with the three types of cardiomyocytes (Supplementary Fig. 18).

To avoid the misclassification of three types of cardiomyocytes by Seurat due to the lack of clustering resolution, we have further explored the influence of cluster number ($K$) on the results of Seurat. Specifically, as illustrated in Fig. 3h and i, with gradually increasing the $K$ from 14 to 23, the atrial and ventricular cardiomyocytes are indeed identified by Seurat at a higher clustering resolution, however, the ventricular and *Myoz2*-enriched cardiomyocytes are still not well separated eventually. The ARI scores of all cell types or cardiomyocytes for Seurat are consistently lower than scSpace (Supplementary Fig. 19a). In short, the present results indicate that scSpace is a relatively accurate and efficient method for identifying the subpopulations that are similar in transcriptome but heterogeneous in space.

## Recovering the spatial order of excitatory/inhibitory neuron subtypes with scSpace

We next perform scSpace to reconstruct the pseudo-space of human middle temporal gyrus (MTG) snRNA-seq data generated by Hodge et al.[30], which contain 15,928 cells, including 10,708 excitatory neurons, 4297 inhibitory neurons, and 923 non-neuronal cells. Compared with other cortex datasets, the MTG data comprise a more complex mixing of cell types, including the layer information of cells and the taxonomy of 69 neuron subtypes. Specifically, the human DLPFC 10X Visium ST data mentioned above is utilized as the spatial reference. As illustrated in Fig. 4a, scSpace successfully reconstructs the spatial hierarchical structure of layer 1 (L1) to layer 6 (L6), and the normalized distance between cells and L1 increases layer by layer from L1 to L6 (Fig. 4b), which consistently with previous results.

Subsequently, scSpace is employed to recover the layer distribution of refined excitatory and inhibitory neuron subclasses to explore its applicability in the spatial reconstruction of more complex architectures. The spatial distribution of each excitatory and inhibitory neuron subclass is accessed from the original publication[30]. As illustrated in Fig. 4c, scSpace accurately reconstructs the span-layer spatial architecture of excitatory neuron subclasses (Fig. 4d), with different excitatory neuron types broadly segregating by layer in the pseudo-space (Fig. 4e). For the two major branches of inhibitory neurons, distinguished by expression of *ADARB2* and *LHX6*, scSpace also spatially restores the positional relationship between them. The *ADARB2* branch shows more diversity in L1–L3 than L4–L6, and the opposite is true for the *LHX6* branch (Fig. 4f–h). In total, these results demonstrate that the pseudo-space reconstructed by scSpace has biological significance (Fig. 4c, g) and rationalizes the subsequent space-informed clustering based on it (Fig. 4i).

Next, by combining transcriptional and spatial information of single cells, we apply space-informed clustering to a total of 69 subclasses (including 10 *RORB*-expressing types, 7 *FEZF2*-expressing types, 4 *THEMIS*-expressing types, and 3 *LAMP5*-expressing types in excitatory neurons, as well as 6 *LAMP5/PAX6* subclasses, 21 *VIP* subclasses, 11 *SST* subclasses, and 7 *PVALB* subclasses in inhibitory neurons) using scSpace, and compare the clustering results with Seurat. As shown in Fig. 4i, j, Supplementary Figs. 20, and 21, scSpace achieves more accurate clustering results in each of the major subclasses, which substantiates the utility of scSpace on spatially heterogeneous subclasses identification.

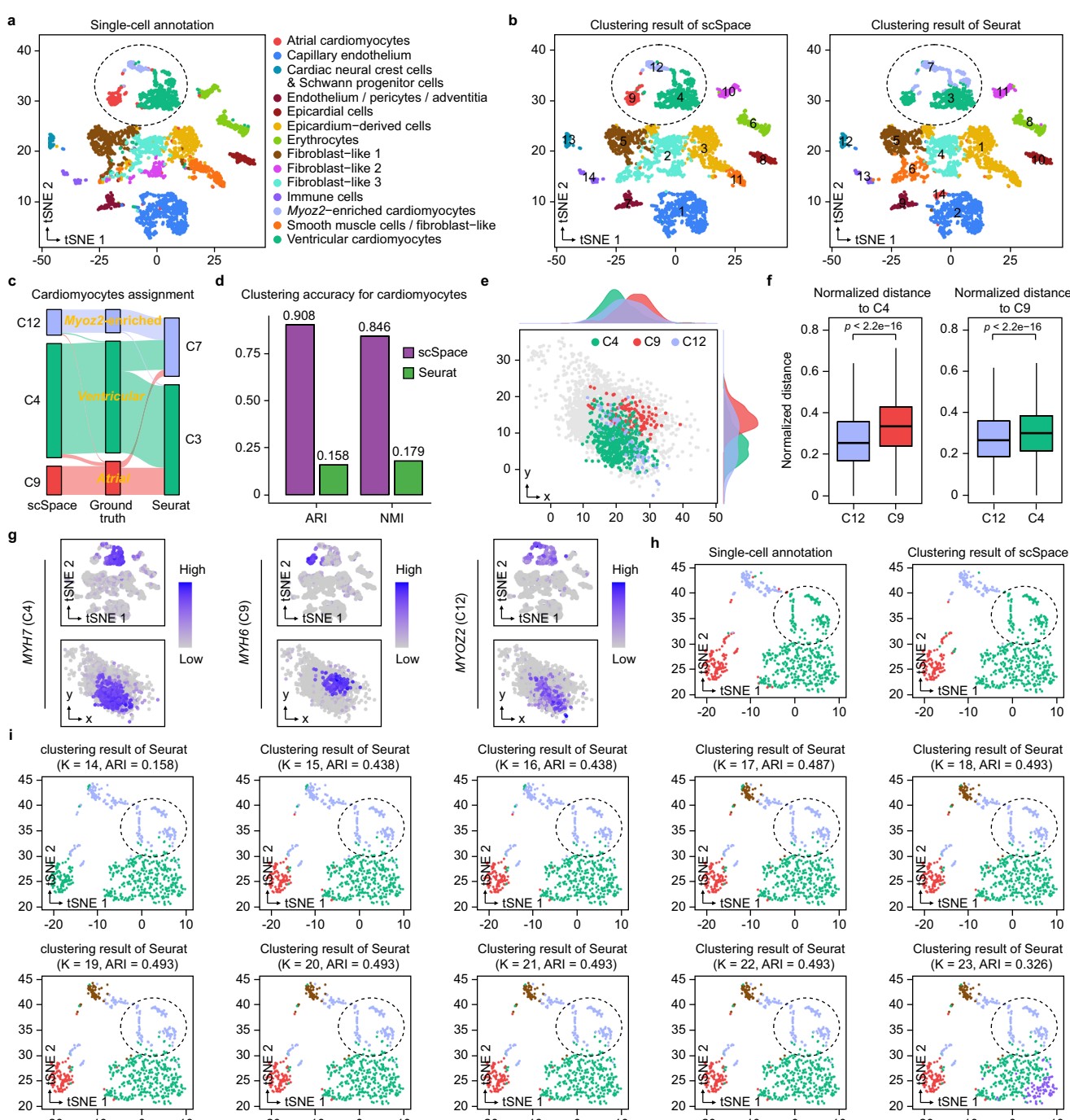

**Fig. 3 | Regional reconstruction of scRNA-seq data in the human embryonic heart using scSpace. a** t-SNE visualization of human embryonic heart scRNA-seq data, the single-cell annotation is obtained from the original publication. **b** Clustering results of scSpace (left) and Seurat (right). **c** Sankey plot showing the assignment of cardiomyocytes in scSpace and Seurat. **d** Comparison of scSpace with Seurat in identifying cardiomyocyte subpopulations. **e** The spatial distribution of three cardiomyocyte subpopulations (C4, C9, and C12) identified by scSpace in the pseudo-space. **f** Normalized pairwise distance between different cardiomyocyte subpopulations to C4 (left) and C9 (right). Data are presented as boxplots (minima, 25th percentile, median, 75th percentile, and maxima). The number of

data points (to C4) are 48,654 and 58,671, respectively; the number of data points (to C9) are 12,546 and 58,671, respectively. *P*-value is calculated with the two-sided Wilcoxon rank-sum test (the exact *P*-values from left to right are 0 and 8.6e−89, respectively). **g** Expression patterns of marker genes for three cardiomyocyte subpopulations in the t-SNE plot (top) and pseudo-space (bottom). **h** t-SNE visualization of three cardiomyocyte subpopulations, colored by the original annotation (left) and clustering result of scSpace (right). **i** t-SNE visualization of three cardiomyocytes with Seurat's clustering result under different targeted cluster number setting (from 14 to 23).

## Discovery of spatially variated subpopulations in human cortex from scRNA-seq data

As mentioned above, scSpace shows the superiority of spatial reconstruction and spatially heterogeneous subpopulation identification in MTG snRNA-seq data. Here, we further demonstrate its ability on

deciphering spatially variated subpopulations in the human cortex from experimental snRNA-seq data accessed from Allen Brain Atlas. The snRNA-seq data include single-nucleus transcriptomes from nuclei across multiple human cortical areas, covering the middle temporal gyrus (MTG), the anterior cingulate cortex (ACC), the primary visual

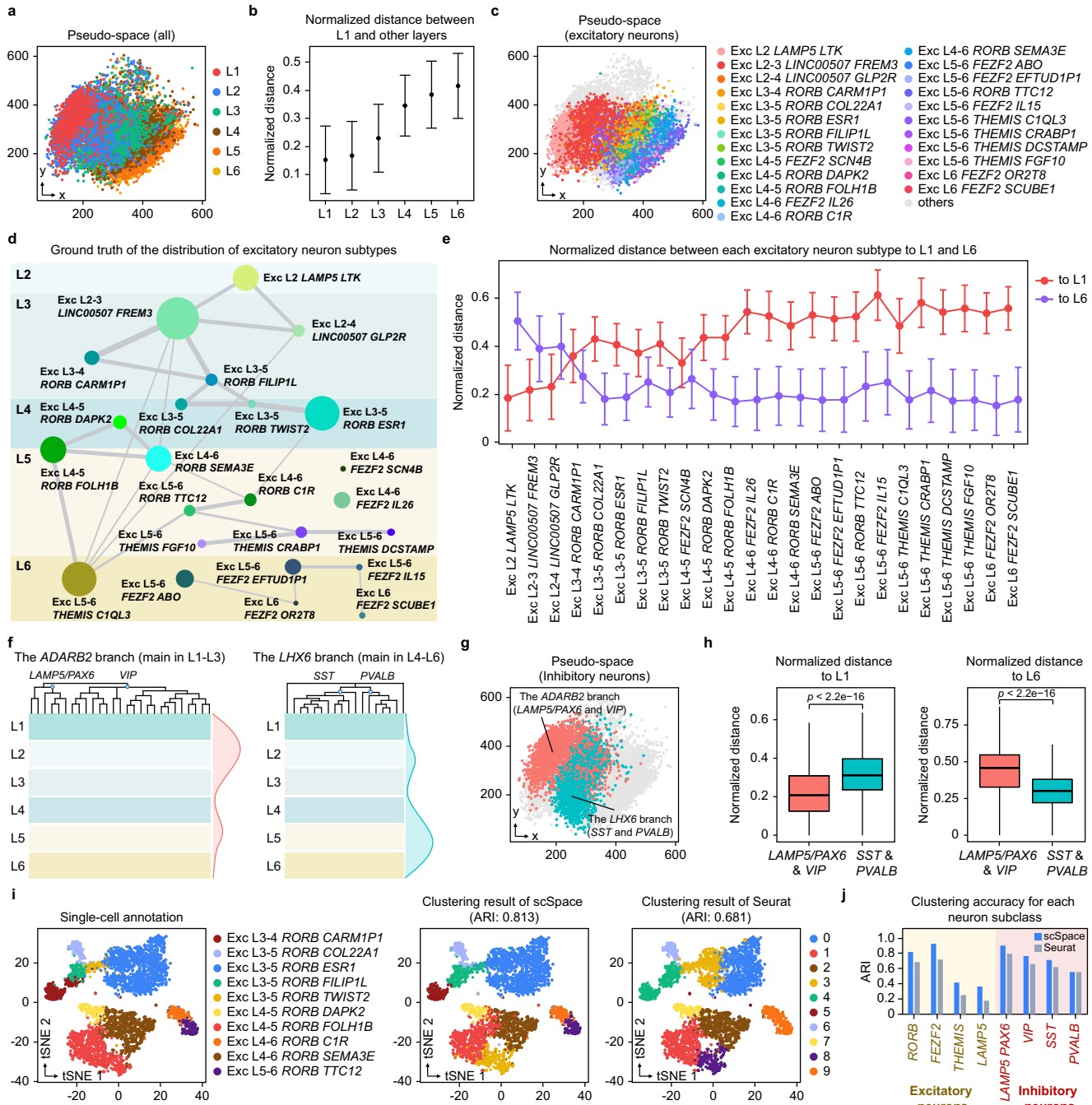

**Fig. 4 | Recovering the spatial order of the excitatory/inhibitory neuron subtypes with scSpace. a** The reconstructed pseudo-space of human MTG snRNA-seq data. **b** Normalized pairwise distance between cells from L1 to different layers. Data are presented as the median ± SD. The number of data points for the error bars from left to right are 579,426, 2,300,488, 3,747,708, 3,094,576, 3,936,008, and 2,901,972, respectively. **c** Spatial distribution of excitatory neuron subclasses in the pseudo-space. All other subclasses are labeled as "others". **d** Schematic of the average layer position for each excitatory neuron subclass. **e** Normalized pairwise distance between different excitatory neuron subclasses to L1 (red) and L6 (purple). Data are presented as the median ± SD. The number of data points (to L1) from left to right are 873,712, 2,457,584, 182,920, 301,280, 164,628, 172,160, 100,068, 1,536,528, 186,148, 936,120, 836,052, 26,900, 172,160, 370,144, 179,692, 158,172, 57,028, 83,928, 337,864, 60,256, 401,348, 1,653,812, 20,444, and 55,952, respectively; the number of data points (to L6) from left to right are 2,189,964, 6,159,948, 158,490, 755,160, 412,641, 431,520, 250,821, 3,851,316, 466,581, 2,346,390,

2,095,569, 67,425, 431,520, 927,768, 450,399, 396,459, 142,941, 210,366, 846,858, 151,032, 1,005,981, 4,145,289, 51,243, and 140,244, respectively. **f** Schematic of average layer positions for two branches of inhibitory neuron subclasses. **g** Spatial distribution of inhibitory neuron subclasses in the pseudo-space. All other subclasses are colored with gray. **h** Normalized pairwise distance between different Inhibitory neuron subclasses to L1 (left) and L6 (right). Data are presented as boxplots (minima, 25th percentile, median, 75th percentile, and maxima). The number of data points (to L1) is 2,496,320 and 1,984,144, respectively; the number of data points (to L6) is 6,257,040 and 4,973,268, respectively. *P*-value is calculated with the two-sided Wilcoxon rank-sum test (the exact *P*-values from left to right are both 0). **i** t-SNE visualization of 10 RORB-expressing excitatory neuron subclasses, colored by the single cell annotation obtained from the original publication (left), the clustering result of scSpace (middle), and Seurat (right). **j** Comparison of the clustering accuracies between scSpace and Seurat.

cortex (V1C), the primary motor cortex (M1C), the primary somato-sensory cortex (S1C), and the primary auditory cortex (A1C), which could be a suitable example to evaluate the ability of scSpace that processing the similar single-cell data from multiple sources. As shown in Supplementary Fig. 22, after spatial reconstruction by scSpace, the distribution density of cells in different cortex layers is significantly different in the pseudo-space (Supplementary Fig. 22a, b), and the normalized pairwise distances between cells and Layer 1 (L1) increase layer by layer from L1 to WM (Supplementary Fig. 22c).

Next, the snRNA-seq data are classified into 19 clusters according to the given number of subclasses in original annotations using scSpace and Seurat, respectively. Among these clusters, scSpace has successfully distinguished two major subclasses of L6 (Supplementary Fig. 22d), L6 CT (C10) and L6b (C15), which are also separated from each other in the reconstructed pseudo-space (Supplementary Fig. 22e). However, the spatial heterogeneity is difficult to distinguish at the same clustering resolution by traditional methods such as Seurat (Supplementary Fig. 22f), which uses only transcriptional information with a clustering ARI score of−0.004 and an NMI score of 0.004, compared with that of 0.725 and 0.637 for scSpace, respectively (Supplementary Fig. 22g). Same as the previous strategy, we further attempt to increase the cluster number (K) of Seurat to avoid mis-classification caused by a low clustering resolution. As shown in Sup-plementary Fig. 22h, these two subclasses of L6 are eventually distinguished by Seurat when K increased to 26. Once again, the results confirm that the spatial information of each cell is crucial for the characterization of its cellular identity.

Notably, as illustrated in Supplementary Figs. 22i and 23, intrate-lencephalic (IT) neurons in the original dataset can be further classified into five subpopulations (C3, C1, C9, C4, and C2) based on their tran-scriptome and spatial characteristics. The scSpace analysis shows that these IT neuron subpopulations are distributed in all layers but accounted for different proportions (Supplementary Fig. 22j). More-over, the density centers of cell spatial distribution in C3, C1, C9, C4, and C2 moved from cortex L2 to L6 monotonously (Supplementary Fig. 22k). We select five target genes coupled with the histological staining images derived from the Allen Brain Atlas and explore their expression patterns in the five IT subpopulations (Supplementary Data 4), the results demonstrate that the spatial expression patterns of these target genes are consistent with the distribution of the corre-sponding subpopulations in layers (Supplementary Fig. 22l), which further supports the ability of space-informed clustering for scSpace.

## Spatial reconstruction of T cell pseudo-space revealed T cell exhaustion in melanoma

Melanoma is widely recognized as one of the most immunogenic human cancer types and a strong correlation between the infiltration of T cells, both in primary lesions and in melanoma metastases, and clinical outcome has been described[51–53]. We, therefore, perform scSpace to reconstruct the pseudo-space of cells in melanoma scRNA-seq data generated by Tirosh et al.[31] and further identify the spatial heterogeneity of 2064T cells. The ST data of the melanoma derived from another experiment[54] is utilized as the spatial reference.

As shown in Fig. 5a, b, T cells are clustered into five refined sub-populations by scSpace. Furthermore, the C5 subpopulation is sig-nificantly nearer than other subtypes to the malignant cells, while the C3 subpopulation is the opposite (Fig. 5c). This result is also observed in the spatial reference. Using the RCTD[20] algorithm to deconvolve the cell-type proportions of each spot in ST data, we discover that the C5 and C3 subpopulations are mainly located in the tumor and lymphatic regions, respectively (Fig. 5d). Notably, the similar spatial relationship between different T cell subpopulations and malignant cells is also revealed when utilizing the ST data from other patients as the spatial reference, suggesting the robustness and universality of the pseudo-space reconstruction by scSpace (Supplementary Fig. 24).

Compared with the C3 subpopulation which is far from the malignant cells, the C5 subpopulation highly expresses a large number of genes related to the dissemination and metastasis of melanoma (Fig. 5e and Supplementary Fig. 26), including thymidine kinase 1 (TK1)[55], NME/NM23 nucleoside diphosphate kinase 1 (NME1)[56], and immunoglobulin superfamily member 8 (IGSF8)[57]. Besides, multiple cell-cycle and apoptosis-related genes (AURKB, BIRC5, MKI67, CDK2, etc.) also exhibit a high expression level in the C5 subpopulation, and cyclin-dependent kinase-2 (CDK2), which has been identified as a bio-logical marker of melanoma, could be a potential therapeutic target in melanoma treatment. The research suggests that targeting CDK2 could suppress melanoma cell growth, induce apoptosis, and overcome melanoma resistance[58–61]. Congruously, Aurora kinase B (AURKB) is also crucial for melanoma proliferation, apoptosis, and cell cycle[62]. These differential expression genes indicate that the character of the "spatially adjacent to melanoma" of the C5 subpopulation may be associated with tumor-infiltrating[31].

Also, we attempt to identify T cell subclusters with Seurat[43], a classical clustering method for scRNA-seq data analysis, and then compare the results with scSpace. Unsurprisingly, the C5 subpopulation with high expression of a series of melanoma-related genes could not be accurately identified by Seurat under the same targeted clustering number setting (Supplementary Fig. 25), which in turn supports the superiority of scSpace in spatially heterogeneous subpopulation identification by reconstructing and considering the pseudo-space information of cells.

Furthermore, to determine the identity of the C5 subpopulation, we calculate the T exhaustion score for the C5 and C3 subpopulations with the exhaustion-related genes defined by Zheng et al.[63] T cell exhaustion is the loss of T cell function in patients with common chronic infections and cancer. As a result of long-term exposure to persistent antigens and inflammation, exhausted T cells gradually lose their effective function[64]. Our results illustrate that the C5 subpopulation had a higher T exhaustion score (Fig. 5f), suggesting this T cell subpopulation may be the tumor-infiltrating lymphocytes (TIL). The pathway enrichment analysis results indicate that the genes highly expressed in the C5 subpopulation are enriched in E2F targets, oxidative phosphorylation, and some cell-cycle-related pathways (Fig. 5h and Supplementary Fig. 26), which are closely related to the tumor microenvironment.

Finally, by linking the highly expressed genes of the C5 subpopulation with clinical outcomes, we demonstrate that patients with a higher expression level of these genes have significantly worse survival time than patients with a lower expression level (Fig. 5g). The results indicate that the C5 subpopulation may play an important role in the occurrence and development of melanoma, and these highly expressed genes are also expected to become potential therapeutic targets for precision medicine in clinics.

## Reconstruction of spatially resolved TSK-stroma communica-tions with scSpace

To further extend the applicability of scSpace, it is applied to recon-struct the spatial crosstalk between tumor-specific keratinocytes (TSKs) and tumor microenvironment (TME) cells over the scRNA-seq data of the human SCC generated by Ji et al.[44] A total of 17,738 single cells are retained from SCC samples (Supplementary Fig. 27a) and three replicated ST data of Patient 2 from the same experiment are used as the spatial reference (Supplementary Fig. 27b). As illustrated in Supplementary Fig. 27c, the TSKs, fibroblasts, and endothelial cells of scRNA-seq data show specific patterns of colocalizations in the pseudo-space, despite using different spatial references to perform spatial reconstruction by scSpace. In concordance with the previous study results[44] and the deconvolution results of the spatial reference calculated by RCTD (Supplementary Fig. 27d), fibroblasts and endo-thelial cells are enriched at the TSK-high leading edge, further

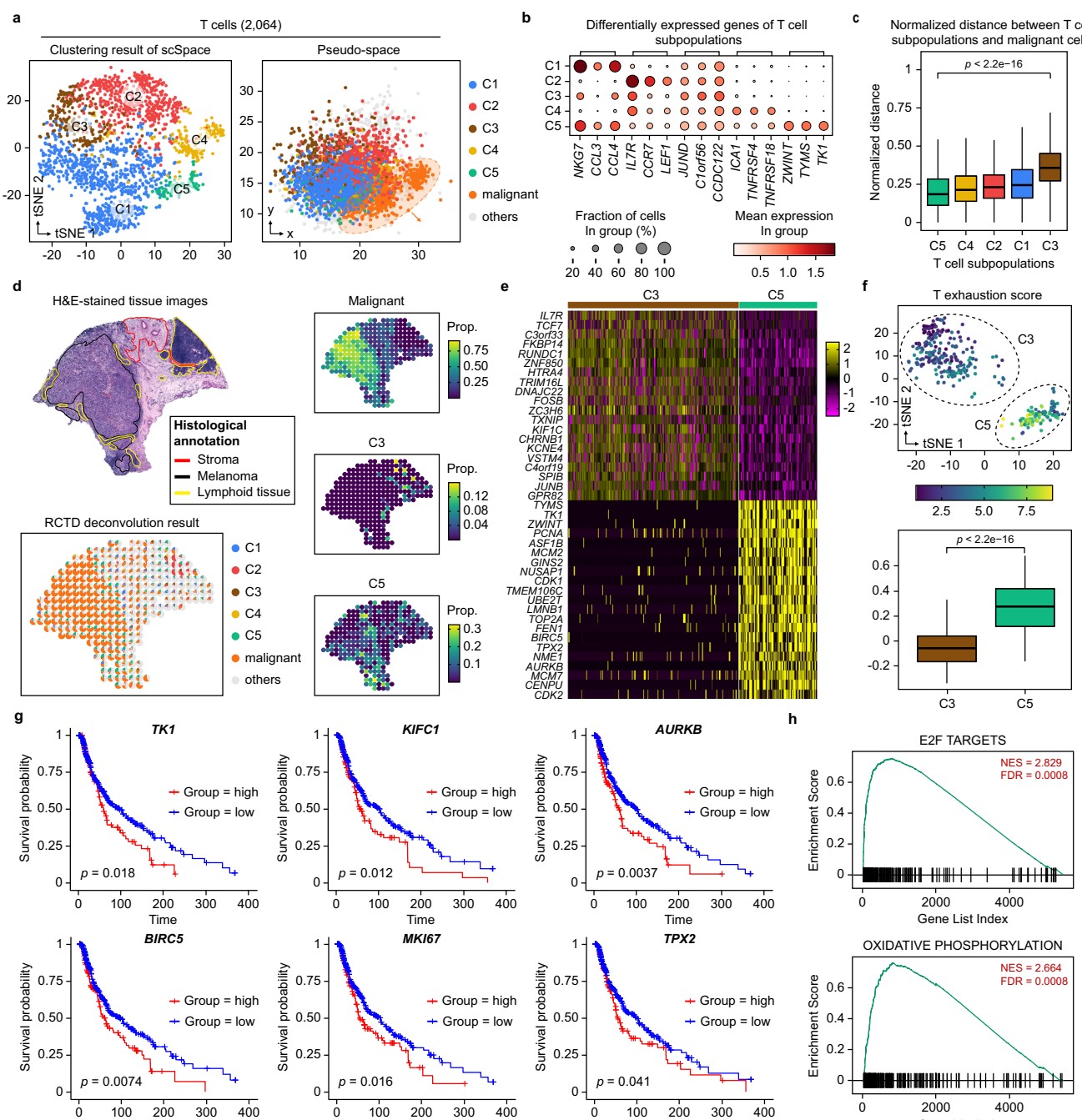

**Fig. 5 | Spatial reconstruction of T cell pseudo-space revealed T cell exhaustion in melanoma. a** t-SNE visualization (left) and the pseudo-space (right) of five T cell subpopulations identified by scSpace. All other cell types except T cells and malignant ones are labeled as "others". **b** Differentially expressed genes of five T cell subpopulations. **c** Normalized pairwise distance between different T cell subpopulations to malignant cells. Data are presented as boxplots (minima, 25th percentile, median, 75th percentile, and maxima). The number of data points for the normalized pairwise distances to malignant cells are 144,555, 251,400, 731,574, 1,148,898, and 318,021, respectively. *P*-value is calculated with the two-sided Wilcoxon rank-sum test (the exact *P*-value is 0). **d** The spatial distribution of C3 and C5 subpopulations in the spatial transcriptomics reference calculated by RCTD. All other cell types expect T cells and malignant ones are labeled as "others". **e** Heatmap of top 20 differentially expressed genes for C3 and C5 subpopulations. **f** Comparison of the T exhaustion score between C3 and C5 subpopulations. Data are presented as boxplots (minima, 25th percentile, median, 75th percentile, and maxima). The number of data points for the boxplots from left to right are 253 and 115, respectively. *P*-value is calculated with the two-sided Wilcoxon rank-sum test (the exact *P*-value is 3.0e−35). **g** Survival analysis of the six highly expressed genes of the C5 T cell subpopulations. *P*-value is calculated with the two-sided log-rank test. **h** Gene set enrichment analysis (GSEA) results of differentially expressed genes of the C5 T cell subpopulation.

supporting a fibrovascular niche surrounding TSK cells. Besides, the distribution pattern of the TSK score based on markers defined by Ji et al.[44] in the pseudo-space is also in agreement with the spatial structure of the TSK-proximal fibrovascular niche (Supplementary Fig. 27e).

Next, we introduce SpaTalk[65], a knowledge-graph-based method that can infer cell−cell communications from ST data, to analyze the cellular crosstalk in the pseudo-space of SCC scRNA-seq data reconstructed by scSpace. To validate whether the pseudo-space constructed by scSpace preserves the spatial interaction between different

cell types, we focus on the TSK-high leading-edge niche because TSKs are reported to participate in extensive autocrine and paracrine interactions (mostly with fibroblasts, endothelial cells, macrophages, and MDSCs)[44]. Consistent with the previous study, the major spatially resolved cell–cell communications between TSKs and stromal cells of the fibrovascular niche in the TME are retained in the pseudo-space of SCC scRNA-seq data. For instance, the prominent TSK signaling to fibroblasts and endothelial cells is mediated by several common ligand-receptor pairs which are close to each other in the pseudo-space, including *PGF-NRP1*, *TNC-SDC1*, *PGF-FLT1*, and *EFNB1-EPHB4* (Supplementary Fig. 28a). Conversely, fibroblasts and endothelial cells prominently co-expressed numerous ligands such as *TFPI*, *FN1*, *ITGB3*, and *MDK* (Supplementary Fig. 28b), matching TSK receptors that promote the proliferation and differentiation of TSKs. Notably, similar spatially resolved ligand–receptor interactions mediating the TSK-stroma communications are also observed in the pseudo-space constructed by scSpace using other spatial references (Supplementary Fig. 28c). Therefore, we further demonstrate that scSpace can precisely reconstruct the spatial arrangement of single cells as well as the spatially resolved cell–cell interactions.

### Reconstruction of cell pseudo-space captured invasion of myeloid subpopulations in COVID-19

To further examine whether scSpace could distinguish the spatial variation of cells between normal and disease conditions, we utilize a 10X Visium ST dataset of the normal lung[66] as the spatial reference and applied scSpace to a COVID-19 scRNA-seq dataset[32] from COVID-19 and normal samples with comparable expression states (Supplementary Fig. 29a). By projecting single cells in normal and diseased tissues into the same pseudo-space with scSpace, we can compare the cell-type composition and proportion, the spatial distribution patterns, and the relative pairwise associations between cell subpopulations. Therefore, a more accurate description of the process of the occurrence and development of the disease can be obtained and key potential therapeutic targets can be uncovered.

Specifically, the pseudo-space of the control and COVID-19 group is established by recovering the spatial relationship between cells by scSpace (Fig. 6a and Supplementary Fig. 29b). We focus the analysis on the myeloid cell and its subpopulations, whose dysregulation is an essential driving factor that leads to severe COVID-19 cases and subsequent death[67-71]. As shown in Fig. 6b, in the reconstructed pseudo-space, the normalized distance between myeloid cells and alveolar/airway epithelial cells is significantly reduced in the COVID-19 group, suggesting that severe immune infiltration may occur. This spatial variability between the control and COVID-19 group identified by scSpace is validated on a GeoMx DSP targeted ST dataset generated by Rendeiro et al.[72] Compared with the control group, we observe a significant increase in myeloid cell abundance in the regions of interest (ROIs) of the alveolar and airway with COVID-19 by single sample gene set enrichment analysis (ssGSEA) and CIBERSORTx[73] deconvolutions, respectively (Supplementary Fig. 30a, b).

Differentially expressed genes are then calculated between the control and COVID-19 group using this GeoMx DSP targeted ST data, and the highly expressed genes are defined in the COVID-19 group as the "COVID-19 signatures" (Supplementary Fig. 30c and Supplementary Data 6). Thus, the accuracy of the spatial variability identified by scSpace can be further evaluated via comparing the expression levels of these signatures in all myeloid cells in the COVID-19 group. Based on the normalized distance to the alveolar/airway epithelial cell in the pseudo-space, myeloid cells are further classified into two populations, where the cells with a close average distance to the alveolar/airway epithelial cell (the top 50%) are defined as the "near to epithelial cell" populations and the rest as "not near to epithelial cell" populations (Fig. 6c and Supplementary Fig. 30d). Unsurprisingly, the "near to epithelial cell" myeloid population shows higher COVID-19 signatures

score (Fig. 6d) and highly expresses multiple MHC Class I and II related genes, such as *B2M*, *HLA-A*, *HLA-B*, *HLA-C*, *HLA-E*, and *HLA-DRA*, which are associated with the interferon (IFN) response (Supplementary Fig. 30f). The comparison indicates that "near to epithelial cell" myeloid population is more likely to be recruited into the alveolar and airway to produce infiltration due to the robust IFN response caused by SARS-CoV-2 viral infection[74].

According to the original publication[32], myeloid cells can be further divided into four subtypes, including resident alveolar macrophages (AM), monocytes (Mon), monocyte-derived macrophages (MDM), and transitioning MDM (TMDM). We thus compare the normalized distance between each subtype and the alveolar/airway epithelial cell in the COVID-19 group with the control group. The result demonstrates that the normalized distance between all four myeloid subtypes and the alveolar/airway epithelial cell is significantly reduced in the COVID-19 group compared with the control group (Fig. 6e). Besides, the distance reduction ratios of the MDM and the TMDM change more than that of the AM and the Mon (Fig. 6f), consistently with previous findings. The results suggest that when infected with COVID-19, monocytes would differentiate into MDM and TMDM with damage response and tissue repair signatures and accumulate in the alveolar and airway[32,75]. Notably, AM appeared to be closest to the alveolar/airway epithelial cell in the pseudo-space in both control and COVID-19 groups, which is in line with the fact that these self-renew tissue-resident macrophages (MFs) arise from fetal monocytes and highly enrich in the alveolar[76], indicating an accurate performance of scSpace on the pseudo-space reconstruction.

To resolve the spatial heterogeneity of MDM/TMDM, 1,149 MDM, and TMDM cells are clustered into six subpopulations with scSpace (Fig. 6g, h). Specifically, the C4 subpopulation, which is nearest to the alveolar/airway epithelial cell, is identified by scSpace (Fig. 6i). Compared with other cell subpopulations, C4 highly expresses multiple mitochondria-related and MHC Class I-related genes, such as *B2M*, *HLA-DRA*, *CD74*, and *CTSS* (Fig. 6j and Supplementary Fig. 29c, d). In addition, pathway enrichment analysis demonstrates that C4 has higher activity in oxidative phosphorylation, proton transmembrane transport, energy metabolism, oxidative stress, and MHC class II antigen processing and presentation pathways (Fig. 6k), suggesting that the C4 subpopulation may be highly and aberrantly activated[32] and thus play a key role in the process of COVID-19.

## Discussion

In this study, we have demonstrated the utility of scSpace in reconstructing the pseudo-space of single cells and identifying spatially variable cell subpopulations with similar expression profiles in scRNA-seq data. While there are similarities in the space-informed clustering between scSpace and other spatial domain identification methods such as SpaGCN[35], BayesSpace[37], DR-SC[38], etc., we highlight the fundamental difference between the methods. Specifically, scSpace focuses on the spatial analysis of scRNA-seq data whereas other methods target the spatial domain identification of ST data. By introducing the concept of "pseudo-space", scSpace can efficiently recover the spatial association between single cells and identify spatially heterogeneous cell subpopulations within the scRNA-seq data. However, other spatial domain identification methods, which are designed more for ST data, only detect the spatial domains with coherent expression and histology but don't attach the importance of cell independence.

In addition, even though current targeted methods such as MERFISH and seqFISH can measure over 10,000 RNA species as well as observe at subcellular resolution, they fail to detect gene variations, which directly reflect the function of specific genes. Simultaneously, other in situ short-read sequencing methods such as FISSEQ and ISS sequence very short segments of a transcript, which leads to a bias of results. Also, even though these methods show success in the in situ sequencing of cultured cells, they can hardly be applied to tissue

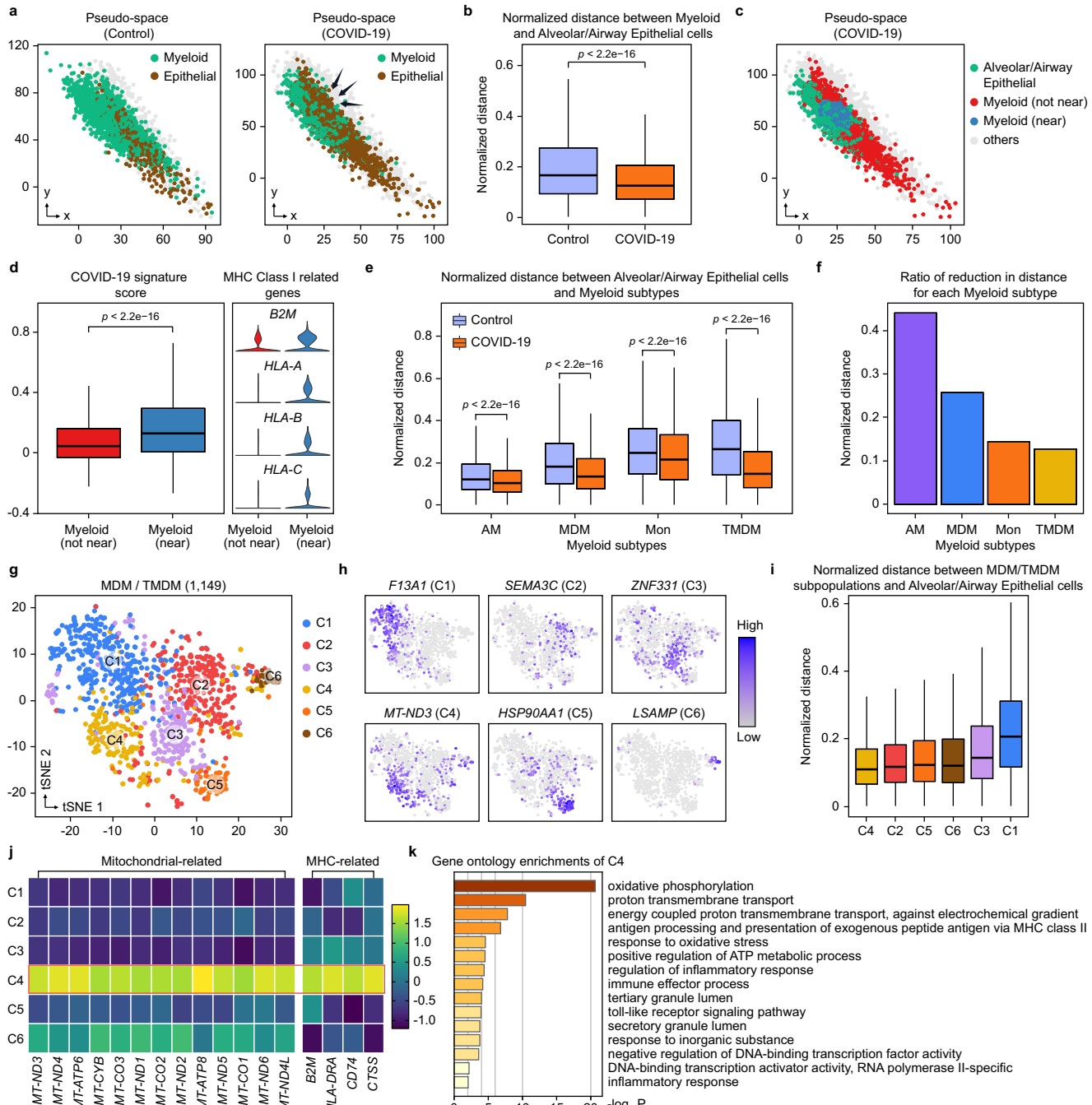

**Fig. 6 | scSpace captures the invasion of myeloid subpopulations in COVID-19.**
**a** Reconstructed pseudo-space for control and COVID-19 groups. All other cell types are labeled as "others". **b** Normalized pairwise distance between myeloid cells and alveolar/airway epithelial cells in the control and COVID-19 groups. Data are presented as boxplots (minima, 25th percentile, median, 75th percentile, and maxima). The number of data points for the boxplots from left to right are 548,877 and 2,273,544, respectively. *P*-value is calculated with the two-sided Wilcoxon rank-sum test (the exact *P*-value is 0). **c** Pseudo-space for the COVID-19 group, wherein myeloid cells are divided into "near to epithelial" and "not near to epithelial" according to their average distances to alveolar/airway epithelial cells. All other cell types are labeled as "others". **d** COVID-19 signature scores (Boxplot) and expression levels of MHC-related genes (Violin) for two types of myeloid cells. Data are presented as boxplots (minima, 25th percentile, median, 75th percentile, and maxima). The number of data points for each type of myeloid cell is 1401. *P*-value is calculated with the two-sided Wilcoxon rank-sum test (the exact *P*-value is 4.1e−20). **e** Normalized pairwise distances between myeloid subtypes and alveolar/airway epithelial cells in control and COVID-19 groups. Data are presented as boxplots

(minima, 25th percentile, median, 75th percentile, and maxima). The number of data points for the control group is 192,705, 237,891, 82,398, and 35,883, respectively; the number of data points for the COVID-19 group is 1,017,744, 677,040, 226,044, and 352,716, respectively. *P*-value is calculated with the two-sided Wilcoxon rank-sum test (the exact *P*-values from left to right are both 0). **f** Ratio of reduction in the distance between each myeloid subtype and alveolar/airway epithelial cells in the COVID-19 group compared to the control group. **g**, t-SNE visualization of six MDM/TMDM subpopulations identified by scSpace. **h**, Feature plot of differentially expressed genes for six MDM/TMDM subpopulations. **i** Normalized pairwise distance between MDM/TMDM subpopulations and alveolar/airway epithelial cells. Data are presented as boxplots (minima, 25th percentile, median, 75th percentile, and maxima). The number of data points for the boxplots from left to right are 450,306, 704,511, 191,259, 79,893, 476,937, and 878,823, respectively. **j** Scaled averaged expression level of mitochondrial-related and MHC-related genes in six MDM/TMDM subpopulations. **k** Significantly enriched gene ontology (GO) pathway terms for the differentially expressed genes of the C4 subpopulation. *P*-value is calculated with the hypergeometric test.

sections. Moreover, scRNA-seq data still dominates the cutting edge of omics technology, with unmet biological and clinical needs. Computational methods such as scSpace that employ ST data to enhance the spatial information that is ignored by scRNA-seq data are desirable and meaningful.

As a reference-based method, scSpace applies to spatial reference profiled by various technologies, including STARmap[42], "Spatial Transcriptomics"[11], 10X Visium, and Slide-seq v2[15]. Since the choice of spatial reference would directly influence the spatial reconstruction, an important feature of scSpace is the ability to maintain robustness when biological replicates (or samples) from different sources but with high spatially shared coherence are utilized as spatial references. For example, while reconstructing the pseudo-space of melanoma, spatial references are derived from different biological replicates, however, the performance of scSpace on spatial variability identification is conserved. Similar results are revealed in the spatial reconstruction of the kidney, where the spatial references are profiled by 10X Visium and Slide-seq v2 from separate experiments, respectively. However, there are still aspects that need to be attended to in performing pseudo-space reconstruction by scSpace. Firstly, scSpace should be applied cautiously to tissues with low spatial coherence, such as Patients 5 and 9 of SCC evaluated in this paper. The intrinsic difference in the spatial homogeneity between tissues may influence the accuracy of scSpace. Secondly, as the spatial architectures of different regions from the same tissue may also be heterogeneous (for example, while DLPFC and dACC both belong to the cortex, their spatial architectures are quite different, as dACC is agranular and lacks cortical layer 4), performing an appropriateness assessment or necessary pre-processing steps of the spatial reference should be considered before applying scSpace.

Identifying the spatially heterogeneous cell subpopulations from single-cell data can give rise to breakthroughs in understanding cell identities and disease mechanisms. These spatially variable cell subpopulations play an important role in the progress of diseases. For instance, a subpopulation of oligodendrocytes located in specific areas enriched with Aβ plaques is verified to strongly associate with Alzheimer's disease[77]. However, they are difficult to identify by traditional single-cell analysis pipelines due to the similarity in transcriptome. To this end, scSpace constructs and leverages the pseudo-space information of single cells to identify the spatial heterogeneity between cell subpopulations. In this paper, we have demonstrated the ability of scSpace to identify the spatially heterogeneous cell subpopulation that may relate to diseases with two main applications, namely melanoma and COVID-19. In the former analysis, we have identified a T cell subpopulation in melanoma, revealed the occurrence of tumor-infiltrating near the TME by scSpace, and uncovered several potential therapeutic targets for precision medicine in clinics. In the latter analysis, scSpace is applied to identify an MDM subpopulation with significant immune infiltration characteristics that are highly enriched in the alveoli and airway in space. The use of scSpace has provided spatial therapeutic markers and brought insight into the study of COVID-19. In a word, these cases convincingly demonstrate the universality of scSpace in identifying the spatially heterogeneous cell subpopulations from scRNA-seq data of normal or disease tissues.

Finally, even though scSpace focuses on transcriptomics data, theoretically, it should apply to other single-cell-omics data such as single-cell chromatin accessibility sequencing (scATAC-seq)[78,79] and single-nucleus methylome sequencing (snmC-seq)[80], if appropriate spatial references are provided. We have also made attempts to reconstruct the pseudo-space of cells with scATAC-seq data from the mouse cerebral cortex generated by ISSAAC-seq[81]. Because the appropriate spatial ATAC-seq data is not available, we chose the spatial reference with the ST data and convert the scATAC-seq data into a gene activity matrix using Signac[82] in advance. In conclusion, scSpace successfully reproduces the spatial distribution of cells that consistently with their layered patterns (Supplementary Fig. 31). The promising result further demonstrates the extensive application of scSpace in other single-cell-omics data.

## Methods

### Data collection and processing

All sc/sn RNA-seq and spatial transcriptomics datasets used in this study are collected from high-quality publications and Gene Expression Omnibus (GEO). The detailed description of each dataset is summarized in Supplementary Data 1. For the human skin squamous cell carcinoma (SCC) scRNA-seq data[44], we retain 17,738 single cells from SCC samples. For the mouse kidney scRNA-seq data[49], we only retain the single cells belonging to the nephrogenic lineage (Ontology ID: 1–19) and ureteric lineage (Ontology ID: 20–32). Besides, we only retain the single cells from male donors, which is consistent with the sex of the donor of the spatial reference. Cluster 3, cluster 5, and cluster 7, which also belong to females in the gene ontology annotation are removed. For the mouse cortex scATAC-seq data[81], we convert the data into a gene activity matrix using Signac[82] with default parameters. For the human brain cortex scRNA-seq data of multiple cortical areas (MTG, ACC, V1C, M1C, S1C, and A1C), we randomly down-sample them to 4000 cells for computational efficiency. For the human lung scRNA-seq data of lethal COVID-19[32], we randomly down-sample them to 10,000 cells for computational efficiency. For the COVID-19 GeoMx DSP targeted ST data[72], we exclude 67 samples of "Non-viral" group and further combine the "COVID-19 High" and "COVID-19 Low" groups into the "COVID-19" group.

### Computing environment

**The workstation for developing scSpace is listed below.** Dell Precision Tower 7929 Workstation (CPU (Intel Xeon Gold 6230, 2.1 GHz × 2), RAM (192 GB, 16 GB × 12, DDR4, 2933 MHz), Hard Drive (SSD, SATA Class 20, 512 GB; HDD, 7200 rpm, SATA), Graphics Card (NVIDIA GeForce RTX 3090, 24 GB), Operating System (Ubuntu 18.04)). Running Environment (CUDA 11.3, Torch 1.12.1, Python 3.8.5, numpy 1.23.4, pandas 1.5.0, scanpy 1.9.1, scikit-learn 1.1.2, scipy 1.9.2, tqdm 4.46.1, igraph 0.10.2, leidenalg 0.9.0).

### Design of scSpace

The scSpace model comprises three components: (1) latent biological feature representation extraction, (2) spatial reconstruction, and (3) space-informed clustering.

### Latent biological feature representation extraction

Given the scRNA-seq data and the spatial reference, scSpace first applies a transfer learning method TCA[28] to eliminate the batch effect of these two types of data and extract the shared feature representation across these two domains with true biological characteristics.

We denote the scRNA-seq data as $X_S = \{\mathbf{x}_{S_1}, \mathbf{x}_{S_2}, \ldots, \mathbf{x}_{S_{n_1}}\}$ and the spatial transcriptomics reference as $X_T = \{\mathbf{x}_{T_1}, \mathbf{x}_{T_2}, \ldots, \mathbf{x}_{T_{n_2}}\}$. Here, $\mathbf{x}_{S_i}, \mathbf{x}_{T_i} \in \mathbb{R}^N$ represent the gene expression vectors of the cell $S_i$ in scRNA-seq data and the spot (or cell) $T_i$ in spatial reference respectively, $n_1$ and $n_2$ represent the number of cells and spots, and $N$ is the number of genes. Let $P(X_S)$ and $Q(X_T)$ be the distributions of $X_S$ and $X_T$. Since scRNA-seq and ST data are obtained from different techniques, we claim that $P(X_S)$ is quite different from $Q(X_T)$.

We assume that there exists a transformation $\phi : \mathbb{R}^N \rightarrow \mathcal{H}$, through which the distributions of $X_S$ and $X_T$ can be similar and the data configuration for the two domains is preserved. Here $\mathcal{H}$ is a universal Reproducing Kernel Hilbert Space (RKHS)[83]. Let $X'_S = \{\mathbf{x}'_{S_i}\} = \{\phi(\mathbf{x}_{S_i})\}$ and $X'_T = \{\mathbf{x}'_{T_i}\} = \{\phi(\mathbf{x}_{T_i})\}$ be the transformed input sets, we expect that $P'(X'_S) \approx Q'(X'_T)$.

To circumvent complex density estimation, we apply a nonparametric method, maximum mean discrepancy (MMD)[84], to estimate the distance between these two distributions:

$$\text{Dist}(X'_{S}, X'_{T}) = \| \frac{1}{n_1} \sum_{i=1}^{n_1} \phi(\mathbf{x}_{S_i}) - \frac{1}{n_2} \sum_{i=1}^{n_2} \phi(\mathbf{x}_{T_i}) \|_{\mathscr{H}}^2. \quad (1)$$

Where $| \cdot |_{\mathscr{H}}$ is the RKHS norm. Then the optimal nonlinear mapping $\phi$ can be obtained by minimizing this distance. Benefiting from the kernel trick, instead of calculating the nonlinear transformation $\phi$ explicitly, the above equation can be written in terms of the kernel matrices as

$$
\begin{aligned}
\text{Dist}(X'_{S}, X'_{T}) = & \left\| \frac{1}{n_1}\left[\phi(\mathbf{x}_{S_1})\phi(\mathbf{x}_{S_2})\cdots\phi(\mathbf{x}_{S_{n_1}})\right]_{1\times n_1} \begin{bmatrix}1\\1\\\vdots\\1\end{bmatrix}_{n_1\times 1} \right. \\
& \left. - \frac{1}{n_2}\left[\phi(\mathbf{x}_{T_1})\phi(\mathbf{x}_{T_2})\cdots\phi(\mathbf{x}_{T_{n_2}})\right]_{1\times n_2} \begin{bmatrix}1\\1\\\vdots\\1\end{bmatrix}_{n_2\times 1} \right\|_{\mathscr{H}}^2 \\
= & \, \text{tr}\left( \frac{1}{n_1^2}\phi(\mathbf{x}_S)\mathbf{1}(\phi(\mathbf{x}_S)\mathbf{1})^T + \frac{1}{n_2^2}\phi(\mathbf{x}_T)\mathbf{1}(\phi(\mathbf{x}_T)\mathbf{1})^T \right. \\
& \left. - \frac{1}{n_1 n_2}\phi(\mathbf{x}_S)\mathbf{1}(\phi(\mathbf{x}_T)\mathbf{1})^T - \frac{1}{n_1 n_2}\phi(\mathbf{x}_T)\mathbf{1}(\phi(\mathbf{x}_S)\mathbf{1})^T \right) \\
= & \, \text{tr}\left( \frac{1}{n_1^2}\phi(\mathbf{x}_S)\mathbf{1}\mathbf{1}^T\phi(\mathbf{x}_S)^T + \frac{1}{n_2^2}\phi(\mathbf{x}_T)\mathbf{1}\mathbf{1}^T\phi(\mathbf{x}_T)^T \right. \\
& \left. - \frac{1}{n_1 n_2}\phi(\mathbf{x}_S)\mathbf{1}\mathbf{1}^T\phi(\mathbf{x}_T)^T - \frac{1}{n_1 n_2}\phi(\mathbf{x}_T)\mathbf{1}\mathbf{1}^T\phi(\mathbf{x}_S)^T \right) \\
= & \, \text{tr}\left( \frac{1}{n_1^2}\mathbf{1}\mathbf{1}^T\phi(\mathbf{x}_S)^T\phi(\mathbf{x}_S) + \frac{1}{n_2^2}\mathbf{1}\mathbf{1}^T\phi(\mathbf{x}_T)^T\phi(\mathbf{x}_T) \right. \\
& \left. - \frac{1}{n_1 n_2}\mathbf{1}\mathbf{1}^T\phi(\mathbf{x}_T)^T\phi(\mathbf{x}_S) - \frac{1}{n_1 n_2}\mathbf{1}\mathbf{1}^T\phi(\mathbf{x}_S)^T\phi(\mathbf{x}_T) \right) \\
= & \, \text{tr}\left( [\phi(\mathbf{x}_S)\,\phi(\mathbf{x}_T)]\begin{bmatrix}\frac{1}{n_1^2}\mathbf{1}\mathbf{1}^T & -\frac{1}{n_1 n_2}\mathbf{1}\mathbf{1}^T\\ -\frac{1}{n_1 n_2}\mathbf{1}\mathbf{1}^T & \frac{1}{n_2^2}\mathbf{1}\mathbf{1}^T\end{bmatrix}\begin{bmatrix}\phi(\mathbf{x}_S)\\\phi(\mathbf{x}_T)\end{bmatrix} \right) \\
= & \, \text{tr}\left( \begin{bmatrix}\phi(\mathbf{x}_S)\\\phi(\mathbf{x}_T)\end{bmatrix}[\phi(\mathbf{x}_S)\,\phi(\mathbf{x}_T)]\begin{bmatrix}\frac{1}{n_1^2}\mathbf{1}\mathbf{1}^T & -\frac{1}{n_1 n_2}\mathbf{1}\mathbf{1}^T\\ -\frac{1}{n_1 n_2}\mathbf{1}\mathbf{1}^T & \frac{1}{n_2^2}\mathbf{1}\mathbf{1}^T\end{bmatrix} \right) \\
= & \, \text{tr}\left( \begin{bmatrix}<\phi(\mathbf{x}_S),\phi(\mathbf{x}_S)> & <\phi(\mathbf{x}_S),\phi(\mathbf{x}_T)>\\ <\phi(\mathbf{x}_T),\phi(\mathbf{x}_S)> & <\phi(\mathbf{x}_T),\phi(\mathbf{x}_T)>\end{bmatrix}\begin{bmatrix}\frac{1}{n_1^2}\mathbf{1}\mathbf{1}^T & -\frac{1}{n_1 n_2}\mathbf{1}\mathbf{1}^T\\ -\frac{1}{n_1 n_2}\mathbf{1}\mathbf{1}^T & \frac{1}{n_2^2}\mathbf{1}\mathbf{1}^T\end{bmatrix} \right) \\
= & \, \text{tr}\left( \begin{bmatrix}K_{S,S} & K_{S,T}\\ K_{T,S} & K_{T,T}\end{bmatrix} L \right) \\
= & \, \text{tr}(KL).
\end{aligned}
\quad (2)
$$

where $K = \begin{bmatrix}K_{S,S} & K_{S,T}\\ K_{T,S} & K_{T,T}\end{bmatrix} \in \mathbb{R}^{(n_1+n_2)\times(n_1+n_2)}$ is a kernel matrix, $K_{S,S}$, $K_{T,T}$ and $K_{S,T}$ are the kernel matrices defined by $\phi(\mathbf{x}_S)$ and $\phi(\mathbf{x}_T)$, and $(L)_{ij} = \begin{cases} \frac{1}{n_1^2} & \mathbf{x}_i,\mathbf{x}_j \in X_S \\ \frac{1}{n_2^2} & \mathbf{x}_i,\mathbf{x}_j \in X_T \\ -\frac{1}{n_1 n_2} & \text{otherwise} \end{cases}$ . Then the calculation of the inner product of $\phi(\mathbf{x})$ can be solved by learning the kernel matrix $K$, which can be formulated as a semi-definite program (SDP)[85] problem. However, it still leads to huge computational resource because $K$ is required to be positive semi-definite.

Note that according to the empirical kernel map[86], the kernel matrix $K$ can be decomposed as $K = (KK^{-1/2})(K^{-1/2}K)$. A more light-weight solution is to conduct dimensionality reduction to transform the empirical kernel map features vectors to an $m$-dimensional space with a matrix $\widetilde{W} \in \mathbb{R}^{(n_1+n_2)\times m}$, where $m \ll n_1+n_2$. Then the kernel matrix can be rewritten as:

$$\widetilde{K} = \left(KK^{-\frac{1}{2}}\widetilde{W}\right)\left(\widetilde{W}^T K^{-\frac{1}{2}}K\right) = KWW^T K. \quad (3)$$

Where $W = K^{-\frac{1}{2}}\widetilde{W} \in \mathbb{R}^{(n_1+n_2)\times m}$. Then we reformulate the MMD distance of the two distributions as:

$$\text{Dist}(X'_S, X'_T) = \text{tr}\left(\left(KWW^T K\right)L\right) = \text{tr}\left(W^T KLKW\right). \quad (4)$$

In order to control the complexity of $W$, we add a regularization term $\text{tr}\left(W^T W\right)$ to the objection function to avoid the rank deficiency of the denominator in the generalized eigendecomposition. The final kernel learning problem reduces to:

$$\min_{W} \text{tr}\left(W^T KLKW\right) + \mu\text{tr}\left(W^T W\right). \quad (5)$$

$$\text{s.t. } W^T KHKW = I$$

Where $\mu$ is a tradeoff parameter, $I \in \mathbb{R}^{m\times m}$ is the identity matrix, $H = I_{n_1+n_2} - (\frac{1}{n_1+n_2})\mathbf{1}\mathbf{1}^T$ is the centering matrix where $I_{n_1+n_2} \in \mathbb{R}^{(n_1+n_2)\times(n_1+n_2)}$. Note that the constraint $W^T KHKW = I$ is introduced to avoid the trivial solution ($W=0$), so that the transformed patterns do not collapse to one point.

The Lagrangian of the above optimization problem is:

$$
\begin{aligned}
& \text{tr}\left(W^T KLKW\right) + \mu\text{tr}\left(W^T W\right) - \text{tr}\left((W^T KHKW - I)Z\right) \\
& = \text{tr}\left(W^T(I + \mu KLK)W\right) - \text{tr}\left((W^T KHKW - I)Z\right).
\end{aligned}
\quad (6)
$$

Where $Z$ is a diagonal matrix whose diagonal entries are the Lagrange multipliers. Taking the derivative of the above Lagrangian with respect to $W$ and setting the derivative to 0, we obtain $(I + \mu KLK)W = KHKWZ$. Multiply both sides of the equation by $(KHK)^{-1}$, we can further obtain $(KHK)^{-1}(I + \mu KLK)W = WZ$, which is a generalized eigenvalue problem. The solution of $W \in \mathbb{R}^{(n_1+n_2)\times m}$ is the eigenvectors corresponding to the $m$ smallest eigenvalues of $(KHK)^{-1}(I + \mu KLK)$.

Therefore, the overall computational process can be summarized as follows. We first collect the scRNA-seq data $X_S$ and the spatial transcriptomics reference data $X_T$, from which the matrices $L$, $K$, and $H$ can be obtained. Then we can construct the transformation matrix W by selecting the eigenvectors corresponding to the top $m$ smallest eigenvalues of $(KHK)^{-1}(I + \mu KLK)$. Finally, with the transformation matrix W, the data from two domains are mapped into the lower dimensional latent space $W^T K \in \mathbb{R}^{m\times(n_1+n_2)}$. After transposing and dividing $W^T K$, we obtain the latent feature representation $X'_S \in \mathbb{R}^{n_1\times m}$ for scRNA-seq data and $X'_T \in \mathbb{R}^{n_2\times m}$ for spatial transcriptomics data with true biological characteristics.

**Spatial reconstruction**

Once the latent biological feature representation across scRNA-seq ($X_S$) and spatial transcriptomics data ($X_T$) are extracted, a multi-layer perceptron (MLP) model is applied to spatial reconstruction of scRNA-seq data. We assume that the spatial locations of cells are related to their latent biological feature representation:

$$[X, Y] \sim X'_T. \quad (7)$$

Where $X$ and $Y$ are spatial coordinates of spots (or cells) in spatial transcriptomics data $X_T$, and $X'_T$ represents the latent biological feature representation extracted from spatial transcriptomics data.

scSpace applies a two-layer fully connected model, where the first layer is followed by a sigmoid (or ReLU) activation. The number of neurons in the first layer is 128. The size of the input layer of this model is equal to the dimension of the latent biological feature representation ($m$), and the size of the output layer is corresponding to the dimension of spatial coordinates. We use the Adam optimizer with the initial learning rate of 1e−3 and the betas parameters of 0.9 and 0.999.

The loss function to be optimized is mean squared error (MSE) loss. using the shared biological characteristics extracted from spatial transcriptomics data, $X'_T$, scSpace trains the model to learn the relationship between characteristics and spatial coordinates. Next, the trained model is applied to $X'_S$, the feature representation extracted from scRNA-seq data. Thus, the spatial information of every single cell is reconstructed (we term it "pseudo-space").

## Space-informed clustering

scSpace applies space-informed clustering to identify spatially heterogeneous single-cell subpopulations based on the gene expression and the generated pseudo-space information of cells in scRNA-seq data. In detail, a gene expression graph $G_g(V, E_1)$ is first constructed on the reduced principal components derived from normalized gene expression using $k$-nearest neighbor (KNN) algorithm (top 50 PCs are selected by default). Since our goal is to find spatially heterogeneous subpopulations that may be similar in gene expression, the pseudo space information of cells is expected to be transformed to the spatial weight $w$ of each edge in gene expression graph $G_g(V, E_1)$:

$$w = W(E_1). \tag{8}$$

So that the spatial relationship of cells could be considered in the later unsupervised clustering step. Here, we provide two strategies to define spatial weight $w$.

(1) The spatial weight of edge $E_{i,j}$ between cell $S_i$ and cell $S_j$ is negatively associated with their direct distance $d_{i,j}$ in the pseudo space, which is defined:

$$w_{i,j} = \exp\left(-\frac{d_{i,j}^2}{2l^2}\right). \tag{9}$$

The hyperparameter $l$, also known as the characteristic length scale, determines how rapidly the covariance decays as a function of distance. This framework that works with distances has also been employed in SpatialDE[40].

(2) A space graph $G_S(V, E_2)$ is constructed on the pseudo space of cells using $k$-nearest neighbor (KNN) algorithm firstly, and the spatial weight of edge $E_{i,j}$ between cell $S_i$ and cell $S_j$ is negatively associated with their distance $d_{i,j}$ on the space graph $G_S(V, E_2)$, which is defined:

$$w_{i,j} = \frac{1}{\alpha + d_{i,j}} + \beta. \tag{10}$$

where $\alpha$ and $\beta$ are pseudocounts to guard against excessively large and small weights, respectively[87]. By default, $\alpha = \beta = 1$. The distance $d_{i,j}$ is calculated based on the adjacency matrix A of the space graph. Specifically, for given cell $S_i$ and cell $S_j$, the distance $d_{i,j} = 1$ if $S_j$ is the neighbor of $S_i$ on the space graph or $d_{i,j} = 2$ if $S_j$ is the neighbor of the neighbor of $S_i$ and so forth.

Finally, scSpace applies unsupervised clustering on space-weighted gene expression graph using Leiden algorithm[33].

## Simulated data analysis

We apply Splatter R package (v1.16.1)[34] to simulate 140 paired scRNA-seq and ST data with 5000 expression genes. To simulate the original batch effect between these two types of data, we set up two batches representing scRNA-seq and ST data, respectively. To simulate the cell subclusters with similar transcriptome profiles but being spatially heterogeneous, we random set 2 to 10 cell populations as spatially heterogeneous subclusters and set the probabilities of a gene being differentially expressed in each of them as 0.01 (the differentially expressed probabilities are set as 0.2 in cell populations with different transcriptome profiles by contrast).

Next, for each cell in ST data, a pseudo spatial coordinate is assigned based on random sampling and normal distribution strategy. Specifically, given a simulated spatial transcriptomic with $N$ cell populations, we random sample $N$ points in the $20 \times 20$ range as spatial coordinate centers of cell populations. Then for cells in each cell population, we randomly generate the spatial coordinates for the normal distribution with a mean equal to the spatial coordinate centers of this cell population and a standard deviation equal to 1.

Moreover, for the robustness of the results, we set a gradient from 500 to 1500 for the number of cells and a gradient from 3 to 14 for the number of cell populations. The detailed description of the experimental design is summarized in Supplementary Fig. 1 and Supplementary Data 2.

To evaluate the space reconstruction results of scSpace, we compute the pairwise distance of cells in original space and pseudo space, respectively, and then calculate the Pearson correlation coefficient (PCC). To evaluate the space-informed clustering results of scSpace, we compare scSpace with three classical clustering algorithms for scRNA-seq data, Louvain, $K$-means, and Hierarchical clustering, four spatial domain identification methods for ST data, SpaGCN[35], STAGATE[36], BayesSpace[37], and DR-SC[38], and one latent space learning algorithm for gene expression data, scCoGAPS[39]. For scCoGAPS, to make the results comparable, we replace the latent biological feature representation extracted by TCA with the output of scCoGAPS, and the remaining steps are the same as scSpace. The number of iterations is set to 1000 and the number of patterns is set to 50 for scCoGAPS. All other methods are set with default parameters. We use the adjusted rand index (ARI) measurement to evaluate the performance of clustering results.

## Differential gene expression analysis

For the scRNA-seq and ST data, the differentially expressed genes between cell populations/subpopulations or spatial domains are calculated using the "FindAllMarkers" function in Seurat R package (v4.1.0)[43] with the default two-tailed Wilcoxon rank sum test. For the GeoMx DSP targeted ST data generated by Rendeiro et al.[72], the differentially expressed genes between the control and COVID-19 groups are calculated by limma R package (v3.52.4). Genes with log2FC less than 0.5 or FDR greater than 0.05 are filtered.

## Pathway enrichment analysis

The Metascape web tool (https://metascape.org)[88] is used to perform the enrichment analysis of pathways and biological processes, wherein the differentially expressed genes of each subpopulation (log2FC > 0.5, FDR < 0.05) are selected. The gene set enrichment analysis (GSEA)[89] is performed using the fgsea R package (v1.18.0), whose hallmark gene sets are downloaded from the Molecular Signatures Database (MSigDB v7.4)[90] using msigdbr R package (v7.4.1).

## Survival analysis

For survival analysis, RNA-seq and clinical data of melanoma patients (cancer study id: skcm tcga) are obtained from TCGA using the cgdsr R package (v1.3.0). The samples are divided into two groups along with low (25%) and high (75%) target genes expression for all patients, and then survival curves of these two groups of patients are estimated by the Kaplan–Meier method using the survival R package (v3.2-13).

## Signature enrichment score calculation

The signature enrichment scores are calculated using the "AddModuleScore" function in Seurat R package (v4.1.0)[43] with default parameters. For T cell exhaustion and TSK, the signature genes are obtained from the original publications by Zheng et al.[63] and Ji et al.[44], respectively. For COVID-19, the signature genes are defined as the differentially expressed genes between the control and COVID-19 groups of GeoMx DSP targeted ST data.

## Spatial transcriptomics data deconvolution analysis

For the human melanoma, the mouse kidney, and the human SCC, we use RCTD[20] to deconvolve the cell type proportions of each spot in spatial transcriptomics data. The "CELL_MIN_INSTANCE" parameter is set to 5 and all other parameters follow the default values.

## Spatially resolved cell–cell interactions analysis

For the human SCC scRNA-seq data, TSKs, fibroblasts, and endothelial cells are extracted to perform spatially resolved cell–cell interactions analysis using SpaTalk R package (v1.0)[65] with default parameters. The complete lists of inferred ligand–receptor interaction pairs which are conserved in three different pseudo-spaces (utilizing three biological replicates of patient 2 as the spatial reference respectively) can be found in Supplementary Data 5.

## GeoMx DSP targeted spatial transcriptomics data analysis

The single-sample gene set enrichment analysis (ssGSEA) is performed to generate an enrichment score of myeloid cells for each ROI in the control and COVID-19 groups using the GSVA R package (v1.44.5). the gene set signatures are defined as the differentially expressed genes of myeloid cells identified by Seurat[43] (log2FC > 0.5, FDR < 0.05). The CIBERSORTx[73] web tool is used to estimate the abundance of myeloid cells for each ROI in the control and COVID-19 groups. All parameters follow default values.

## Statistics

Python (version 3.8.5) and R (version 4.1.0) are used for the statistical analysis.

## Reporting summary

Further information on research design is available in the Nature Portfolio Reporting Summary linked to this article.

## Data availability

The original data used in this paper can be accessed through the following links: (1) 10X Visium data of the "human dorsolateral prefrontal cortex (DLPFC) [http://spatial.libd.org/spatialLIBD/][41]; (2) STARmap data of the "mouse primary visual cortex V1 [https://zenodo.org/record/7830764#.ZDpObi-1HUI][42]; (3) "Spatial Transcriptomics" data of the "human HER2 breast cancer [https://zenodo.org/record/5511763#.Y6kMduxBzUI][45]; (4) single-cell RNA-seq data and "Spatial Transcriptomics" data of the human skin squamous cell carcinoma (SCC): GEO accession: "GSE144240[44]"; (5) single-cell RNA-seq data of the mouse intestine: GEO accession: "GSE109413[46]"; (6) 10X Visium data of the mouse intestine: GEO accession: "GSE169749[91]"; (7) single-cell RNA-seq data of the mouse liver: GEO accession: "GSE84498[47]"; (8) 10X Visium data of the "mouse liver [https://www.livercellatlas.org][92]; (9) single-cell RNA-seq data of the "mouse neocortex [https://portal.brain-map.org/atlases-and-data/rnaseq/mouse-v1-and-alm-smart-seq]"[48]; (10) single-cell RNA-seq data of the mouse kidney: GEO accession: "GSE129798[49]"; (11) 10X Visium of the mouse kidney [https://www.10xgenomics.com/resources/datasets]; (12) Slide-seq v2 data of the "mouse kidney [https://cellxgene.cziscience.com/collections/8e880741-bf9a-4c8e-9227-934204631d2a]"[93]; (13) single-cell ATAC-seq data of the mouse cortex: ArrayExpress: "E-MTAB-11264[81]"; (14) single-cell RNA-seq data and "Spatial Transcriptomics" data of the "human embryonic heart [https://data.mendeley.com/datasets/mbvhhf8m62/2]"[29]; (15) single-nucleus RNA-seq data of the "middle temporal gyrus (MTG) of the human cortex [https://portal.brain-map.org/atlases-and-data/rnaseq/human-mtg-smart-seq][30]; (16) single-nucleus RNA-seq data of the "multiple cortical areas (MTG, ACC, V1C, M1C, S1C and A1C) of the human cortex [https://portal.brain-map.org/atlases-and-data/rnaseq/human-multiple-cortical-areas-smart-seq]"; (17) single-cell RNA-seq data of the human melanoma: GEO accession: "GSE72056[31]"; (18) "Spatial Transcriptomics" data of the "human melanoma [https://www.spatialresearch.org/resources-published-datasets/doi-10-1158-0008-5472-can-18-0747/] or [https://zenodo.org/record/7830764#.ZDpObi-1HUI][54]; (19) single-cell RNA-seq data of the "human lung of lethal COVID-19 [https://singlecell.broadinstitute.org/single_cell/study/SCP1219][32]; (20) 10X Visium of the normal human lung: GEO accession: "GSE178361"[66]; (21) GeoMx DSP targeted ST data of the "human lung of COVID-19 [https://doi.org/10.5281/zenodo.4635285]"[72]. All other relevant data supporting the key findings of this study are available within the article and its Supplementary Information files or from the corresponding author upon reasonable request. Source data are provided with this paper.

## Code availability

The scSpace algorithm and related analysis are available at GitHub[94].

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

## Acknowledgements

This work is supported by the National Natural Science Foundation of China (81973701, 82204772), the Natural Science Foundation of Zhejiang Province (LZ20H290002), the Innovation Team and Talents Cultivation Program of National Administration of Traditional Chinese Medicine (ZYYCXTD-D-202002), the Fundamental Research Funds for the Central Universities (226-2022-00226), the China Postdoctoral Science Foundation (2022M712811), and Alibaba Cloud provided by Alibaba-Zhejiang University Joint Research Center of Future Digital Healthcare. The authors gratefully thank Jindong Wang for implementing the Python version of the TCA algorithm and thank Dr. Kim Thrane for providing the H&E-stained tissue images of melanoma.

## Author contributions

X.F., B.Z., and J.L. conceived the study. J.L. wrote the first draft of the manuscript, J.Q. revised the manuscript and added experiments. J.Q., J.L., Z.L., and Y.C. collected datasets involved in this article, benchmarked all methods, and participated in the development of the scSpace algorithm. Y.Z. analyzed the biological meaning of the results predicted by scSpace. X.S., Y.C., W.G., P.Y., Y.H., H.B., Q.C., and M.L. provided a lot of advice on algorithm implementation and biological applications. Y.F. and B.L. proofread and described the method. W.G., Y.H., and H.B. provided valuable advice on gene ontology analysis in this study. All authors read and approved the final manuscript.

## Competing interests

The authors declare no competing interests.
