## [Peer Review File · Nature Communications]

REVIEWER COMMENTS

Reviewer #1 (Remarks to the Author):

Liao et al. present a novel algorithm called scSpace that integrates single cell and spatial transcriptomics data to find spatially heterogeneous subclusters by reconstructing cells onto a pseudo-space. The algorithm appears to be working well and was evaluated on several different spatial transcriptomic and single cell datasets in different biological contexts (brain, liver, intestine, lung). The manuscript could be strengthened by addressing clarity in several sections and benchmarking in the context of other integrative or transfer learning algorithms recently published (Tangram, Cell2Location, scCoGaps). The authors focus on transcriptomics data but do not address how the algorithm performs with other single cell -omics data.

Major comments:

1. Does the algorithm work with other types of single cell -omics data such as snATAC-seq and sn DNA methylation sequencing? If so, the authors should provide examples of this. One data set that might be of interest is from Zhang et al., Nature, 2021 (<https://www.nature.com/articles/s41586-021-03223-w>) where they have profiled cortical projection neurons using sn DNA methylation seq.
2. Why do the authors downsample the single cell data? Can the algorithm be run with the entire dataset? As datasets are growing into the hundreds of thousands and millions of cells, it will be important to leverage this rich information. Relatedly, can the algorithm leverage more than one tissue section from the spatial transcriptomics data as tissue morphology can be variable among donors and technical replicates? Limitations in this regard should be addressed in the discussion.
3. The authors do not discuss details regarding the scalability, speed, or efficiency of employing this algorithm. This should be provided.
4. How does scSpace compare to other transfer learning algorithms such as scCoGaps (Stein-O'Brien et al., Cell Systems, 2019)? Can the authors justify why Louvain was used as a benchmark algorithm and whether other algorithms were used for benchmarking?
5. From the methods, it appears that the single cell data from the Allen Brain Institute spans multiple cortical regions. In the MTG study specifically (Hodge et al., Nature, 2019), cortical layers were manually dissected and single cells were assigned a layer based on this ground truth at manual dissection. How well does scSpace predict the layer of each cell based on this ground truth assignment in this Hodge dataset?
6. The authors provide code and vignettes on GitHub, but it is unclear if they have provided code to reproduce all figures in the manuscript. This should be provided and clearly documented in the repo.

Minor comments:

1. Extended Data 1: Panel B (5 cell types; 4 subtypes) has a floating bar from another plot (looks like from panel D?)
2. Authors have left out recent algorithms that integrate single cell and spatial transcriptomics data such as Cell2Location (Kleshchevnikov et al., Nat Biotechnol, 2022) and Tangram (Biancalani et al., Nat Methods, 2021) in the introduction.
3. Line 57 of introduction, "fail to achieve cellular resolution" is problematic as Slide-seq data is subcellular.

4. Line 134 "because cell coordinates" should be "cell or spot" coordinates as Visium is not capturing single cells.
5. There should be some discussion of caveats regarding which datasets are put into the algorithm. For example, cortical regions can be quite different and it may not be biologically accurate to use DLPFC spatial data with single cell data from other cortical regions (for example dACC, as the dACC is agranular and lacks cortical layer 4).
6. Figure 4J please provide layer annotations on ISH images for clarity
7. Visualizations are labeled as "pseudo-space" (ex. Fig. 3b) and "scSpace" (ex. Fig. 3e), it might be helpful in Fig 1 extended data to show how these visualization relate to the model. Some figures show only pseudo-space (Fig 2) and others show both.
8. T-cell exhaustion hypothesis (beginning line 297) should be cited
9. Beginning line 327- the spatial dataset used in this analysis is not clear as it is not mentioned in the results. It is mentioned in the methods, but should be clarified in the results.

Reviewer #2 (Remarks to the Author):

All of my comments can be found in the attached pdf-file called "comments.pdf".

Reviewer #3 (Remarks to the Author):

In this manuscript Liao et al. present scSpace, a method to retrieve spatial relationship between single-cells by integrating single-cell and spatial transcriptomics using transfer learning model. They apply their method initially on simulated data but also on real experimental datasets. They claim that scSpace can map the single-cell data into their pseudo-space which corresponds to their real spatial localization on the tissue. Also, they try to show using a spatially-informed clustering approach of scSpace improves the clustering accuracy. The whole idea is very interesting but I believe there is still some work needed to make the manuscript much more understandable and the significance of the scSpace clearer. My main concerns are:

If what is represented as pseudo-space is really capturing the spatial localization of the cells/spots or is it very similar to what a diffusion component would provide as well without integrating spatial information.

Are the claims that are made related to improved clustering of scSpace compared to other algorithms really fair, or this is just due to under-clustering using other benchmarked algorithms?

I will go through the revision points case by case as follows:

Line 85-85: Since the authors are describing the use of TCA to integrate single-cell and spatial data, it would be great if they could elaborate more on how they define the model? Which dataset is being used as the source and which one as target? (Is it from Single-cell to spatial or the other way around?)

In Figure 1a, the schematic that describes the algorithm, it would be great if it reflects better what is explained in paragraph one. For example, to show that in the "spaced informed expression graph", spatial information is embedded as edge weights.

The result of space-informed clustering is mapped back on the pseudo-space right? It would be better if the authors can show that on the schematics.

Also in supp.Fig 1a, I can't follow what is happening after the Transfer component analysis in the schematic. I expect the output of the TCA is n space_informed reduced dimensionality data points (so single-cell and spatial data merged), then why after this step we still have a separate green and blue box? What do they represent? In the same figure, the Multilayer perception is mentioned but I am missing this in the text, can the author explain why and where this has been used, what is the output and how they have used it? (I have read the methods section, but still feel this schematic is not representing well the algorithm)

Line 90-92 the authors mention that "After the transformation was determined, single cells were mapped to an assumed coordinate system (pseudo-space) characterized by the MNN model using existing spatial references", which means the "pseudo_space" is basically the MNN determined x-y coordinates right? This MNN step has not been described in none of the schematics (neither Fig1a nor Supp.Fig 1 a)

In general, I think the schematic that describes the algorithm should be revised extensively and be more followable.

Line 106-108: Comparing scSpace with k-means, Louvain and Hclust is very tricky. These algorithms can produce different clusters by different parameter settings. How the authors are confident that what they represent in the benchmark as the result of the for-example k-means clustering, is the best result of k-mean clustering? Maybe just increasing the K would result on a very similar clustering as scSpace! Authors should provide more evidence on this.

Line 137-139: The idea of the paper is to be able to map single-cell data to their spatial location and Improve clustering. While in this paragraph, the authors are applying scSpace on two spatial transcriptomics datasets as another validation to show the algorithm works well. I think the scenario of wanting to predict the location of another spatial transcriptomics dataset is not of significance as this data has already the spatial information integrated.

I also suggest to change the title of the second paragraph (line 131-132) mostly because of using "biological data". To something more similar to the title of Fig 2.

Line 140-145: Here I am very curious to see how the projection of the test spatial dataset on a diffusion map would look like? I doubt we would be able to capture the structure of the layers with diffusion maps (or a trajectory analysis) as well. Can authors show how the pseudo-space representation is much more informative than diffusion maps of the test slide of spatial transcriptomics both in terms of structure reconstruction or normalized pairwise distances?

Line 155-156: Here the single-cell is from the Alan atlas, but which spatial dataset has been used?

Line 158-159, and figure 2g: the result of this part which is the main idea of this paper-reconstructing the spatial location of single-cells- does not look very promising, the representation in pseudo-space looks not very well defined. And this is on a very structured organ like brain. I am wondering how scSpace and spatial reconstruction would look on a spatially unorganised tissue such as tumors metastasis tissue samples. Following the manuscript, I see the application of scSpace on Melanoma, but I would like to see how the original spatial transcriptomics dataset (I guess Visium) looks with these clusters shown on Melanoma.

Line 193-194, fig 3e: Here the author claims that scSpace clusters better the embryonic heart dataset by comparing it to Louvain clustering. I am doubtful about Louvain not being able to cluster those subpopulations as they are being clustered even on the tsne representation (which I guess is based on only single-cell gene expression data). So even without scSpace, on a reduced dimensionality space of

tSNE we can see these clusters and I believe Louvain not capturing them is merely a matter of clustering resolution. I don't think this example is a good example to point out scSpace's power in improving clustering.

Line 232-233: "scSpace first embedded single cells in a pseudo-space to reconstruct the spatial associations of cells". Using which spatial dataset as the spatial component of the scSpace?

Also, here the authors are mentioning applying scSpace on single-cell data from Alan brain atlas to reconstruct spatial localization of the cells. But isn't it exactly what they described in paragraph 2 line 155-159? If it is a different analysis, I guess it's very difficult to catch that difference.

Line 232: using which spatial dataset?

Line 264: Here, I assume the tsne is only based on expression profile, but it also shows a segregated cluster for each scSPACE clustered clusters. So maybe only the gene expression is enough to get those subpopulations?

Line 266, figure 4F, maybe not clustering them separately is due to the resolution of Louvain clustering and if you increase it you will get those clusters as we can see them separate already in tsne space

The same point goes for Fig4 d,e and f (line 240-248). Figure 4d represents tsne (which I guess comes from gene_expression only), but it is titled as scSpace. I would expect the pseudo-space to be marked as scSpace. This is a bit confusing. Would be great if authors explain this better.

Line 281-283: It would be much more informative if the authors show at least the tsne and cluster distribution of both original single-cell and spatial datasets in all analysis (spatially mapped transcript of the spatial data as well, for example visium images with clustered spots on top) in supplementary without applying scSpace and analyzed by a classic single-cell/spatial data analysis pipeline (Seurat as authors have used this). As a reader I am curious to see how these initial datasets look without applying scSpace and how much is the effect of scSpace.

Fig 5c: Can authors comment on what are the gray cells in this pseudo-space? A legend is missing for those cells!

Line 327-332: What is the tissue of the single-cell RNA-seq data that you have covid and healthy conditions for?

Which spatial dataset is used in the scSpace TCA model as the spatial reference?

Here again in line 335-336: "After reconstructing spatial relationships of cells, the pseudo-space of the Control and Covid-19 group was established (Fig. 6a)", how the spatial relationship has been built without a spatial reference? Are the authors using the scSPACE here merely as a batch correction method between healthy and disease single-cell data

Some minor points:

Line 38-39: Unnecessary comma

Line 391: Extra "the" after discussing.

Reviewer #4 (Remarks to the Author):

Reconstruction of the cell pseudo-space from single-cell RNA sequencing data with scSpace

Overview:

In this article by Liao et al, the researchers developed a computational technique, scSpace, to identify the spatially variability within scRNAseq data. This is a worthwhile endeavor as scRNAseq requires the dissociation of the tissue which results in the loss of this information. Likewise spatial transcriptomic platforms regularly make use of scRNAseq data to deconvolve the spots to understand cellular heterogeneity, but this information is not often applied in the opposite direction. The authors make use of simulated data as well as described sc and spatial datasets with distinct anatomical regions (brain, heart, intestine, and liver) to validate their methodologies. They further apply scSpace to different datasets of disease including melanoma and COVID-19 patient samples. scSpace is a promising approach. Some further validation and expansion on derived biological significances gained with scSpace will strengthen the article.

- In the simulated datasets, it appears the ARI of scSpace continues to decrease whereas Louvain and Kmeans remain relatively stable as subclusters increase. At what number of subclusters, if any, does scSpace either match Louvain and Kmeans ARI score or have worse ARI scores than these 2 techniques?
- In the IT neuron data analysis, figure 4j uses LAMP5 and GRIK1 which were not identified in the differentially expressed marker genes from Extended data 4a. Please explain the rationale for choosing these genes as the current rationale for their choice does not match (line 254-255).
- Figure 4j uses anatomically distinct regions of the brain. Although this information is found in the methods, it would be worthwhile informing the reader in the figure/text. Are both of these anatomic regions captured in the spatial dataset used as a reference?
- In the melanoma dataset analysis, the authors reference a gene TPX2 (line 307, ref 40) but this gene is not shown in the analysis like the others (Fig 4h). Similarly, the authors discovered MKI67 is highly expressed in C5, but isn't shown in Fig 4h. Is anything known about MKI67 in the literature? Is this potentially a novel therapeutic marker identified with scSpace? Recommend adding TPX2 and MKI67 in figure 4h.
- In the discussion, it is stated memory T cell features begin to be lost. Where is this data presented? It is unclear in the results section where the authors identify the loss of these features and the specific features/genes that are associated with memory T cells.
- As is the case with any reference-based approach, the reference used will heavily affect the results. Addressing this in scSpace would be worthwhile. One way this can be done is with the melanoma dataset.
 - o First, when using the melanoma dataset, it is unclear if all spatial sequencing datasets were used as a reference or just a single slide/slice (reference paper appears to have performed spatial sequencing on 4 patient samples). In previous cases it appears a single slice is used, so I assume this is the case here. If not, how was the use of multiple slides/spatial data addressed in scSpace?
 - o To validate that the spatially variable information gained from scSpace is conserved, I recommend authors make use of the multiple spatial datasets from Thrane et al (ref 36). The authors should use all 4 spatial slices from Thrane et al. and show in their article how closely related the spatially variable information is within the sc data when using each slice as a reference. This could be an optimal dataset to help show that regardless of the reference spatial dataset, spatial variability is generally conserved. These spatial datasets are also optimal to use as the spatial sequencing reference is done from the same lab and article, which could help mitigate some batch effects in the spatial data.
- For the COVID-19 dataset, several things should be addressed:
 - o First, the only pseudospace shown is for epithelial and myeloid cells. Please present a pseudospace that shows all cell types either in the figure or in the supplementary figure.
 - o Figure 6d, is there significance in the normalized distance between infected and control samples? Please add significance indicators.
 - o What non-mitochondrial genes were enriched in C4? Are there other non-mitochondrial genes that support the pathway in figure 5j? Also, C6 appears to have a relatively high enrichment of these mitochondrial genes too, proposed thoughts on this?
 - o Upon reading the methods, the only spatial reference used was an uninfected lung sample. However, I do not believe this is the best approach. In a viral infection, there will be major anatomical changes in response to the virus, such as inflammation and infiltrating cells. These responses will

heavily influence the spatial variation and heterogeneity. It does not seem totally appropriate to me to use a healthy lung reference to make inferences about infected samples and spatially altered variation. Authors should revisit this analysis to address these impacts of infection. It would be worthwhile using an infection reference instead or to use one to validate observed spatial variability. Example spatial references include the GeoMX references generated by Rendeiro et al. 2021 (doi: 10.1038/s41586-021-03475-6) and/or Margaroli et al. 2021 (doi: 10.1016/j.xcrm.2021.100242).

- The discussion is a bit underwhelming as a reader. There seems to be little connection with other scientific literature and biological significance/insight. Recommended spending more time expounding on the last paragraph. Suggest splitting melanoma and COVID-19 discussion into 2 different paragraphs and highlighting what biological insights were gained specifically using scSpace.

Minor edits:

- Some choppy wording/incomplete sentences are occasionally used and should be revised. For example, see lines 55-56 and 332-334
- Covid-19 should be COVID-19. This version is the preferred/recommended style for multiple public health agencies such as the World Health Organization (WHO).
- Figure 2i is difficult to read with the many black outlined cells which covers the expression level colors. Recommend revising.
- Line 299-300 add reference
- The Github repo has the code for using scSpace but lacks the code used for additional analyses on the datasets, such as gene set enrichment analyses. For reproducibility, it would be best practice for authors to provide all code used for analyses, not just the code to run scSpace on a given dataset.

Reviewer #5 (Remarks to the Author):

The authors propose scSpace, a new method to perform the spatial clustering of a tissue sample by embedding the high-resolution level of scRNA-seq experiments with the spatial information carried by spatial transcriptomics data.

The authors illustrate the advantages of using their method with a series of key studies.

*** Major comments***

-) Is it that common to have for a single tissue sample both the single-cell and the s.t. expression matrices? I presume that producing two kinds of experiments from the same tissue sample would be expensive and more complex than using just one protocol. Please, discuss this aspect more in detail, especially in the introduction. This is something that should be clear from the beginning, in such a way that people will know if this method could be used or not for their problem.

-) Even if Figure 2a shows that the pseudo space looks somehow similar to the original coordinate system, my concern is that the morphology of the data is not completely retained and so this might introduce some kind of bias in the analysis. I know that the original s.t. coordinates and the pseudo space coordinates cannot be easily compared because they represent in practice two different things, but I am wondering if there exists a way to evaluate if the neighbouring system from the s.t. data is somehow preserved in the new pseudo space.

Some recent methods for the analysis of spatial transcriptomics data, such as SpatialDE, SPARK (Sun et al. 2020) and SpaRTaCo (Sottosanti and Risso 2021), exploit the distances across the spots in the space and not just the neighbours. I would like the authors to discuss if their idea of using a pseudo space would work also in a framework that works with the distances, or if instead there would be some critical issues that might affect the results.

-) The authors should mention Zhao et al. (2021), who propose a space-informed clustering for spatial transcriptomics data called BayesSpace that artificially augments the data resolution. I believe that scSpace would be preferable over BayesSpace because it combines two different sources of

information, the scRNA-seq data and the s.t. data. However, having the same tissue be processed with two different protocols is probably not that common, as it requires a specific experimental design and it would be more expensive than using just one of the two methods. I think that the authors should discuss the pro and contra (both in terms of computational and monetary cost) of using scSpace, which embeds the information coming from two different types of experiments, against BayesSpace, which is applied only to the spatial transcriptomics data and artificially augments the data resolution.

In addition, once the latent biological feature representation extraction and the spatial reconstruction steps of scSpace are performed, it would be interesting to run the space-informed clustering phase but using BayesSpace instead of the method described in the paper (from line 488). Based on these results, it will be possible to comprehend if the results presented in this article are mainly the fruit of the pseudo space idea, or if instead all the three phases described in the section Methods have a fundamental role.

To wrap up, I would compare the results obtained from the following models:

1. scSpace
2. BayesSpace (on the original coordinate system, using only the Visium data)
3. scSpace + BayesSpace instead of what is shown from line 488

-) My opinion is that the authors should not present the results just by referring to the figures, but they should also explain and comment more in detail the content of the figures. This consideration applies to each and every section where some results are presented.

-) figure 2c: what does "distance to Layer 1" mean? Is there any sort of centroid that you use to compute the distance to Layer 1? The authors should explain better this part, as well as the purpose of the second graph in figure 2c.

-) In some cases, the authors had to consider two different samples from different subjects in order to find an appropriate set of spatial coordinates to be applied to the scRNA-seq dataset of interest. However, it seems to me that this operation inevitably carries the effects due to the biological differences between the two samples in the analysis. Please, discuss this aspect.

-) lines 147-150: please clarify this sentence and discuss it by referring to the figure, not just citing the figure.

The followings are some comments to the section "Identification of spatially associated subpopulations by scSpace" (lines 432-498):

-) This section aims to describe the statistical theory at the base of scSpace. In my opinion, the level of writing is quite poor and the notation is at times inconsistent or is left unexplained. I would encourage the authors to carefully revise this part, explaining with more precision the operations that are performed within the three steps of their method, also citing the proper literature. Understanding the math behind a method is crucial to determine its advantages and limitations, thus this section must be thorough.

Just to give some examples:

lines 441-442: marginal distribution of what? Say clearly what are the rows and the columns of your data matrix;

line 448: what are n_1 and n_2 ? what does the summation index i stand for?

line 452: what are the four elements in the K matrix?

lines 459-460: say clearly how to determine m and \tilde{W} ;

line 479: say clearly how to determine X'_S and X'_T (if I got it correctly, they are the output of the previous step. Thus, it must be clear how to compute them);

lines 484-486: I am confused about this sentence. Probably it should be "scSpace first trains the model on s.t. data using mean squared error,", but maybe I misunderstood what the authors wanted to say.

-) In Section "spatial reconstruction" (lines 474-487), I'm wondering what could be the advantage of running the multi-layer fully connected neural network over the entire set of coordinates from the s.t. dataset without testing its prediction ability. To me, the idea of creating a pseudo space sounds like a prediction problem which answers the question "What would be the coordinate system of a scRNA-seq dataset X'_S , given that there exists a certain relation between the coordinates of a s.t. dataset and its expression matrix X'_T captured by the neural network?"

If the neural network is trained on a given s.t. dataset and tested over a separate s.t. dataset, it would be possible to evaluate if such a model is reliable to create the pseudo space.

Please, discuss your choice.

-) In the Section "space-informed clustering", please describe more in detail the idea of using the space graph G_S as a weight function of the gene expression graph G_g (especially line 496).

MINOR COMMENTS

-) figure 1b: what are the x and the y axes?

-) lines 38-40: the sentence seems to be incomplete.

-) lines 57-60: the sentence seems to be incomplete.

-) To me, lines 377-395 belong more to the Introduction than to the Discussion.

Response to reviewers

Overview of Changes

We sincerely appreciate the reviewers' constructive comments and positive feedback. Our work has been much improved based on their valuable suggestions. We have tried our best to address the concerns raised by the reviewers point by point. Compared to the original version, the revised manuscript adds substantial validation results, adequately describes the method, and illustrates the performance of scSpace from multiple perspectives. The main changes are summarized as follows.

1. The manuscript is reorganized to include more validations with more complex tissue structures and more refined cell clusters. The Introduction and Discussion sections are comprehensively described and deeply discussed, and the core design concept and potential application of scSpace are fully explained.
2. The Method section has been carefully revised and proofread to make every formula explicit and correct. Besides, we have consulted several mathematical experts, provided an exhaustive formula derivation process, and adequately described the choice of each condition.
3. The performance of scSpace is benchmarked more systematically. Additional simulated and existing datasets are included to evaluate the robustness and scalability of scSpace. More space-aware and clustering algorithms are introduced to compare with scSpace. More parameters of other methods are adjusted to ensure the transparency of the comparison.
4. The validation of scSpace is more diverse and complete. Specifically, other single-cell omics data are used to verify the performance of scSpace, the pairwise distance of cells predicted by scSpace is employed as a weight for further clustering, the reconstructed pseudo-space is validated by spatial cell-to-cell communications, and the results are confirmed by spatial deconvolution analysis.
5. To ensure the availability and transparency of scSpace, the algorithm is well documented, step-by-step tutorials are provided for the reproducibility of the results, and all relevant source code has been deposited to GitHub.

We hope this edition will satisfy the reviewers and address all the concerns to win their approval for the publication of our manuscript. Please find detailed responses below.

Reviewer #1 (Remarks to the Author):

Reviewer's summary

Liao et al. present a novel algorithm called scSpace that integrates single cell and spatial transcriptomics data to find spatially heterogeneous subclusters by reconstructing cells onto a pseudo-space. The algorithm appears to be working well and was evaluated on several different spatial transcriptomic and single cell datasets in different biological contexts (brain, liver, intestine, lung). The manuscript could be strengthened by addressing clarity in several sections and benchmarking in the context of other integrative or transfer learning algorithms recently published (Tangram, Cell2Location, scCoGaps). The authors focus on transcriptomics data but do not address how the algorithm performs with other single cell-omics data.

Response: Thank you very much for your positive feedback on the idea of the scSpace algorithm, the concept of the cell pseudo-space, and the evaluation results on transcriptomics datasets. We strongly agree with your valuable suggestions on the clarity in several sections, comparison of different methods, and application with other single-cell-omics data. According to your constructive comments, we have revised the manuscript comprehensively to address these issues, and we believe that these valuable suggestions will greatly improve the performance and application scope of scSpace.

Specifically, we have reorganized the manuscript by including the Hodge dataset provided by you in the main text as a featured section. There is no better place to validate the performance of scSpace than the Hodge dataset, which includes a highly diverse set of neuron types and subtypes, detailed annotation of cortex layers, and patterned laminar distributions of cells. This set of data can verify the reliability of scSpace from multiple perspectives, and we have fully mined its information. Also, we have compared the performance of scSpace with a transfer learning algorithm scCoGaps, and four published spatial clustering algorithms STAGATE, BayesSpace, DR-SC, and SpaGCN in the benchmarking. Besides, we have applied scSpace to other single-cell-omics data such as scATAC-seq, and successfully reconstructed the pseudo-space of the mouse cerebral cortex.

All significant modifications are marked in blue in the revised manuscript. We hope this edition will address your concerns.

Major comments

1. Does the algorithm work with other types of single cell -omics data such as snATAC-seq and sn DNA methylation sequencing? If so, the authors should provide examples of this. One data set that might be of interest is from Zhang et al., *Nature*, 2021 (<https://www.nature.com/articles/s41586-021-03223-w>) where they have profiled cortical projection neurons using sn DNA methylation seq.

Response: Thanks for your constructive comments, which greatly inspired us in the extension of the applicable scope for the scSpace algorithm. Indeed, scSpace was only designed to deal with single-cell transcriptomics data because of two reasons. First, most of the current single-cell and spatial omics data sets are based on transcriptomics. Second, the development of a method requires rigorous benchmarking and extensive training, yet the paired spatial reference is lacking in other omics such as DNA methylation sequencing. On the other hand, we also did make efforts to ask the authors of the Spatial-ATAC-seq for the paired spatially resolved chromatin accessibility and transcriptomics data, but unfortunately, we have not received any reply. Meanwhile, spatially resolved ATAC-seq or methylation-seq data is in its infancy, unlike single-cell ATAC-seq or methylation-seq, which has considerably explored. Currently, there is no suitable spatial reference for other single-cell-omics data, so it is difficult to conduct the integrative analysis.

Nevertheless, we fully agree with you that the performance of scSpace on the transcriptomics data should theoretically apply to other single-cell-omics data. To this end, we turn to a mouse cortex scATAC-seq dataset generated by ISSAAC-seq (Xu et al., *Nature Methods*, 2021, <https://www.nature.com/articles/s41592-022-01601-4>), a novel technology that enables multimodal profiling of chromatin accessibility and gene expression in single-cell simultaneously, and followed a widely used data analysis pipeline applied in the multi-omics integration task to converted the scATAC-seq data into a gene activity matrix using Signac (Stuart et al., *Nature Methods*, 2021, <https://www.nature.com/articles/s41592-021-01282-5>) in advance. scSpace could then reconstruct the pseudo-space of cells in scATAC-seq data still with the spatial transcriptomics reference (STARmap).

As illustrated in Figure 1a, after spatial reconstruction of the pseudo-space by scSpace, cells from different cortex layers exhibited heterogeneous distribution patterns. Besides, the spatial associations in the pseudo-space from layer 2/3 to layer

6 are consistent with the exact structure of the cortex layers. The results are also validated by the spatial distribution of cell subpopulations from different layers of the mouse cortex (Figure 1b). The analysis of the ISSAAC-seq data demonstrates that scSpace has the potential for spatial reconstruction of the pseudo-space with other single-cell-omics data. We have included these results in the Extended Data.

Figure 1. The pseudo-space of the mouse cortex is reconstructed by scSpace using single-cell chromatin accessibility data derived from ISSAAC-seq. a, Spatial distribution of single cells from different layers of the mouse cortex in the reconstructed pseudo-space. **b,** Spatial distribution of cell subpopulations of the mouse cortex in the reconstructed pseudo-space.

As mentioned above, there are currently not enough paired datasets (non-transcriptomics) for scSpace to benchmark, so there is still a distance to fully implement a cross-omics analysis. The example of scATAC-seq data indicates that scSpace has the potential to work with other types of single-cell-omics data. However, the main application scenario of scSpace is still focused on the transcriptomics data.

2. Why do the authors downsample the single cell data? Can the algorithm be run with the entire dataset? As datasets are growing into the hundreds of thousands and millions of cells, it will be important to leverage this rich information. Relatedly, can the algorithm leverage more than one tissue section from the spatial transcriptomics data as tissue morphology can be variable among donors and technical replicates? Limitations in this regard should be addressed in the discussion.

Response: Thanks for your professional comments. We have fully considered your concerns and made great efforts regarding the down-sample procedure and the validation with multiple tissue sections.

The question can be separated into two parts, and I will respond one by one.

Question 1: Can scSpace run with entire datasets instead of down-sampling single-cell data?

Typically, down-sampling is a common strategy in the integrated analysis of single-cell RNA and spatial transcriptomics data such as CellTrek (*Nat Biotechnol*, 2022), Bulk2Space (*Nat Commun*, 2022), etc. for saving computational resources without affecting the accuracy of results. Similarly, scSpace also down-samples the single-cell data to speed up the analysis process, but this is not an indispensable step. As you suggested, we reanalyzed the scRNA-seq dataset of the mouse V1 neocortex derived from Allen Brain Atlas by running the entire dataset with scSpace and compared the performance with entire dataset with previous analysis using down-sample data.

Figure 2. Comparison of the pseudo-space constructed by scSpace between down-sampled and entire dataset. **a**, Reconstructed pseudo-space of 15,413 cells from different cortical layers of the mouse V1 neocortex using the entire dataset. **b**, Spatial distribution density of cells from different layers using the entire dataset. **c**, Reconstructed pseudo-space of 5,000 cells from different cortical layers of the mouse V1 neocortex using down-sampled data. **d**, Spatial distribution density of cells from different layers using down-sampled data.

As shown in Figure 2, scSpace successfully reconstructed the pseudo-space of the mouse V1 neocortex (Figure 2a) using the entire dataset containing 15,413 single cells. The distribution density of the cells in different cortical layers shifted layer by layer in

space (Figure 2b). This result reproduced the spatial relationships between cells predicted by scSpace using the down-sampled data previously (Figure 2c, and 2d) and demonstrated that scSpace can run the integrative analysis stably on large datasets with or without down-sample procedure.

Question 2: can the algorithm leverage more than one tissue section?

In addition, we couldn't agree more with you about how scSpace can leverage the rich information from different tissue sections with variable tissue morphology and distinct spatial transcriptomics in different donors or technical replicates. In the current version, scSpace doesn't include the function to directly leverage more than one tissue section as the spatial reference, but there are two optional strategies as trade-off. First, we can run the spatial reconstruction for different replicates, respectively, and then average the output pseudo-spaces. Second, a spatial alignment procedure such as PASTE (Zeira et al., *Nature Methods*, 2022, <https://www.nature.com/articles/s41592-022-01459-6>), can be used in advance to integrate spatial transcriptomics data as one reference. However, it needs to be discussed separately in practice, because different replicates can share a large spatial coherence or a small one. Therefore, three examples including 12 slices from the human dorsolateral prefrontal cortex (DLPFC), 12 slices from the human skin squamous cell carcinoma (SCC), and 36 slices from the human breast cancer (BC), are introduced to comprehensively discuss the performance of scSpace in processing heterogeneous tissue sections.

As shown in Figure 3, when scSpace is used in tissue sections with inherent spatial patterns, such as cortical layers of brains whose structure is conservative even between multiple individuals, the resulting Pearson's correlation coefficient (PCC) scores remain at a high level robustly across many consecutive or discrete slices. For instance, the PCC scores range from 0.448 to 0.696 in the analysis of the DLPFC dataset (Figure 3a). Moreover, when more complex conditions are encountered, such as cancers, the molecular signature of the tissues is highly heterogeneous among different patients and even within tumors. Specifically, for the SCC (Figure 3b) and BC (Figure 3c) datasets, PCC scores obtained by scSpace analysis on the same tissue from several donors are under 0.4 (Patient 5 and Patient 9 in the SCC dataset, and Patient B and Patient F in the BC dataset), indicating a striking inconsistency between tissue sections, because the biological replicates from these patients share low spatial coherences. Nevertheless, in most cases, the PCC scores were consistent across tissue

replicates from the same donors, and some results even demonstrate that PCC scores were reproducible in distinct donors by scSpace analysis. For instance, the high PCC scores for three replicates of Patient 2 in SCC dataset are also supported by the PASTE analysis, where Patient 2 is separated from Patients 5, 9, and 10 based on a higher spatial coherence score.

Figure 3. Performance evaluation of scSpace using multiple tissue sections from different organs, donors, and replicates. a, Pearson’s correlation coefficient (PCC) of pairwise distance between different tissue sections (Left) and different donors (Right) from the brain cortex in the DLPFC dataset. **b,** PCC of pairwise distance between different tissue sections (Left) and different patients

(Right) in the SCC dataset. \mathbf{c} , PCC of pairwise distance between different tissue sections (Left) and different patients (Right) in the BC dataset.

The ground truth of the spatial heterogeneity of transcriptomics, the reconstructed pseudo-space by scSpace, and the pairwise distance correlation between cells in the DLPFC, SCC, and BC datasets are illustrated as follows.

Figure 4. The original distribution of transcriptomics, the reconstructed pseudo-space by scSpace, and the pairwise distance correlation between cells among different donors and tissue replicates in the DLPFC dataset.

As shown in Figure 4, the molecular structure of the human brain cortex is highly organized, therefore, the intrinsic gene expression distribution also exhibits a layered spatial pattern, which results in the robust performance of scSpace across different donors and tissue replicates.

Figure 5. The original distribution of transcriptomics, the reconstructed pseudo-space by scSpace, and the pairwise distance correlation between cells among different donors and tissue replicates in the SCC dataset.

In the SCC dataset (Figure 5), the composition and spatial distribution of spot types were highly heterogeneous across patients and even replicates from same patients because the tumor involves a very complex pathological process.

Figure 6. The original distribution of transcriptomics, the reconstructed pseudo-space by scSpace, and the pairwise distance correlation between cells among different donors and tissue replicates in the SCC dataset.

Similar results are also derived from the BC dataset (Figure 6). For example, tissue sections from Patient E show obvious spatial architecture in transcriptomics. However, little spatial coherence is shared in three replicates of Patient F. We fully agree with you that the limitation regarding variable morphology of the tissue microenvironment,

the heterogeneity across multiple donors, and the intrinsic intra-disease difference in distinct tissue sections should be discussed specifically. From this perspective, our findings are in line with the published work, such as PASTE (*Nat Methods*, 2022), which attributed the difficulty in the alignment and integration of spatial transcriptomics data from Patient 5 and Patient 9 to the intrinsic differences in the spatial homogeneity of tumors. We have included the discussion in the revised manuscript. Taken all, scSpace can achieve robust performance in multiple tissue sections in most scenarios where spatial heterogeneity exists stably within tissues.

3. The authors do not discuss details regarding the scalability, speed, or efficiency of employing this algorithm. This should be provided.

Response: Thanks for your constructive comments and we fully agree with you that an evaluation of the scalability, speed, and efficiency of scSpace should be included. Therefore, we have experimented by increasing the cell number from 500 to 50,000 to evaluate the efficiency of operation and the accuracy of clustering.

Figure 7. Performance evaluation of scSpace based on the scalability, speed, and accuracy of scSpace. a, Running time of scSpace as the cell number increases. **b,** ARI and NMI values of the scSpace prediction as the cell number increases.

As illustrated in Figure 7, with a 24 GB NVIDIA GeForce RTX 3090 GPU, scSpace could process 50,000 single cells within 20 minutes, and the performance achieved stability with the number of cells increasing. The results have been included in the Extended Data.

4. How does scSpace compare to other transfer learning algorithms such as scCoGaps (Stein-O’brien et al., *Cell Systems*, 2019)? Can the authors justify why Louvain was used as a benchmark algorithm and whether other algorithms were used for benchmarking?

Response: Thanks for your valuable comments. In the original manuscript, the main task for scSpace is to identify spatially variable cell subpopulations from scRNA-seq data, we thus selected some common single-cell clustering methods, such as Louvain, K-means, and Hierarchical Cluster (Hclust), as the benchmark algorithms.

Nevertheless, we fully agree to compare the performance of scSpace with other algorithms. As you suggested, scCoGaps (*Cell Systems*, 2019) have been included in the updated benchmarking. Besides, four spatial domain identification methods, STAGATE (*Nat Commun*, 2022), BayesSpace (*Nat Biotechnol*, 2021), DR-SC (*Nucleic Acids Res*, 2022), and SpaGCN (*Nat Methods*, 2021) are also included. Since scCoGaps is a latent space learning algorithm to find cell-type signatures from single-cell RNA-seq data using non-negative matrix factorization (NMF), we thus replaced the latent biological feature representation extracted by TCA to the output of scCoGAPS, and the remaining steps were same as scSpace. All other parameters were followed the default settings.

Figure 8. Comparison of the performance between scSpace and other algorithms. y axis, Adjusted Rank Index (ARI) of each method using simulated data.

As shown in Figure 8, scSpace outperforms other algorithms with a significantly higher ARI score using 140 simulated datasets with subcluster numbers ranging from

2 to 10. Because scSpace targets scRNA-seq rather than spatial transcriptomics data, and considers not only transcriptional features, but also pairwise distances between cells via reconstructing the latent pseudo-space during clustering, its performance is better than other methods that only focus on transcriptional features such as Louvain, Kmeans, scCoGAPS, and Hclust. Meanwhile, other four spatial domain identification methods (BayesSpace, STAGATE, DR-SC, and SpaGCN) perform poorly in the space-informed single-cell analysis since they aim to identifying spatial niches or domains in spatial transcriptomics data rather than spatially heterogeneous cell subpopulations in scRNA-seq data. The benchmark result and corresponding discussion are included in the Extended Data.

5. From the methods, it appears that the single cell data from the Allen Brain Institute spans multiple cortical regions. In the MTG study specifically (Hodge et al., *Nature*, 2019), cortical layers were manually dissected and single cells were assigned a layer based on this ground truth at manual dissection. How well does scSpace predict the layer of each cell based on this ground truth assignment in this Hodge dataset?

Response: Thanks for your constructive suggestion and we fully agree with you that the single-cell data from Allen Brain Institute is an excellent example to illustrate the performance of scSpace since it spans multiple cortical regions and provides ground truth of refined cell-type annotations at manual dissection. We, therefore, employ the Hodge dataset to evaluate the performance of scSpace to predict the corresponding layer for each cell.

As shown in Figure 9, cells in different cortical layers also exhibit spatial heterogeneity in the pseudo-space reconstructed by scSpace (Figure 9a). The averaged pairwise cellular distance between each layer and the first layer presents a monotonically increasing trend which is identical to the ground truth of layer distribution (Figure 9b). Furthermore, we have compared the clustering performance of scSpace with Seurat on four excitatory neuron clusters and four inhibitory neuron clusters. In all cases, scSpace has higher ARI scores than Seurat in clustering (Figure 9c). Two examples, eight RORB-expressing subclusters and seven FEZF2-expressing subclusters, are exhibited to illustrate whether refined cell subpopulations can be identified by scSpace and Seurat. The ground truths of cell-type annotations for RORB-expressing neurons and FEZF2-expressing neurons are shown in Figure 9d and Figure

9g, respectively. The results demonstrate that scSpace (Figure 9e and Figure 9h) outperforms Seurat (Figure 9f and Figure 9i) in cell subtype clustering with significantly higher ARI scores. In conclusion, scSpace can well predict the cortical layer of each cell based on the ground truth assignment in the Hodge dataset.

Figure 9. Reconstruction of the pseudo-space of cortical regions using the Hodge dataset by scSpace. **a**, Pseudo-space reconstructed by scSpace. **b**, Pairwise distances between cells from different cortical layers to Layer 1. **c**, Comparison of the ARIs for excitatory/inhibitory neuron subtype clustering between scSpace and Seurat. **d**, The ground truth of the 8 RORB-expressing excitatory neuron subtypes. **e**, Cluster by scSpace. **f**, Cluster by Seurat. **g**, The ground truth of the 7 FEZF2-expressing excitatory neuron subtypes. **h**, Cluster by scSpace. **i**, Cluster by Seurat.

Notably, based on your valuable advice, we decide to present this comprehensive analysis (See response to Question 10 of the Method for Reviewer 2 on Page 27 for detailed information) as an essential part of the main content of our revised manuscript.

6. The authors provide code and vignettes on GitHub, but it is unclear if they have provided code to reproduce all figures in the manuscript. This should be provided and clearly documented in the repo.

Response: Thanks for your treasurable suggestion. We take this matter very seriously and have made all relevant source code available in an updated version of scSpace to ensure reproducibility of the results in the manuscript. In addition, all source code is well documented and detailed tutorials for performing the scSpace algorithm are provided. (<https://github.com/ZJUFanLab/scSpace/tree/master/AnalysisPaper>)

Minor comments

1. Extended Data 1: Panel B (5 cell types; 4 subtypes) has a floating bar from another plot (looks like from panel D?)

Response: Thanks for your correction, we are so sorry for the mistake. We have thus thoroughly checked all figures and texts in the revised manuscript.

2. Authors have left out recent algorithms that integrate single cell and spatial transcriptomics data such as Cell2Location (Kleshchevnikov et al., Nat Biotechnol, 2022) and Tangram (Biancalani et al., Nat Methods, 2021) in the introduction.

Response: Thanks for your insightful comments, we fully agree with you that these references regarding the integration of single-cell and spatial transcriptomics should be cited and discussed in the manuscript based on their popularity and applications.

3. Line 57 of introduction, “fail to achieve cellular resolution” is problematic as Slide-seq data is subcellular.

Response: Thanks for your professional comments. We have modified the statement to “failed to sequence individual single cells”. We agree with you that several spatial transcriptomics approaches such as Slide-seq v2 and Stereo-seq have achieved higher resolution than a regular size of a cell. However, it is possible for transcripts captured on the same bead to come from different cells.

4. Line 134 “because cell coordinates” should be “cell or spot” coordinates as Visium is not capturing single cells.

Response: Thanks for your kind reminding. We have revised the sentence to “because cell or spot coordinates” based on your advice.

5. There should be some discussion of caveats regarding which datasets are put into the algorithm. For example, cortical regions can be quite different and it may not be biologically accurate to use DLPFC spatial data with single cell data from other cortical regions (for example dACC, as the dACC is agranular and lacks cortical layer 4).

Response: Thanks for your professional comments and we fully agree with you that there may be bias in prediction if single-cell and spatial transcriptomics data are not from the same spatial area. Therefore, users are encouraged to use paired single-cell and spatial transcriptomics data with shared transcriptional features and conditions or perform some necessary pre-processing steps of the spatial reference when performing the scSpace algorithm. The discussion is included in the revised manuscript, as reflected below.

“Secondly, as the spatial architectures of different regions from the same tissue may also be heterogeneous (for example, while DLPFC and dACC both belong to the cortex, their spatial architectures are quite different, as dACC is agranular and lacks cortical layer 4), performing appropriateness assessment or necessary pre-processing steps of the spatial reference should be considered before applying scSpace.”

6. Figure 4J please provide layer annotations on ISH images for clarity

Response: Thanks for your valuable suggestion. In Figure 4J in the original manuscript, the ISH images were downloaded from Allen Brain Atlas and we didn’t make any changes to ensure authenticity of the original data. We fully agree with you that the layer annotations should be included for better understanding of the result. Therefore, in the revised manuscript, we have annotated the layers in the ISH images. In addition, many new results are added to the manuscript and thus the entire Figure 4 is included in the Extended Data.

7. Visualizations are labeled as “pseudo-space” (ex. Fig. 3b) and “scSpace” (ex. Fig. 3e), it might be helpful in Fig 1 extended data to show how these visualization relate to the model. Some figures show only pseudo-space (Fig 2) and others show both.

Response: Thanks for your constructive comments. The core function of scSpace is to reconstruct the pairwise distance between cells from the scRNA-seq data, which is defined as the ‘pseudo-space’ of single cells. Notably, spatial patterns of cell types can be illustrated and averaged pairwise distances between cell types can be calculated from the ‘pseudo-space’. Based on the ‘pseudo-space’, we can extract spatial information of cells and conduct further clustering by combining their transcriptional information. The ‘scSpace’ labeled in Fig. 3e represents the clustering results in tSNE layout. We apologize for the confusion or misunderstanding caused by our unclear statement. In the revised manuscript, we have stated the layout as ‘clustering results of scSpace’. Indeed, we didn’t show both figures in every result because the tasks are quite different. For instance, in Fig. 2, the spatial transcriptomics data from human and mouse cortex were used to validate the spatial distribution of cells or spots in the ‘pseudo-space’ predicted by scSpace. While in Fig. 3, we want to identify spatially heterogeneous cell subtypes from the single-cell data, so we need to illustrate the clustering results.

8. T-cell exhaustion hypothesis (beginning line 297) should be cited

Response: Thanks for your comments, we have cited relevant publications supporting the T-cell exhaustion hypothesis in the revised manuscript.

9. Beginning line 327– the spatial dataset used in this analysis is not clear as it is not mentioned in the results. It is mentioned in the methods, but should be clarified in the results.

Response: Thanks for your constructive comments and we fully agree with you. In the revised manuscript, we have clarified all datasets used in this research. Specifically, the human lung spatial transcriptomics data (10X Visium) are used as the spatial reference in the COVID-19 analysis and the dataset is downloaded from GEO (GSE178361).

Reviewer #2 (Remarks to the Author):

Reviewer's summary

The authors present a new computational method, scSpace, which uses a combination of matched (but not necessarily paired) single cell and spatial transcriptomics data to reconstruct the spatial structure of the single cells in "pseudospace". To achieve this, the authors first use TCA (transfer component analysis) to correct for batch effects between the different modalities, thereafter they use a multi-layered neural network (MNN) to learn a mapping between latent expression profiles to spatial position. This MNN is trained using the spatial data and then applied to the single cell data (for the purpose of reconstruction). In a last step, the authors create a spatially weighted gene expression graph (by combining an unweighted expression graph and a graph based on positions in pseudo-space), which is used to cluster the data using, for example, Louvain clustering.

The authors evaluate their method using both synthetic data and multiple real data sets. The method is not the first to address the task of spatial reconstruction, but does present a new approach, which could be of interest to larger community. However, I'm not sure about what this method will add to the current space of methods – although different, its results seem to be obtainable with other already existing methods. I'm also concerned about the level of detail, or rather the lack thereof, in the Method section describing the actual method. In addition, I would like to see some more extensive validation, not just quantity-wise but also in terms of difficulty. Finally, although the method is benchmarked, the competing methods are not well-chosen or representative of SOTA methods for spatially-aware clustering. If these points were to be addressed, the manuscript might be fit for publication in the journal.

I've separated my comments by section in the within and numbered. My intention has been to provide constructive comments, hoping that no matter what the editorial decision is, the authors will find these useful.

Response: Thank you very much for your positive comments and professional insight. As a senior expert in mathematics, your comments are insightful, clear, and logical. and we fully agree with you that these comments are constructive, innovative, and useful, both in terms of methodological rigor and validation integrity for the performance evaluation of scSpace.

As you pointed out, scSpace is not the first algorithm to address the task of spatial reconstruction, but it is quite different from other spatial reconstruction methods, that is, using spatial transcriptomics data to enhance the correlation between single cells, to construct the pseudo-space of single-cell data instead of mapping cells to spatial coordinates as other methods do such as Cell2location (Kleshchevnikov et al., *Nat Biotechnol*, 2022), CellTrek (Wei et al., *Nat Biotechnol*, 2022), RCTD (Cable et al., *Nat Biotechnol*, 2022), and SPOTlight (Elosua-Bayes et al., *Nucleic Acids Res*, 2021). The fundamental difference between scSpace and other methods is that scSpace focuses on single-cell data, while other methods focus on spatial data analysis.

In addition, through the reconstruction of pseudo-space, the further clustering of single cells by scSpace not only takes into account the differences of transcriptomes but also considers the distance information between cells, to obtain spatially heterogeneous cell subpopulations that are difficult to identified by traditional clustering methods such as Louvain, K-means, and hierarchical clustering, which only use the transcriptional information of cells. In summary, we hope to use spatial reference to enhance the information of the single-cell data, and to focus on the single-cell data analysis. We also benchmarked current methods on the synthetic data, and scSpace showed the highest performance across 140 datasets compared with other space-informed clustering methods.

Meanwhile, we value your concerns very much. In the revised manuscript, we have first carefully checked the details of the method and thoroughly described it for both experts and average readers, and then designed additional validations using much more datasets from diverse biological circumstances with distinct quantities and difficulties. Moreover, we have included five more existing clustering algorithms, STAGATE (Dong et al., *Nat Commun*, 2022), BayesSpace (Zhao et al., *Nat Biotechnol*, 2021), DR-SC (Liu et al., *Nucleic Acids Res*, 2022), scCoGAPS (Stein-O'Brien et al., *Cell Syst*, 2019), and SpaGCN (Hu et al., *Nat Methods*, 2021), in the benchmarking for spatially informed clustering as you suggested.

We have collated your constructive comments, carefully analyzed each issue and fully responded point-by-point with descriptive figures and discussion. All significant changes have been marked in blue in the revised manuscript. We hope the revised manuscript will address all your concerns.

Introduction

1. To do the field justice, I'd encourage the authors to cite some additional spatial deconvolution methods:

- cell2location: <https://www.nature.com/articles/s41587-021-01139-4>
- stereoscope: <https://www.nature.com/articles/s42003-020-01247-y>
- Tangram: <https://www.nature.com/articles/s41592-021-01264-7>

Response: Thanks for your valuable advice, we totally agree to cite these relevant references in the revised manuscript.

2. As this is a methods paper, I feel as if the authors could have devoted some additional space to explain the fundamental aspects of TCA (transfer component analysis) to the reader, this would make the text easier to follow.

Response: Thanks for your constructive comments, we agree with you that the method should be more clearly described and easily interpreted for both experts and average readers. Therefore, a thorough revision has been made to explain more about the fundamental aspects of TCA based on your professional suggestion. The descriptions are listed below and marked in blue in the revised manuscript.

“Using a transfer learning model, termed Transfer Component Analysis (TCA), which was first used in domain adaptation to solve a learning problem in a target domain by utilizing the training data in a different but related source domain, scSpace could eliminate the batch effect between scRNA-seq and spatially resolved transcriptomics data and extract the shared characteristic across these two types of data, enabling reconstruction of spatial architectures of single-cell data with high accuracy and precision.”

In addition, we have described the detailed analytical process of TCA as an important part in the Methods section of the manuscript, namely **“Latent biological feature representation extraction”**.

Method

To me, as a method developer, I value a clear and descriptive method section that outlines the core concepts used in the manuscript. Assuming that the reader has the

correct background, he/she/they should, by studying the Method section, be able to understand the mechanisms of the method presented. Unfortunately, I feel that this is not the case in this submission - there's a clear lack of detail, some incorrect formulas (probably typos), and poor explanation of certain elements. I don't want to sound too harsh, but at the same time, I feel as if it is my job to highlight the part of this manuscript that don't quite fit the standard that's requested from a journal like Nature Communications. I both have some general comments, as well as some more specific comments pertaining to the equations.

Response: Thanks for your constructive comments and we are sorry for the lack of detail and explanation in the Method section. In the revised manuscript, we have devoted much more efforts on this important part for all readers. We are very glad to receive your professional comments, otherwise these unclear description and careless mistakes will bring great obstacles to readers' understanding of scSpace. We fully agree with you that all descriptions and formulas should be accurate and canonical. Therefore, we consulted several experts in mathematics and reorganized all the mathematical derivation and method description. All related sections are re-written and highlighted in blue in the updated manuscript. We look forward to having your second review.

1. The authors are **strongly** encouraged to be more descriptive in their method section. Explaining why certain steps are taken or, for example, why a given condition is used (examples will follow).

Response: Thanks for your valuable suggestion. We totally agree with you that the accurate explanation and adequate description of the method are the key points of the article. In the revised manuscript, we have devoted great effort into explaining the involved steps of scSpace and the selection of parameters to make the method clear, informative, logical, and correct. All significant changes are highlighted in blue in the revised manuscript (Line 708-835).

2. I would **strongly** recommend the authors to provide a larger derivation, reference to a proof, or calculations for some of the steps in the section about MMD and TCA. I

taught linear algebra for five years, and even to me some of the steps aren't that obvious, something I would assume is true for the average reader then as well.

Response: Thanks for your professional comments. We are truly sorry for the poor organization of the initialed manuscript and difficulty in interpreting the method. In the revised manuscript, we have extensively supplemented the derivation, proof, and calculation of the MMD and TCA methods with the help of several mathematical experts. All significant changes are highlighted in blue in the revised manuscript (Line 708-784).

3. In the equation on line 448 there's no index over which the summation is done, my guess is that the equation should read:

$$MMD(X_S, X_T) = \left\| \frac{1}{n_1} \sum_{i=1}^{n_1} \phi(X_{S_i}) - \sum_{i=1}^{n_2} \phi(X_{T_i}) \right\|_{\mathcal{H}}^2 \quad (1)$$

Response: Thanks for pointing it out. We apologize for our carelessness and unprofessionalism. We have therefore revised the formula according to your advice.

4. I believe the equation on line 463 is somewhat incorrect. I'm fairly sure it should be:

$$MMD(\tilde{X}_S, \tilde{X}_T) = \text{tr}((KWW^T K)L) = \text{Tr}(W^T K L K W) \quad (2)$$

not

$$MMD(X_S, X_T) = \text{tr}((KWW^T K)L) = \text{Tr}(W^T K L K W) \quad (3)$$

That is, it is the projected data ($\tilde{X} = XW$) that $\text{tr}((KWW^T K)L)$ gives the MMD of, not the original data (X).

Also, it might be worthwhile to mention that the equality is due to the cyclic invariance of the trace operator.

Response: Thank you very much for your correction. We did make a mistake in writing this formula. Your writing is correct and your explanation is convincing. Based on your professional comments, we have thoroughly checked all formulas and relevant descriptions in the revised manuscript to avoid additional errors with the help of several experts in mathematics.

5. The condition $W^T K H K W = I_m$ is introduced quite arbitrarily, i.e., there's no explanation as to why this is a condition that we need in the optimization problem. To me, stating that this orthogonality is imposed to preserve the variance in the data (like in PCA), or something along those lines would be beneficial to the reader's understanding.

Response: Thanks for your insightful comments. We fully agree with you that there needs to be an explanation for the introduction of the condition. As described in the manuscript, the Maximum Mean Discrepancy (MMD) distance between the distributions of the single-cell and spatial transcriptomics data are formulated as:

$$\text{Dist}(X'_S, X'_T) = \text{tr}((K W W^T K) L) = \text{tr}(W^T K L K W).$$

In order to control the complexity of W , we add a regularization term $\text{tr}(W^T W)$ to the objection function to avoid the rank deficiency of the denominator in the generalized eigendecomposition. The final kernel learning problem reduces to:

$$\min_W \text{tr}(W^T K L K W) + \mu \text{tr}(W^T W)$$

$$\text{s. t. } W^T K H K W = I$$

where μ is a tradeoff parameter, $I \in \mathbb{R}^{m \times m}$ is the identity matrix, $H = I_{n_1+n_2} - (\frac{1}{n_1+n_2}) \mathbf{1}\mathbf{1}^T$ is the centering matrix where $I_{n_1+n_2} \in \mathbb{R}^{(n_1+n_2) \times (n_1+n_2)}$. Note that the constraint $W^T K H K W = I$ is introduced to avoid the trivial solution ($W = 0$), so that the transformed patterns do not collapse to one point. We have included the explanation in the revised manuscript.

6. Personally, I find it less than obvious how we go from the minimization problem presented at lines 465 and 466, to the maximization problem on line 470. The same is true w.r.t. how W is equivalent to the leading eigenvectors of $(I + \mu K L K = I_m)^{-1} K H K$. Could the authors please provide, either in the methods or the supplementary, a more detailed calculation/proof of this - or at least reference the theorems they are using.

Response: Thanks for your professional comments. As described above, the Lagrangian of the optimization problem is:

$$\begin{aligned} & tr(W^T K L K W) + \mu tr(W^T W) - tr((W^T K H K W - I)Z) \\ & = tr(W^T (I + \mu K L K) W) - tr((W^T K H K W - I)Z), \end{aligned}$$

where Z is a diagonal matrix whose diagonal entries are the Lagrange multipliers. Taking the derivative of the above Lagrangian with respect to W and setting the derivative to 0, we obtain $(I + \mu K L K)W = K H K W Z$. Multiply both sides of the equation by $(K H K)^{-1}$, we can further obtain $(K H K)^{-1}(I + \mu K L K)W = W Z$, which is a generalized eigenvalue problem. The solution of $W \in \mathbb{R}^{(n_1+n_2) \times m}$ is the eigenvectors corresponding to the m smallest eigenvalues of $(K H K)^{-1}(I + \mu K L K)$.

Therefore, the overall computational process can be summarized as follows. We first collect the scRNA-seq data X_S and the spatial transcriptomics reference data X_T , from which the matrices L , K , and H can be obtained. Then we can construct the transformation matrix W by selecting the eigenvectors corresponding to the top m smallest eigenvalues of $(K H K)^{-1}(I + \mu K L K)$. Finally, with the transformation matrix W , the data from two domains are mapped into the lower dimensional latent space $W^T K \in \mathbb{R}^{m \times (n_1+n_2)}$. After transposing and dividing $W^T K$, we obtain the latent feature representation $X'_S \in \mathbb{R}^{n_1 \times m}$ for scRNA-seq data and $X'_T \in \mathbb{R}^{n_2 \times m}$ for spatial transcriptomics data with true biological characteristics.

As you suggested, we have included the detailed description of calculation and derivation process in the revised manuscript.

7. The authors write (line 483-487): "scSpace first trains the model on scRNA-seq data using mean squared error (MSE) loss function. Once training is finished, scSpace then applied the model to scRNA-seq data and the spatial information of each single cell is reconstructed (we term 'pseudo-space')." Unless I've misunderstood the model, which is possible, I would assume that this is a typo and that the part in bold should say: "first trains the model on spatial transcriptomics data". Or am I mistaken?

Response: Thanks for your valuable comments. We apologize for the mistakes. You are right about the true meaning here. We intended to state that "scSpace first trains the model on spatial transcriptomics data" in the original manuscript. We have revised the sentence according to your advice and thus thoroughly checked all statements.

8. I could not find any details about the architecture of the multi-layer neural network (MNN). Going through the associated code, it seems to me as if they are using a two-layer fully connected model, with a sigmoid activation function. In addition, perhaps the authors could elaborate on why they choose the sigmoid instead of the more common relu - this would be of interest to the community and other people who are attempting to build similar models.

Response: Thanks for your constructive comments. In the original manuscript, we described the MNN algorithm but with only a little space. As you suggested, we have extensively elaborated the methodology. Specifically, when we introduced MNN algorithm at the beginning, we just thought that it could realize the scientific problem concerned by this research, and did not adjust the relevant parameters. We fully agree with you that the choice of parameters should be explained. In the modified version of scSpace, according to your valuable suggestion, we have added the ReLU activation function in addition to sigmoid and give the choice of activation function to users. We have included the description of the choice of the activation function in the revised manuscript.

9. Could the authors please elaborate on what sort of distance metric that is used to compute the KNN graph as well as the spatial weights $w = W(E_1)$.

Response: Thanks for your comments. In this study, scSpace applies space-informed clustering to identify spatially heterogeneous single-cell subpopulations based on the gene expression and the generated pseudo-space information of cells in scRNA-seq data. In detail, a gene expression graph $G_g(V, E_1)$ is first constructed on the reduced principal components derived from normalized gene expression using k -nearest neighbor (KNN) algorithm (top 50 PCs are selected by default). Since our goal is to find spatially heterogeneous subpopulations that may be similar in gene expression, the pseudo-space information of cells is expected to be transformed to the spatial weight w of each edge in gene expression graph $G_g(V, E_1)$:

$$w = W(E_1)$$

The spatial relationship of cells could be considered in the later unsupervised clustering step. We provide two strategies to define the spatial weight w .

(1) The spatial weight of edge $E_{i,j}$ between cell S_i and cell S_j is negatively

associated with their direct distance $d_{i,j}$ in the pseudo-space, which is defined:

$$w_{i,j} = \exp\left(-\frac{d_{i,j}^2}{2l^2}\right)$$

The hyperparameter l , also known as the characteristic length scale, determines how rapidly the covariance decays as a function of distance. This framework that works with the distances has also been employed in SpatialDE (Svensson et al., *Nat Methods*, 2018, <https://www.nature.com/articles/nmeth.4636>).

(2) A space graph $G_S(V, E_2)$ is constructed on the pseudo-space of cells using k -nearest neighbor (KNN) algorithm firstly, and the spatial weight of edge $E_{i,j}$ between cell S_i and cell S_j is negatively associated with their distance $d_{i,j}$ on the space graph $G_S(V, E_2)$, which is defined:

$$w_{i,j} = \frac{1}{\alpha + d_{i,j}} + \beta$$

Where α and β are pseudocounts to guard against excessively large and small weights, respectively. By default, $\alpha = \beta = 1$. The distance $d_{i,j}$ is calculated based on the adjacency matrix A of the space graph. Specifically, for given cell S_i and cell S_j , the distance $d_{i,j} = 1$ if S_j is the neighbor of S_i on the space graph or $d_{i,j} = 2$ if S_j is the neighbor of the neighbor of S_i and so forth.

Finally, scSpace applies unsupervised clustering on space-weighted gene expression graph using Leiden algorithm.

10. Important: I have one key concern here which relates to the theoretical justification of this method. The method embeds single cell and spatial transcriptomics features in the same space using TCA (which the authors state is used to correct for batch effects between the platforms), but these (spatial and single cell data) represent two inherently different types of observations. In Visium/ST data (which the authors use), the spots consists of a mixture of cells, not all necessarily from the same cell type – which the authors also acknowledge – meaning that the spatial expression profiles are vastly different from the single cell data, and unless the spatial observations are "near pure" (i.e., consists of mainly one cell type) it seems a bit "forced" to reconstruct single cell data (no mixing) using mixed data as a reference. I would have liked to see the model account for this mixing to some extent, but as of now, I see no such attempts.

Perhaps a decompositional (into cell type contributions) step in the network would be one such example. The results from the cortex show some success in the spatial reconstruction, but these layers are also more pure than what you'd expect to see in other tissues. In the developmental (embryonic) heart, where the observations are less "pure", the pseudo-space reconstruction is not overtly impressive. Before I would feel comfortable to approve this manuscript, I would have to see results showing a good reconstruction of a more complex tissue where there are more mixing of cell types.

Response: Thanks for your professional and constructive comments. The question raised can be separated into two parts, and I will respond one by one.

Question 1: Spatial (mixture of cells in each spot) and single-cell (pure) data represent two inherently different types of observations. How does scSpace account for this mixing?

We fully agree with you that this is an important issue underlying the integration analysis of heterogenous datasets with different scales or resolutions. Therefore, many methods have been developed to address this challenge such as RCTD (Cable et al., *Nat Biotechnol*, 2022), Cell2location (Kleshchevnikov et al., *Nat Biotechnol*, 2022), and SPOTlight (Elosua-Bayes et al., *Nucleic Acids Res*, 2021). However, most of the current deconvolution methods can only predict the composition and proportion of different cell types in each spot (Figure 10), rather than directly convert the transcriptome data of each spot into single-cell transcriptome data, which means the resulting "pure" data inherit the cell-type label instead of the corresponding gene expression profiles of single cells.

Figure 10. The workflow for above mentioned spatial deconvolution methods such as RCTD, Cell2location, and SPOTlight.

Nevertheless, the underlying hypothesis of these spatial deconvolution approaches is similar, namely that transcriptional features are shared between scRNA-seq and spatial transcriptomics data, resulting in pairwise correspondence between

cells and spots. The TCA used in scSpace also takes advantage of this intention by integrating single-cell and spatial features from both datasets. Because scSpace focuses on single-cell data instead of spatial transcriptomics data, the task is accomplished when spatial information is superimposed onto single cells without the need to know the cell-type composition of each spot. We have included the discussion in the revised manuscript.

Question 2: Can scSpace achieve good reconstructions of more complex tissues where there are more mixing of cell types?

In the original manuscript, we validated the spatial reconstruction performance of scSpace with the human and mouse cortex datasets and achieved success in the pseudo-space reconstruction across multiple tissue sections. Also, we acknowledge that the molecular structure of the human and mouse brain cortex is highly organized and the intrinsic gene expression distribution exhibits a layered spatial pattern. Therefore, merely reconstructing the layered structure of the cortex may not be convincing, and validations with a more complex organization should be included to evaluate the performance of scSpace. Herein, three examples with a more complex mixing of cell types are employed to illustrate the spatial reconstruction of scSpace.

(1) **Cortex data of the human middle temporal gyrus (MTG)**. Compared with the DLPFC dataset used in the original manuscript, the MTG dataset comprises a more complex mixing of cell types, including the layer information of cells and the taxonomy of 69 neuron subtypes.

(2) **Mouse kidney data with transcriptional zonation**. In addition to the three-layer zonal distribution pattern of cells, the kidney data also provides refined spatial architectures of the various cell populations in the thin limb of Loop of Henle (tl-LoH) and spatial diversity of the principal cells (PCs) distribution in the ureteric epithelium. As illustrated in the original publication, the diversity of cell subclusters are closely related to their spatial locations.

(3) **Squamous cell carcinoma (SCC) data of the human skin**. Compared with highly organized tissues such as the brain cortex, the tumor microenvironment has more molecular diversity and spatial heterogeneity. SCC slices from three replicates of a patient are used to evaluate the spatial reconstruction performance of scSpace.

The results are shown as follows.

1. Reconstructing the pseudo-space of neuron subtypes with scSpace

To begin with, we select the cortex data of the human MTG (Hodge et al.) derived from Allen Brain Institute (<https://www.nature.com/articles/s41586-019-1506-7>) as an example because the data span multiple cortical regions and provide ground truth of refined annotations of neuron subtypes at manual dissection. Compared with the cortex dataset used in the original manuscript, the MTG dataset shows more complex cell-type constitute (not just clustering by layers but both neuron subtypes and spatial location). Therefore, we employ the MTG dataset to evaluate the performance of scSpace to reconstruct the spatial relationship between excitatory/inhibitory neuron subtypes.

Figure 11. Spatial analysis of the Hodge dataset. **a**, The reconstructed pseudo-space of human MTG scRNA-seq data. **b**, Normalized pairwise distance between cells from L1 to different layers. **c**, The spatial distribution of excitatory neuron subclasses in the pseudo-space. **d**, Schematic map indicating average layer position of each excitatory neuron subclass. **e**, Normalized pairwise distance between different excitatory neuron subclasses to L1 (red) and L6 (purple). **f**, Schematic map indicating average layer position of two branches of inhibitory neuron subclasses. **g**, The spatial distribution of inhibitory neuron subclasses in the pseudo-space. **h**, Normalized pairwise distance between different Inhibitory neuron subclasses to L1 (left) and L6 (right).

As shown in Figure 11a, scSpace successfully reconstructed the spatial hierarchical structure of layer 1 (L1) to layer 6 (L6), and the normalized distance between cells and L1 increased layer by layer from L1 to L6 (Figure 11b), which was consistent with previous results.

We further applied scSpace to recover the layer distribution of excitatory and inhibitory neuron subclasses to explore its capability of spatial reconstruction of refined and complex spatial architectures. The spatial distribution of each excitatory and inhibitory neuron subclass was accessed from the original publication (Figure 11d and f). As illustrated in Figure 11c and e, scSpace accurately reconstructed the span-layer spatial architecture of excitatory neuron subclasses, different excitatory neuron types broadly segregated by layer in the pseudo-space.

For the two major branches of inhibitory neurons, which were distinguished by expression of ADARB2 and LHX6, respectively, scSpace successfully restored the spatial position relationship between them. In line with the ground truth of cell-subtype distribution in cortex layers for the ADARB2 branch (L1-L3) and the LHX6 branch (L4-L6) (Figure 11f), the predicted distribution of ADARB2 and LHX6 branches show heterogeneous spatial patterns in the pseudo-space (Figure 11g). Specifically, the distance between the ADARB2 branch and L1 is significantly closer than that between the LHX6 branch and L1 (Figure 11h), and the distance to L6 is opposite for the two branches (Figure 11h).

In conclusion, the results indicate that the pseudo-space reconstructed by scSpace has biological significance and rationalizes the subsequent cell distance recovering and space-informed clustering based on it.

2. Spatial analysis of the kidney zonation and the cell distribution in the tl-LoH and ureteric epithelium structure with scSpace

To investigate the ability of scSpace in restoring the relative spatial associations among cells, we focused on existing scRNA-seq data that are obtained from the mouse kidney dataset (Ransick et al., <https://doi.org/10.1016/j.devcel.2019.10.005>). The single-cell data are well-annotated based on the cell types and the zones (Figure 12a). After being allocated to the pseudo-space, cells distribute variedly from region to region, exhibiting a diverse functioning zonation in the tissue microenvironment, which is consistent with the spatial deconvolution results by RCTD (Cable et.al., *Nat*

Biotechnol, 2022, <https://www.nature.com/articles/s41587-021-00830-w> (Figure 12b). The “cortex – outer medulla – inner medulla” three-layer zonal distribution pattern of cells in the kidney (Figure 12c) is restored by scSpace (Figure 12d). Moreover, the pairwise distances between cells from different zones to Z1 and Z3 in the pseudo-space also show a consistent pattern with the ground truth of the zone distribution (Figure 12e).

Figure 12. Spatial analysis of the mouse kidney dataset. **a**, t-SNE visualization of mouse kidney scRNA-seq data, the single-cell annotation was obtained from the original publication, colored by cell types (left) and kidney zones (right). **b**, Left, pseudo-space of the single-cell data reconstructed

by scSpace. Right, deconvolution of the spatial transcriptomics data by RCTD. **c**, Kidney zonation and the spatial genomics of the zonal dissections. **d**, Spatial distribution of the single cells from different zones in the reconstructed pseudo-space. **e**, Pairwise distance between cells from different zones to Z1 (left) and Z3 (Right). **f**, Schematic map indicating anatomic position for cell populations of thin limb of loop of Henle (tl-LoH). **g**, Spatial distribution of the tl-LoH cell populations in the pseudo-space. **h**, Pairwise distance between cells from different tDL subpopulations to tDL (17) (red) and tDL (4/14) (purple). **i**, Schematic map indicating anatomic position and ontology terms for principal cells (PCs) of the ureteric epithelium. **j**, Spatial distribution of PCs in the pseudo-space. **k**, Pairwise distance between cells from different regions to region 20 (red) and region 32 (purple).

In addition, the reconstruction performance of scSpace is evaluated with more refined spatial architectures of the various cell populations of the thin limb of Loop of Henle (tl-LoH) as well as the principal cells (PCs) of the ureteric epithelium, whose diversities relate to the positions along the cortical-medullary axis at least in part. As illustrated in Figure 12f, the cell populations of tl-LoH exhibit the relative order from the Outer medulla to the Inner medulla. As shown, scSpace accurately reconstructs the spatial distribution of these cell populations in the pseudo-space (Figure 12g). Notably, in line with the ground truth, tAL (15) distributes between tDL (6) and tDL (4/14) in the pseudo-space. Furthermore, for the four tDL subtypes, the average pairwise cellular distance between subtypes and tDL (17) presents a monotonically increasing trend along the “tDL (17) - tDL (12) - tDL (6) - tDL (4/14)” axis and an opposite trend in reverse, which is identical to the ground truth of cell distribution (Figure 12h).

Similarly, we investigate whether scSpace can recover the real spatial distribution of PCs in the ureteric epithelium (Figure 12i). As illustrated in Figure 12j, PCs from different regions reveal a spatial diversity of distribution in the pseudo-space. Along the cortical-medullary axis, the pairwise distance between cells from different regions to region 20 (red) and region (32) exhibit an increasing and a decreasing trend, respectively.

The spatial analysis of the mouse kidney dataset indicates that scSpace can recover the spatial association between cells in a more complex tissue.

3. Spatial reconstruction of the fibrovascular niche in human SCC data by scSpace

The tumor microenvironment is very complex due to its transcriptional and spatial

heterogeneity. Therefore, it is appropriate to use tumor data to evaluate the robustness of scSpace. Here, the SCC dataset is employed to show the spatial reconstruction performance of scSpace.

Figure 13. Spatial reconstruction of human SCC scRNA-seq data using scSpace. **a**, t-SNE visualization of human SCC scRNA-seq data, the single-cell annotation is obtained from the original publication. **b**, Three replicates of spatial transcriptomics data from Patient 2 are utilized as the spatial references. **c**, Left, the pseudo-spaces of scRNA-seq data reconstructed by scSpace using different replicates of spatial data. Right, the fibrovascular niches in different pseudo-spaces. **d**, Cell-type deconvolution results for different spatial transcriptomics references by RCTD. **e**, Expression patterns of TSK score in the pseudo-space.

As shown in Figure 13a, the SCC data comprise 14 cell types that were annotated

by the original publication (Ji et al., <https://doi.org/10.1016/j.cell.2020.05.039>). Three replicates of spatial transcriptomics data from the same patient (Patient 2) are used as spatial references (Figure 13b). As illustrated in Figure 13c, the TSKs, fibroblasts, and endothelial cells of scRNA-seq data showed specific patterns of colocalizations in the pseudo-space, despite using the different spatial references to perform spatial reconstruction by scSpace. Consistent with the original publication by Ji et al. and the deconvolution results of the spatial reference by RCTD (Figure 13d), fibroblasts and endothelial cells were enriched at the TSK-high leading edge, further supporting a fibrovascular niche surrounding TSK cells. Besides, the pattern of the TSK score based on markers defined by Ji et al. in the pseudo-space reconstructed by scSpace is also in agreement with the spatial structure of the TSK-proximal fibrovascular niche (Figure 13e).

With the success in spatial analysis of more complex tissues in multiple circumstances, we have full confidence to believe that scSpace can reconstruct the pseudo-space of single cells, restore the spatial relationship between cells, and identify spatially heterogeneous cell subclusters. The spatial analysis of three examples are included in the Extended data.

11. Important: The "Simulated data analysis" section is also lacking in detail. It's not clear to me how the synthetic data was generated, which makes it hard to evaluate whether this is an appropriate approach or not. How is spatial structure introduced into the synthetic spatial data? Furthermore, I could not find scripts or code to reproduce this data, if those exist – could the authors please point me to their location.

Response: Thanks for your professional comments. We fully agree with that the simulation approach is very important for reader to interpret the function of scSpace. Therefore, we have included more descriptive information in the revised manuscript for the data simulation.

Specifically, we applied Splatter R package (v1.16.1) to simulate 140 paired scRNA-seq and spatial transcriptomics data with 5,000 expression genes. In order to simulate the original batch effect between these two types of data, we set up two batches representing scRNA-seq and spatial transcriptomics data, respectively. In order to simulate the cell subclusters with similar transcriptome profiles but being spatially heterogeneous, we random set 2 to 10 cell populations as spatially heterogeneous

subclusters and set the probabilities of a gene being differentially expressed in each of them as 0.01 (the differentially expressed probabilities are set as 0.2 in cell populations with different transcriptome profiles by contrast).

Next, for each cell in spatial transcriptomics data, a pseudo spatial coordinate was assigned based on random sampling and normal distribution strategy. Specifically, given a simulated spatial transcriptomic with N cell populations, we random sampled N points in the 20×20 range as spatial coordinate centers of cell populations. Then for cells in each cell population, we randomly generated the spatial coordinates for the normal distribution with mean equal to the spatial coordinate centers of this cell population and standard deviation equal to 1.

Moreover, for the robustness of results, we set a gradient from 500 to 1500 for the number of cells, and a gradient from 3 to 14 for the number of cell populations. The detailed description of the experimental design was summarized in Extended Data Fig. 1 and Supplementary Data 2. In addition, a well-documented source code for the synthesis of simulated data is provided in the GitHub. (<https://github.com/ZJUFanLab/scSpace/blob/master/AnalysisPaper/scripts/constructSimulations.R>).

12. The formula on line 523 doesn't really make sense to me, should the denominator be the max taken over all pairs of indices, and not specifically i and j?

Response: Thanks for your constructive comments and we fully agree with you that the formula is not necessary to list. In the revised manuscript, we have removed this formula.

Results

I'm not a biologist, so I've refrained from evaluating any of the "biological conclusions", instead my comments pertain to the construction of the analysis and the strategies used to evaluate the method's performance.

Response: Thanks for your constructive comments. In fact, the questions raised by you are professional, insightful, and important in the biological perspective. We value your comments very much and thus we have considered your advice carefully and tried to

address the concerns point by point.

1. I appreciate that the authors benchmark their method, but would argue that their choice of methods to benchmark against is a bit surprising. None of these are "spatially aware" methods, and many such methods have been developed - specifically for spatial transcriptomics data. One example being: <https://doi.org/10.1093/nar/gkac219>. To make the comparison more relevant I would encourage the authors to revise their selection of methods for benchmarking and pick a set of methods designed for similar purposes as theirs.

Response: Thanks for your valuable comments. We fully agree to compare the performance of scSpace with other algorithms. In the original manuscript, we didn't compare scSpace with spatial clustering algorithms because the main task for scSpace is to identify spatially variable cell subpopulations from scRNA-seq data, while spatial clustering methods target the spatial transcriptomics data for spatial domains identification. As you suggested, the DR-SC (*Nucleic Acids Res*, 2022) as well as three other spatial domain identification methods, STAGATE (*Nat Commun*, 2022), BayesSpace (*Nat Biotechnol*, 2021), and SpaGCN (*Nat Methods*, 2021) are included in the updated benchmarking. Besides, the scCoGAPS (*Cell systems*, 2019), a non-negative matrix factorization (NMF) algorithm that can learn latent spaces and find cell-type signatures from single-cell RNA-seq data, is included in the benchmarking.

Figure 14. Comparison of the performance between scSpace and other algorithms. y axis, Adjusted Rank Index (ARI) of each method using simulated data.

As shown in Figure 14, scSpace outperforms other algorithms with a significantly higher ARI score using 140 simulated datasets with subcluster numbers ranging from 2 to 10. Because scSpace targets scRNA-seq rather than spatial transcriptomics data, and considers not only transcriptional features, but also pairwise distances between cells via reconstructing the latent pseudo-space during clustering, its performance is better than other methods that only focus on transcriptional features such as Louvain, Kmeans, scCoGAPS, and Hclust. Meanwhile, other four spatial domain identification methods performed (BayesSpace, STAGATE, DR-SC, and SpaGCN) perform poorly in the space-informed single-cell analysis since they aim to identifying spatial niches or domains in spatial transcriptomics data rather than spatially heterogeneous cell subpopulations in scRNA-seq data. The benchmark result and corresponding discussion are included in the Extended Data.

2. From the images in Figure 1b, it's not clear whether we are looking at the single cell data in pseudo-space or in some other reduced space.

Response: Thanks for your valuable comments. In the previous version, single-cell data were presented as the reduced space in tSNE layout. We have revised this figure and included it in the Extended data.

3. What sort of statistical test was used to compare the ARI scores in Figure 1d? I'm not sure if these scores are valid to compare using some of the classical parametric tests, as they aren't iid.

Response: Thanks for your professional comments. In the manuscript, we used the Wilcoxon rank-sum test, a nonparametric test used for comparison of two independent samples. Wilcoxon rank-sum test has been widely used in scRNA-seq data analysis for comparison of an index or characteristic between two groups, such as cell-type proportion (Melms et al., *Nature*, 2021, <https://www.nature.com/articles/s41586-021-03569-1>), expression level of differentially expressed genes (Hao et al., *Cell*, 2021, <https://doi.org/10.1016/j.cell.2021.04.048>), and of course, ARI scores of cell-type clustering.

4. In Figure 1c the title says "distance to origin" while the figure text states that this is the correlation between cells in pseudo-space and the real space. Which one is it, and how exactly was this computed? Personally, I would not have measured the distance to the origin, this doesn't necessitate that spatial relations are preserved. Instead I would recommend to create two – one for the pseudo-space and one on the original data – $0.5 \cdot (n^2 - n)$ long vectors containing the distance between every pair of cells.

Response: Thanks for your insightful comments. We fully agree with you that the original 'distance to origin' illustration doesn't represent the exact spatial relationship between pseudo-space and the origin. In the revised manuscript, according to your valuable advice, we compute the pairwise distance of cells in the original space and the pseudo-space, respectively, and then calculate the Pearson Correlation Coefficient (PCC) between them.

5. I endorse the evaluation of the method's performance using Visium and STARmap data, this is - in my opinion - a clever strategy. However, I would have liked to see a similar evaluation using a more complex tissue where cells aren't sorted into layers.

Response: Thanks for your constructive comments. As we know, the molecular signature of the tissues is highly heterogeneous between patients and even within tumors, thus using complex tissues such as cancer data to evaluate scSpace may better validate the performance of the method. Therefore, to further evaluate the spatial reconstruction performance of scSpace, it is applied to spatial transcriptomics data from more complex tissues, such as the above mentioned human SCC dataset (Ji et al., <https://doi.org/10.1016/j.cell.2020.05.039>), and the human HER2 breast cancer (BC) dataset (Andersson et al., <https://www.nature.com/articles/s41467-021-26271-2>). The ground truth of the spatial heterogeneity of transcriptomics, the reconstructed pseudo-space by scSpace, and the pairwise distance correlation between cells in the SCC and BC datasets are illustrated in Figures 15.

As shown, the Pearson correlation coefficient (PCC) scores are consistent across tissue replicates from the same donors (Patient 2 in SCC dataset and Patient E in BC dataset), suggesting that scSpace can achieve robust performance in multiple tissue sections in most scenarios where spatial heterogeneity exists stably within tissues. Meanwhile, it needs to be discussed separately in practice, because different replicates from the same tissues can share a large spatial coherence or a small one. In

conclusion, we demonstrate that scSpace can reconstruct the pseudo-space of more complex tissues where cells aren't sorted into layers. We have included the analysis and the discussion in the revised manuscript.

Figure 15. The original distribution of transcriptomics, the reconstructed pseudo-space by scSpace, and the pairwise distance correlation between cells among different donors and tissue replicates in the SCC (a) and BC (b) datasets.

6. Another kind of evaluation that would be interesting is to see how the pseudo-space structure compares with mapping of the single cells using, for example, stereoscope, Tangram, cell2location, or RCTD. This could, to some extent, validate the methods performance in "mixed" spatial transcriptomics data (e.g., Visium or ST).

Response: Thanks for your constructive comments. We fully agree with you that the spatial deconvolution results can be used to validate the performance of scSpace. Here, three examples are used to illustrate the pseudo-space reconstruction of single cells by scSpace.

(1) Comparison between pseudo-space reconstructed by scSpace and spatial deconvolution result by RCTD with the human melanoma dataset.

We performed scSpace to reconstruct the pseudo-space of scRNA-seq data from the human melanoma. Here, we focused on the spatial organization of T cell

subpopulations. As shown in Figure 19a and 19b, the C5 T cell subpopulation was significantly nearest to the malignant cells, while the C3 T cell subpopulation was the opposite in the pseudo-space reconstructed by scSpace. Then, we investigated whether the spatial relationship between T cell subpopulations predicted by scSpace can be validated by spatial deconvolution of the spatial reference with RCTD. As shown in Figure 19c, the cell-type proportions of each spot in spatial transcriptomics data is calculated by RCTD. Conformably, C5 and C3 subpopulations were mainly located in the tumor and lymphatic regions, respectively (Figure 19c).

Figure 19. Comparison between the spatial deconvolution by RCTD and the pseudo-space reconstruction by scSpace on human melanoma scRNA-seq data. **a**, The spatial distribution of five T cell subpopulations in the pseudo-space. **b**, Normalized pairwise distance between different T cell subpopulations to malignant cells. The Wilcoxon rank-sum test was performed. **c**, The spatial distribution of C3 and C5 subpopulations in the spatial transcriptomics reference calculated by RCTD.

(2) Comparison between pseudo-space reconstructed by scSpace and spatial deconvolution result by RCTD with the human SCC dataset.

In addition, scSpace is applied to identify the spatial niche in the tumor microenvironment (TME) with the above-mentioned human SCC dataset. As illustrated in Figure 20a, the pseudo-spaces of single cells from three replicates of Patient 2 are reconstructed with similar spatial architectures. Then, we focus on the fibrovascular niche, which consists of three cell types, namely the TSKs, fibroblasts, and endothelial cells. As shown in Figure 20b, these three cell types show specific patterns of

colocalizations in the pseudo-space, despite using different spatial references to perform spatial reconstruction by scSpace. In concordance with the previous study and the deconvolution results of the spatial reference calculated by RCTD (Figure 20c), fibroblasts and endothelial cells are enriched at the TSK-high leading edge, further supporting a fibrovascular niche surrounding TSK cells.

Figure 20. Comparison between the spatial deconvolution by RCTD and the pseudo-space reconstruction by scSpace on human SCC scRNA-seq data. **a**, The reconstructed pseudo-space of human SCC scRNA-seq data by scSpace. **b**, The spatial distribution of TSKs, fibroblasts, and endothelial cells in the pseudo-space. **c**, The RCTD deconvolution results of the spatial transcriptomics reference.

(3) Comparison between pseudo-space reconstructed by scSpace and spatial deconvolution result by RCTD with the mouse kidney dataset.

Furthermore, scSpace is performed to validate whether the pseudo-space of refined cell types can be accurately reconstructed. Therefore, we compared the spatial distribution of different cell types in the pseudo-space with the spatial deconvolution results of the spatial reference. As shown in Figure 21a, the different cell types in kidney demonstrate a clear zonal distribution pattern in the pseudo-space

reconstructed by scSpace. The spatial relationship between cell types are consistent with the ground truth provided by the original publication. Moreover, the spatial deconvolution of the spatial reference by RCTD also supports this observation. As illustrated in Figure 21b, the distal tubules and proximal tubules are mainly located in the cortex, while the thin limb of Loop of Henle and principal cells are mainly located in the medulla.

Figure 21. Comparison between the spatial deconvolution by RCTD and the pseudo-space reconstruction by scSpace on mouse kidney scRNA-seq data. **a**, The reconstructed pseudo-space of mouse kidney scRNA-seq data by scSpace. **b**, The RCTD deconvolution results of the spatial transcriptomics reference.

The spatial deconvolution results by RCTD have extensively supported the performance of scSpace on pseudo-space reconstruction. Relevant analysis and discussion have been included in the Extended data.

7. The authors state that traditional clustering methods like "Louvain" had issues finding the finer subclusters in DLPFC – did the authors evaluate different clustering settings here or did they just use the default ones?

Response: Thanks for your professional comments. When datasets are used to compare the clustering between scSpace and other traditional clustering methods such as Louvain, the cluster number for all benchmarking methods is identical with the ground truth provided by the original publication. Also, we agree that the investigation on whether clustering settings would affect the performance of Louvain on identifying finer subclusters should be included. Therefore, the cardiomyocytes data with refined annotations and spatial locations of cell clusters are used again to evaluate the clustering performance of scSpace and Louvain.

Figure 20. Comparison the clustering performance of scSpace and Seurat on human embryonic heart scRNA-seq data. **a**, t-SNE of human embryonic heart scRNA-seq data, the single-cell annotation was obtained from the original publication. **b**, t-SNE of three cardiomyocytes with the original annotation and scSpace's clustering result. **c**, t-SNE of three cardiomyocytes with Seurat's clustering result under different targeted cluster number setting (from 14 to 23). **d**, The change of ARI for Seurat under different targeted cluster number setting.

In order to avoid the misclassification of three types of cardiomyocytes (Figure 20a) by Louvain (Seurat) due to the lack of clustering resolution, we have also explored the influence of targeted cluster number (K) on the clustering results of Louvain. The K for scSpace is identical with the original number annotations, and scSpace has successfully

distinguished the ventricular cardiomyocytes from the MYOZ2-enriched cardiomyocytes (Figure 20b). For Seurat, we gradually increase K from 14 to 23, and as shown in Figure 20c, with the increase of K, the atrial and ventricular cardiomyocytes are indeed identified by Seurat at a higher clustering resolution, however, the ventricular and MYOZ2-enriched cardiomyocytes are still not well separated. The ARI scores of all cell types or cardiomyocytes are consistently lower than scSpace (Figure 20d). In conclusion, the present results indicate that scSpace is a relatively accurate and efficient method for identifying the subpopulations that are similar in transcriptome but heterogeneous in space.

The results from multiple circumstances further validate the superior performance of scSpace on identifying spatially heterogeneous cell subpopulations to traditional clustering methods such as Louvain.

8. Would it be possible for the authors to validate their pseudo-space arrangement by looking at receptor-ligand interactions. If any single cell data set with known cell interactions exists, one way to evaluate the spatial reconstruction is to see whether cells expressing cognate ligands and receptors locate in each others vicinity.

Response: Thanks for your insightful comments, we value the suggestion very much. Therefore, we applied SpaTalk (Shao et al., *Nature Communications*, 2022, <https://www.nature.com/articles/s41467-022-32111-8>), a method for decoding the cell–cell communications in space, over the SCC scRNA-seq data with pseudo-space information, to validate whether the pseudo-space constructed by scSpace preserves the spatial interaction between different cell types.

We focused on the cellular crosstalk at the TSK-high leading edge niche, since it was reported that TSKs participated in extensive autocrine and paracrine interactions (mostly with fibroblasts, endothelial cells, macrophages, and MDSCs) (Ji et al., *Cell*, 2020, <https://doi.org/10.1016/j.cell.2020.05.039>). Consistent with the previous study results, the major spatially resolved cell–cell communications between TSKs and stromal cells of the fibrovascular niche in the TME are preserved in the pseudo-space of SCC scRNA-seq data.

For example, prominent TSK signaling to fibroblast and endothelial cell is mediated by several common ligand-receptor pairs which are close to each other in the pseudo-space, including PGF-NRP1, TNC-SDC1, PGF-FLT1, and EFNB1-EPHB4 (Figure 21a).

Conversely, fibroblast and endothelial cell prominently co-expressed numerous ligands such as TFPI, FN1, THBS1, and HMGB1 (Figure 21b), matching TSK receptors that promote the proliferation and differentiation of TSKs. Notably, similar spatially resolved ligand-receptor-interactions mediating the TSK-stroma communications were also observed in the pseudo-space constructed by scSpace using other spatial references (Figure 21c).

In conclusion, we demonstrate that scSpace can precisely reconstruct the spatial arrangement of single cells as well as the spatially resolved cell-cell interactions. We have included the analysis of spatial cell-cell communications between cell types in the SCC dataset in the revised manuscript.

Figure 21. Spatial cell-cell communication of cell types in the pseudo-space. a, Spatial cell-cell

communication from TSK signaling to fibroblasts and endothelial cells are enriched by SpaTalk in the the pseudo-space. **b**, Spatial cell-cell communication from fibroblasts and endothelial cells to TSK signaling are enriched by SpaTalk in the the pseudo-space. **c**, Shared ligand-receptor pairs enriched in three replicates from Patient 2 in the SCC dataset.

Discussion

1. The authors do a great job of summarizing their work in the Discussion, but it's still not clear to me what the USP (unique selling point) of their method is. For example, albeit the methods are different, what sort of insights will this kind of analysis give us that deconvolution of spatial transcriptomics data can't. Also, although – as the authors point out - many of the (sub) single cell resolution spatial transcriptomics methods target ~10000 genes (and some can even target the full-transcriptome, e.g., FISSEQ), which is usually sufficient to distinguish fine cell types. Could the authors give some examples of cases where their method offer unique insights that cannot be obtained by other means.

Response: Thanks for your thought-provoking comments. I think the USP of scSpace recovers the spatial relationship between cells for scRNA-seq data compared with other spatial integration methods that focus on the cell-type decomposition of the spatial transcriptomics data. As reported in many publications, the spatial information of a cell could become the determinant in its identity. However, most spatial integrative methods use the existing cell-type annotations of cells characterized by only transcriptional information. In contrast, scSpace fully considers the pairwise spatial association between cells and embeds the spatial information in the clustering. Not surprisingly, scSpace is able to find spatially heterogeneous cell subpopulations that could not be identified by conventional clustering methods. And for spatial integrative methods such as Cell2location, RCTD, and BayesSpace, etc., the fundamental task is different from scSpace.

In addition, even though current targeted methods such as MERFISH and seqFISH can measure over 10,000 RNA species as well as observe at subcellular resolution, they failed to detect gene variations, which directly reflect the function of specific genes. In this perspective, sequencing methods have the natural advantage of detecting unbiased transcripts from tissues, which is why spot-based RNA-seq is still the mainstream of the spatial transcriptomics technology, despite its inability to sequence individual cells. On the other hand, in situ short-read sequencing methods such as

FISSEQ and ISS sequence very short segments of each transcript (perhaps 20-30 bases), which leads to a bias of results. Also, even though these methods showed success in in situ sequencing with cultured cells, they can hardly be applied to tissue sections. Notably, single-cell transcriptome data still dominates the cutting edge of omics technology, with unmet biological and clinical needs. Currently, transcriptome atlas with hundreds of thousands to millions of samples are basically scRNA-seq, which cannot be replaced by spatial transcriptomics data. The reanalysis of scRNA-seq data by combining spatial information is meaningful. Besides, the limitations of spatial transcriptome techniques make computational methods such as scSpace to reconstruct spatial relationships between cells within scRNA-seq data more desirable than ever before.

2. Another aspect that I'm not fully clear on is what the main feature is of this method, is it the spatial reconstruction or the spatially informed clustering? What is it that the authors themselves consider most novel about their approach.

Response: Thanks for your insightful comments. In my opinion, the key idea of scSpace is to reconstruct the pseudo-space of single cells for scRNA-seq data. All subsequent analysis such as spatially heterogeneous cell-type identification, spatial cell-cell communication inference, immune cell invasion prediction, and spatial cell dynamics discovery can be revealed based on the successful reconstruction of the pseudo-space. Besides, this is the intrinsic difference between scSpace and other spatial integration methods. scSpace focuses on single-cell data and thus the spatial transcriptomics data are used to enhance the information of single-cell data. This sets scSpace apart from other methods in that they focus on spatial transcriptomics data and attempt to solve the issue of obtaining single-cell resolution from ST data. In the revised manuscript, we have highlighted the main feature of scSpace and fully discussed its novelty.

3. In non-stereotypical tissues, e.g., cancer where the cells' spatial distribution are less consistent between tissue samples, and might even be specific to the patient – does it really make sense learn the structure from one individual in order to reconstruct the spatial structure of the single cell data of another individual? This is something I'd like to see the authors address in their discussion. The inverse operation, mapping defined cell types (from single cell data) onto an existing spatial assay, resonates much better

with me – what is the authors' opinions on this matter?

Response: Thanks for your professional comments. We fully agree with you that in non-stereotypical tissues, such as cancer, the spatial distribution of cells is less consistent between tissue samples, and might even be specific to the patient. Indeed, in the analysis of 12 slices from human cortex and 12 slices from the human SCC (cancer), we observed a robust performance of scSpace on the well-organized cortex, yet inconsistent result of several samples in the SCC, where low spatial coherence is shared between patients or even replicates of slices. This is certainly an issue worth discussing, and therefore, users are encouraged to select paired datasets with comparable conditions, states, and origins before conducting the scSpace analysis.

However, we still feel positive to scSpace because in most scenarios, such as the cortex, the intestine, the liver lobule, the kidney, the embryonic heart, the melanoma, and the breast cancer, etc., scSpace can robustly reconstruct the spatial relationship between cells with spatial references from different individuals. These successes also indicate that even though the tissues vary in morphology, their intrinsic spatial relationship between cells is preserved across multiple slices. In this perspective, the reconstruction of the pseudo-space (relative cell distances) is as important as the mapping of cell types on the spatial reference (absolute coordinates). On the contrary, such unpaired issue still exists in the inverse operation by mapping cell types to the spatial reference. Typically, single-cell and spatial transcriptomics data are derived from separated experiments. When considering highly heterogeneous tumor tissues, the different expression level, cell-type composition, and cell states between single-cell and spatial transcriptomics data can affect the result.

Moreover, while there are similarities in the integration of single-cell and spatial transcriptomics data between scSpace and other spatial reconstruction methods such as Cell2location, RCTD, etc., we highlight the fundamental difference between the methods. Specifically, scSpace focuses on the spatial analysis of scRNA-seq data whereas other methods target the cell-type decomposition of ST data. As mentioned above, scRNA-seq still dominates the cutting-edge omics technology and it cannot be replaced by spatial transcriptomics methods. The reanalysis of scRNA-seq data by combining spatial information is desirable and meaningful.

Reviewer #3 (Remarks to the Author):

Reviewer's summary

In this manuscript Liao et al. present scSpace, a method to retrieve spatial relationship between single-cells by integrating single-cell and spatial transcriptomics using transfer learning model. They apply their method initially on simulated data but also on real experimental datasets. They claim that scSpace can map the single-cell data into their pseudo-space which corresponds to their real spatial localization on the tissue. Also, they try to show using a spatially-informed clustering approach of scSpace improves the clustering accuracy. The whole idea is very interesting but I believe there is still some work needed to make the manuscript much more understandable and the significance of the scSpace clearer. My main concerns are:

If what is represented as pseudo-space is really capturing the spatial localization of the cells/spots or is it very similar to what a diffusion component would provide as well without integrating spatial information.

Are the claims that are made related to improved clustering of scSpace compared to other algorithms really fair, or this is just due to under-clustering using other benchmarked algorithms?

Response: We thank the review for the positive feedback and insightful summary of our work. As shown below in our detailed responses, we have carefully addressed the concerns raised and find that the constructive comments have extensively improved the quality of the manuscript in this revised version. In this study, a transfer learning algorithm, termed transfer component analysis (TCA), is used to extract the shared latent features between single-cell and spatial transcriptomics data. The represented pseudo-space of single cells shows consistent spatial architecture compared with the spatial transcriptomics data when we conduct subsequent space-informed clustering and spatial cell-cell communication inference. In my opinion, scSpace captures the pairwise spatial association between cells, which is different from the diffusion component. We fully agree with you that more clustering and space-aware algorithms should be included for the benchmarking of the performance of scSpace. Therefore, we comprehensively benchmarked different methods in the revised manuscript to ensure transparency of comparison between methods. All significant changes have been highlighted in blue.

Major comments

1. Line 85-85: Since the authors are describing the use of TCA to integrate single-cell and spatial data, it would be great if they could elaborate more on how they define the model? Which dataset is being used as the source and which one as target? (Is it from Single-cell to spatial or the other way around?)

Response: Thanks for your constructive comments. In the revised manuscript, we have devoted large efforts to make the method more descriptive and easy interpreting. Briefly, the workflow of scSpace comprises three main components: (1) extract the shared latent biological feature representation across scRNA-seq and spatial transcriptomics data, (2) reconstruct the spatial architectures of scRNA-seq data, and (3) identify spatially heterogeneous cell subpopulations over single-cell data that has been spatially reconstructed (optional).

Specifically, scSpace first applies a transfer learning model, termed Transfer Component Analysis (TCA) to eliminate the batch effect between scRNA-seq and spatial transcriptomics data and extract shared biological characteristics across these two domains. After this, using the shared biological characteristics extracted from spatial transcriptomics data, scSpace trains a multi-layer perceptron model to learn the relationship between characteristics and spatial coordinates. The trained model is then applied to the feature representation extracted from scRNA-seq data for spatial reconstruction, where the spatial coordinates generated for the single cells of scRNA-seq data are termed “pseudo space”.

Additionally, scSpace can further perform the space-informed clustering step to identify spatially heterogeneous cell subpopulations in scRNA-seq data. Based on the Leiden algorithm (Traag et al., *Scientific reports*, 2019, <https://www.nature.com/articles/s41598-019-41695-z>), a classical clustering method widely used in single-cell data analysis, we extend its applicability by introducing the spatial weight of edges in the gene expression graph constructed from gene expression profiles. Therefore, scSpace can take into account both gene expression and pseudo-space information of cells in the clustering process.

The detailed description of the TCA and pseudo-space reconstruction is highlighted in Line 708-804 in the revised manuscript.

2. In Figure 1a, the schematic that describes the algorithm, it would be great if it reflects better what is explained in paragraph one. For example, to show that in the “spaced informed expression graph”, spatial information is embedded as edge weights.

Response: Thanks for your insightful comments. We fully agree with you that the workflow of scSpace should be informative for better interpreting. According to your valuable suggestions, we have extensively reorganized the workflow as follows. As shown in Figure 23a, the workflow of scSpace comprises three main components: (1) extract the shared latent biological feature representation across scRNA-seq and spatial transcriptomics data, (2) reconstruct the spatial architectures of scRNA-seq data and (3 optional) identify spatially heterogeneous cell subpopulations over single-cell data that has been spatially reconstructed.

Given the scRNA-seq data (SC) and spatial transcriptomics reference (ST), scSpace co-embeds these two types of data into a shared latent space and extracts the shared latent features (Figure 23b). Using the characteristic matrix from ST data, scSpace trains a multi-layer perceptron model with spatial coordinates as the outcome and latent features as the predictors. The trained model is then applied to the characteristic matrix from SC data for pseudo-space reconstruction. Based on the gene expression profiles as well as the pseudo-space information, scSpace identifies the spatially variable cell subpopulations from scRNA-seq data (Figure 23c).

The detailed description of “spaced informed clustering” can be found in Line 805-835 in the revised manuscript.

Figure 23. Workflow of the scSpace. a, Overall procedure of scSpace. **b**, Transfer component analysis (TCA). **c**, Space-informed clustering.

3. The result of space-informed clustering is mapped back on the pseudo-space right? It would be better if the authors can show that on the schematics.

Response: Thanks for your professional comments. You are right about the space-informed clustering. In the revised manuscript, we therefore have included that on the schematics as shown in Figure 23.

4. Also in supp.Fig 1a, I can't follow what is happening after the Transfer component analysis in the schematic. I expect the output of the TCA is n space_informed reduced dimensionality data points (so single-cell and spatial data merged), then why after this step we still have a separate green and blue box? What do they represent? In the same figure, the Multilayer perception is mentioned but I am missing this in the text, can the author explain why and where this has been used, what is the output and how they have used it? (I have read the methods section, but still feel this schematic is not representing well the algorithm)

Response: Thanks for your professional comments. We apologize for the misunderstanding of the schematic of scSpace. In the revised version, the schematic of the Transfer component analysis (TCA) has been included in Fig. 1 in the main text. As shown, since single-cell and spatial transcriptomics are two different types of data, direct integration of them could lead to huge bias. Therefore, TCA is employed to extract the latent features from both data. Then, the shared latent feature is merged by optimizing the maximum mean discrepancy (MMD) between transformed single-cell and spatial transcriptomics. Indeed, the output of TCA is a space-informed matrix, however, the characteristic matrixes of SC and ST can be divided from the shared feature and are used for the next spatial reconstruction step of scRNA-seq data. Specifically, using the shared biological characteristics extracted from spatial transcriptomics data, scSpace trains a multi-layer perceptron model to learn the relationship between characteristics and spatial coordinates. The trained model is then applied to the feature representation extracted from scRNA-seq data for spatial reconstruction, where the spatial coordinates generated for the single cells of scRNA-seq data are termed "pseudo-space". the detailed description of "spaced informed clustering" can be found in Line 805-835 in the revised manuscript.

5. Line 90-92 the authors mention that “After the transformation was determined, single cells were mapped to an assumed coordinate system (pseudo-space) characterized by the MNN model using existing spatial references”, which means the “pseudo_space” is basically the MNN determined x-y coordinates right? This MNN step has not been described in none of the schematics (neither Fig1a nor Supp.Fig 1 a)

Response: Thanks for your insightful comments. We are sorry for the difficulty in interpreting the methods. The multi-layer neural network (MNN) model stated in the original manuscript is actually multi-layer perceptron (MLP). We didn't have a unified statement in the original version, which caused misunderstanding. As described in Question 4, the 'pseudo-space' is determined by applying the trained MLP model with ST features to the single-cell features.

In the revised manuscript, we have thoroughly checked all statements and consulted several mathematical experts to ensure accuracy of all description.

6. In general, I think the schematic that describes the algorithm should be revised extensively and be more followable.

Response: Thanks for your constructive suggestion, we fully agree with you that the schematic of scSpace should be revised extensively for all readers. As illustrated in Figure 23, we have reorganized the workflow. The updated Fig. 1 can be found in Line 142 in the revised manuscript.

7. Line 106-108: Comparing scSpace with k-means, Louvain and Hclust is very tricky. These algorithms can produce different clusters by different parameter settings. How the authors are confident that what they represent in the benchmark as the result of the for-example k-means clustering, is the best result of k-mean clustering? Maybe just increasing the K would result on a very similar clustering as scSpace! Authors should provide more evidence on this.

Response: Thanks for your professional comments. We fully agree with you that these classical single-cell clustering algorithms can produce different clusters by different parameter settings. At the very beginning, in the simulated data analysis, the targeted

cluster number (the K in k-means or the resolution in Louvain) for each clustering algorithm was determined by the ground truth. We set the same number of clusters for each method based on the number of cell populations in the simulated data when comparing scSpace with k-means, Louvain and Hclust to ensure fairness of the comparison. On the other hand, as you suggested, increasing the targeted cluster number is indeed a common and effective strategy for identifying cell subpopulations in scRNA-seq data analysis. However, since the classical single-cell clustering algorithms only work on gene expression data, this strategy may fail to identify the cell subpopulations that were similar enough in transcriptome but show significant heterogeneity in space. We have thus explored the effect of artificially improving the clustering resolution on the clustering results of k-means, Louvain and Hclust. Specifically, for each simulated data, we still fix the clustering number K of scSpace as the ground truth, while for other methods, we gradually increase K to K+5. Then we use the adjust rand index (ARI) score of clustering to benchmark the performance of all methods.

As shown in Figure 24a and Figure 24b, the performance of k-means, Louvain and Hclust didn't improve with the artificial increase of targeted clustering number, and scSpace still overperformed other methods. Besides, the strategy of increasing the number of targeted clusters may only make sense when the number of spatially heterogeneous cell subpopulations is small (Figure 24c). The results have been included in the supplementary materials and discussed in regard in the manuscript.

Figure 24. Exploration of the effect of artificially improving the clustering resolution on the

results of classical single-cell clustering algorithms using simulated datasets. **a**, ARI of all clusters for Louvain, Kmeans and Hclust on 50 simulated datasets as the number of targeted clusters increased. **b**, ARI of only spatially varied subclusters (from 2 to 4) for Louvain, Kmeans and Hclust on 50 simulated datasets as the number of targeted clusters increased. **c**, ARI of only spatially varied subclusters for Louvain, Kmeans and Hclust on 15 simulated datasets with 2 spatially varied subclusters (left), 20 simulated datasets with 3 spatially varied subclusters (middle), and 15 simulated datasets with 4 spatially varied subclusters (right), respectively.

8. Line 137-139: The idea of the paper is to be able to map single-cell data to their spatial location and Improve clustering. While in this paragraph, the authors are applying scSpace on two spatial transcriptomics datasets as another validation to show the algorithm works well. I think the scenario of wanting to predict the location of another spatial transcriptomics dataset is not of significance as this data has already the spatial information integrated.

Response: Thanks for your insightful comments. Let me explain, the purpose of applying scSpace on two spatial transcriptomics datasets (one for training and the other for validation) is to further validate the spatial reconstruction performance of scSpace, because cell or spot coordinates in spatial data are biologically meaningful and objectively present compared to simulated datasets where the spatial locations are artificially defined. This evaluation further demonstrates that scSpace is not only capable of reconstructing simple spatial structures in simulated datasets, but also more complex and biologically existing spatial architectures of various tissues, such as the cerebral cortex and tumor microenvironments. Moreover, refined cell clustering is just one application of pseudo-space reconstruction. We can also analyze other spatial events, such as cell-cell communications and immune cell invasion. Therefore, we hope to verify the performance of scSpace by using real spatial transcriptomics data with coordinates of cells or spots removed.

9. I also suggest to change the title of the second paragraph (line 131-132) mostly because of using “biological data”. To something more similar to the title of Fig 2.

Response: Thanks for your professional comments, we have revised the title of the second paragraph to “Reconstruction of the hierarchical structure of human and mouse cortex using existing data by scSpace”.

10. Line 140-145: Here I am very curious to see how the projection of the test spatial dataset on a diffusion map would look like? I doubt we would be able to capture the structure of the layers with diffusion maps (or a trajectory analysis) as well. Can authors show how the pseudo-space representation is much more informative than diffusion maps of the test slide of spatial transcriptomics both in terms of structure reconstruction or normalized pairwise distances?

Response: Thanks for your constructive comments and we fully agree with you that the comparison with diffusion map could illustrate the performance of scSpace on pseudo-space representation. Therefore, we have computed the diffusion map space of all 12 slices in human DLPFC spatial transcriptomics data using the “sc.tl.diffmap” function in scanpy package (v1.9.1).

Figure 25. Comparison of scSpace and diffusion map using human DLPFC data from donor 1 (slice 151507, 151508, 151509, and 151510).

Figure 26. Comparison of scSpace and diffusion map using human DLPFC data from donor 2 (slice 151669, 151670, 151671, and 151672).

As illustrated in Figure 25-27, the spatial distribution of cortex layers exhibits obvious hierarchical structure in the original space and shows good spatial alignment on different replicates of each donor. In addition, the pseudo-space reconstructed by scSpace successfully preserves the hierarchical structure of all cortex layers, while diffusion map fails to distinguish the hierarchical structure of layer 1 to layer 5. Furthermore, we computed the pairwise distances between spots in the original space and pseudo-space (or diffusion map), respectively, and then calculated the Pearson correlation coefficient (PCC) for each method. As shown, scSpace outperforms diffusion map for the reconstruction of the spatial relationship between spots with higher PCC scores in all tissue sections.

Figure 27. Comparison of scSpace and diffusion map using human DLPFC data from donor 3 (slice 151673, 151674, 151675, and 151676).

Specifically, the spatial reconstruction performance of scSpace on 12 slices of the human DLPFC data shows significantly higher PCC scores of the correlation between pseudo-space and original space ($p=7.4e-07$) compared with diffusion map.

Figure 28. Boxplot comparing the pairwise distances correlation between scSpace and diffusion map on 12 slices of human DLPFC.

11. Line 155-156: Here the single-cell is from the Alan atlas, but which spatial dataset has been used?

Response: Thanks for your constructive comments. Sorry that we had not clearly described the detailed information of the spatial dataset utilized as the spatial reference. We have added it in the Method section, as reflected in the revised manuscript:

“The mouse primary visual cortex V1 STARmap data was utilized as the spatial reference. We excluded the cells of the layer “CC” and “HPC” to be consistent with the layers contained in the scRNA-seq data.”

12. Line 158-159, and figure 2g: the result of this part which is the main idea of this paper-reconstructing the spatial location of single-cells- does not look very promising, the representation in pseudo-space looks not very well defined. And this is on a very structured organ like brain. I am wondering how scSpace and spatial reconstruction would look on a spatially unorganised tissue such as tumors metastasis tissue samples. Following the manuscript, I see the application of scSpace on Melanoma, but I would like to see how the original spatial transcriptomics dataset (I guess Visium) looks with these clusters shown on Melanoma.

Response: Thanks for your thought-provoking comments. The concern also raised by other reviewers. Here, we have applied RCTD (Cable et.al., *Nat Biotechnol*, 2022, <https://www.nature.com/articles/s41587-021-00830-w>), a cell-type deconvolution method for spatial transcriptomics data to validate the performance of scSpace.

As shown in Figure 29a, RCTD is employed to deconvolve the cell-type composition of the spatial transcriptomics data and explore the spatial location of C3 and C5, two T cell subpopulations farthest and nearest to malignant cells observed in the pseudo-space, in the spatial transcriptomics reference. As illustrated, the C5 and C3 subpopulations are mainly located in the tumor and lymphatic region, respectively, which is consistent with the observation of scSpace. Notably, the similar spatial relationship between different T cell subpopulations and malignant cells is also observed when utilizing the spatial transcriptomics data from other patients as the spatial reference, suggesting the robustness and universality of the pseudo-space constructed by scSpace (Figure 29b).

The spatial deconvolution results by RCTD have extensively supported the performance of scSpace on pseudo-space reconstruction. Relevant analysis and discussion have been included in the Extended data.

Figure 29. Validation of the spatial location of C3 and C5 T cell subpopulations observed by scSpace on the spatial reference deconvoluted by RCTD from biopsy 1 (a) and biopsy 2 (b).

13. Line 193-194, fig 3e: Here the author claims that scSpace clusters better the embryonic heart dataset by comparing it to Louvain clustering. I am doubtful about Louvain not being able to cluster those subpopulations as they are being clustered even on the tsne representation (which I guess is based on only single-cell gene expression data). So even without scSpace, on a reduced dimensionality space of tSNE we can see these clusters and I believe Louvain not capturing them is merely a matter of clustering resolution. I don't think this example is a good example to point out scSpace's power in improving clustering.

Response: Thanks for your professional comments. As answered in Question 6, we set the targeted cluster number to 14 for both scSpace and Seurat (Louvain) based on the number of cell types in the original annotation (Figure 30a). Besides, in order to avoid

the misclassification of three types of cardiomyocytes by Seurat due to the lack of clustering resolution, we have also explored the influence of targeted cluster number (K) on the clustering results of Seurat.

Figure 30. Comparison the clustering performance of scSpace and Seurat on human embryonic heart scRNA-seq data. **a**, t-SNE of human embryonic heart scRNA-seq data, the single-cell annotation was obtained from the original publication. **b**, t-SNE of three cardiomyocytes with the original annotation and scSpace’s clustering result. **c**, t-SNE of three cardiomyocytes with Seurat’s clustering result under different targeted cluster number setting (from 14 to 23). **d**, The change of ARI for Seurat under different targeted cluster number setting.

Specifically, we gradually increased K from 14 to 23, and as shown in (Figure 30c),

with the increase of K, the atrial and ventricular cardiomyocytes were indeed identified by Seurat at a higher clustering resolution, however, the ventricular and MYOZ2-enriched cardiomyocytes were still not well separated (Figure 30c). The ARI scores of all cell types or cardiomyocytes were consistently lower than scSpace (Figure 30d). In short, the present results indicated that scSpace is a relatively accurate and efficient method for identifying the subpopulations that were similar in transcriptome but heterogeneous in space.

In addition, the comparison is conducted by more datasets such as the human MTG dataset (Hodge et al., *Nature*, 2019) as illustrated in Figure 9 (Page 14, No. 5 Major comment raised by Reviewer 1). The results from multiple circumstances further validate the superior performance of scSpace on identifying spatially heterogeneous cell subpopulations to traditional clustering methods such as Louvain.

14. Line 232-233: “scSpace first embedded single cells in a pseudo-space to reconstruct the spatial associations of cells”. Using which spatial dataset as the spatial component of the scSpace?

Response: Thanks for your comments. Sorry that we had not clearly described the detailed information of the spatial dataset utilized as the spatial reference. We have added it in the Method section, as reflected in the revised manuscript:

*“We utilized the human DLPFC spatial transcriptomics data (Maynard et al., *Nat Neurosci*, 2021, <https://www.nature.com/articles/s41593-020-00787-0>) as the spatial reference.”*

15. Also, here the authors are mentioning applying scSpace on single-cell data from Alan brain atlas to reconstruct spatial localization of the cells. But isn't it exactly what they described in paragraph 2 line 155-159? If it is a different analysis, I guess it's very difficult to catch that difference.

Response: Thanks for your constructive comments. Let me explain, the scRNA-seq data described in paragraph 2 lines 155-159 was obtained from mouse V1 neocortex, while the scRNA-seq data described in paragraph 4 lines 231-232 was obtained from human multiple cortical areas (MTG, ACC, V1C, M1C, S1C and A1C). We have described it

clearly in the Methods section of revised manuscript, as reflected below:

“1. Mouse neocortex data

The mouse neocortex scRNA-seq (SMART-Seq v4) was obtained from <https://portal.brain-map.org/atlas-and-data/rnaseq/mouse-v1-and-alm-smart-seq>.

2. Human brain cortex data

The human brain cortex scRNA-seq data (SMART-Seq v4) of multiple cortical areas (MTG, ACC, V1C, M1C, S1C and A1C) was obtained from <https://portal.brain-map.org/atlas-and-data/rnaseq/human-mtg-smart-seq>.”

16. Line 232: using which spatial dataset?

Response: Thanks for your comments. Sorry that we had not clearly described the detailed information of the spatial dataset utilized as the spatial reference. We have added it in the Method section, as reflected in the revised manuscript:

“We utilized the human DLPFC spatial transcriptomics data (Maynard et al., Nat Neurosci, 2021, <https://www.nature.com/articles/s41593-020-00787-0>) as the spatial reference.”

17. Line 264: Here, I assume the tsne is only based on expression profile, but it also shows a segregated cluster for each scSPACE clustered clusters. So maybe only the gene expression is enough to get those subpopulations?

Response: Thanks for your insightful comments. Sorry that we had not clearly described the figure 4d. the t-SNE was based on the gene expression profile, while the result of different cell clusters was obtained by the space-informed clustering step of scSpace. We have updated the figure as well as the legend in the revised version, as reflected in the manuscript:

Figure 31. t-SNE of human cortex scRNA-seq data, colored by the clusters identified by scSpace.

18. Line 266, figure 4F, maybe not clustering them separately is due to the resolution of Louvain clustering and if you increase it you will get those clusters as we can see them separate already in tsne space

Response: Thanks for your constructive comments. As answered in Question 6 and 12, we set the targeted cluster number to 19 for both scSpace and Seurat based on the number of subclasses in the original annotation (Figure 32a).

Consistent with the ground truth, scSpace successfully identified the spatially heterogeneous clustering (Figure 32b). Same as the previous strategy, we have also attempted to increase the cluster number (K) of Seurat, and as shown in (Figure 32c), the ARI scores for Seurat (Louvain) don't increase with the increase of K. The two spatially variable subclasses of L6 are finally distinguished by Seurat when K increases to 26. Once again, the results confirmed that the spatial information of each cell is crucial for the characterization of its cellular identity, where classic single-cell clustering methods such as Seurat missed the spatial variation within the single-cell data at the same clustering resolution compared with scSpace. The results have been included in the supplementary materials.

Figure 32. Comparison the clustering performance of scSpace and Seurat on human cortex scRNA-seq data. **a**, t-SNE of human cortex scRNA-seq data, the single-cell annotation was obtained from the original publication. **b**, t-SNE of two subclasses of L6 (L6b and L6 CT) with the original annotation and scSpace’s clustering result. **c**, The change of ARI for Seurat under different targeted cluster number setting (from 19 to 26). **d**, t-SNE of two subclasses of L6 (L6b and L6 CT) with Seurat’s clustering result under different targeted cluster number setting.

19. The same point goes for Fig4 d,e and f (line 240-248). Figure 4d represents tsne (which I guess comes from gene_expression only), but it is titled as scSpace. I would expect the pseudo-space to be marked as scSpace. This is a bit confusing. Would be great if authors explain this better.

Response: Thanks for your comments. As answered in Question 16, we have revised the titled of figure 4d from “scSpace” to “clustering result of scSpace”.

20. Line 281-283: It would be much more informative if the authors show at least the tSNE and cluster distribution of both original single-cell and spatial datasets in all analysis (spatially mapped transcript of the spatial data as well, for example visium images with clustered spots on top) in supplementary without applying scSpace and analyzed by a classic single-cell/spatial data analysis pipeline (Seurat as authors have used this). As a reader I am curious to see how these initial datasets look without applying scSpace and how much is the effect of scSpace.

Response: Thanks for your constructive comments. We fully agree with you that a comparison with the original spatial data could help validate the performance of scSpace. As answered in Question 11, we have applied RCTD to deconvolute the spatial transcriptomics reference to validate the spatial relative location of C3 and C5 observed by scSpace (Figure 29a).

Figure 33. Comparison the clustering performance of scSpace and Seurat on human melanoma scRNA-seq data. a, t-SNE of 2064 T cells in human melanoma scRNA-seq data, colored by the clusters of scSpace. **b,** The expression level of melanoma-related genes in T cell subpopulations clustered by scSpace. **c,** t-SNE of 2064 T cells in human melanoma scRNA-seq data, colored by the clusters of Seurat. **d,** The expression level of melanoma-related genes in T cell subpopulations clustered by Seurat.

In addition, we have also attempted to further cluster the 2,064 T cells in to refined subclasses using Seurat (Figure 33c) and then compared the results with scSpace (Figure 33a). As shown in Figure 33b, The C5 subpopulation with high expression of a series of melanoma-related genes, such as TK1, AURKB, BIRC5, KIFC1, and MKI67, are identified by scSpace, yet failed to be accurately identified by Seurat (Figure 33d). The result further demonstrates the superiority of scSpace in spatially heterogeneous subpopulations identification by constructing and utilizing the pseudo-space information of cells.

The results have been included in the Extended Data.

21. Fig 5c: Can authors comment on what are the gray cells in this pseudo-space? A legend is missing for those cells!

Response: Thanks for your comments. The gray cells are other cells in the scRNA-seq data except T cells and malignant cells. We have added the legend in the revised version as follows.

Figure 34. Pseudo-space of the melanoma single-cell data reconstructed by scSpace.

22. Line 327-332: What is the tissue of the single-cell RNA-seq data that you have covid and healthy conditions for?

Response: Thanks for your valuable comments. The single-cell RNA-seq data are derived from a molecular single-cell lung atlas of lethal COVID-19 by Melms et.al. (<https://www.nature.com/articles/s41586-021-03569-1>), including 19 patients and 7 control individuals.

23. Which spatial dataset is used in the scSpace TCA model as the spatial reference?

Response: Thanks for your comments. The spatial reference used for TCA model is derived from the human distal lung maps (<https://www.nature.com/articles/s41586-022-04541-3>, GSE178361) of the normal human lung spatial transcriptomics data with 10X Visium technology. Notably, there is currently no available spatial transcriptomics data of COVID-19 paired to the single-cell RNA-seq data. Even though, a spatial analysis has conducted with the GeoMx DSP technology, which profiled region of interests (ROIs) of the COVID-19 infected lung. Unfortunately, the dataset cannot be used as the spatial reference because of three reasons. First, the sample size is too small for mapping the entire single-cell data. Second, the spatial coordinates of the ROIs are meaningless since they are derived from discrete regions in different tissues. Third, the transcriptomics data for each ROI can be treated as a bulk sample containing hundreds or even thousands of cells. Moreover, though, the proportion of cell types changes, the cell-type composition is conserved between the control and COVID-19 groups. Therefore, despite the change in the state of cells, their identities remain. Meanwhile, the spatial reference only provides relative spatial relationship between cells, and does not represent the real spatial transcriptomics in different conditions. In fact, even paired single-cell and spatial transcriptomics data have different cell-type proportions. In conclusion, it is reasonable to reconstruct the pseudo-space of cells from same tissues in different states or conditions using the identical spatial reference, and our results support this hypothesis.

Nevertheless, the GeoMX DSP dataset is an excellent candidate for the validation of scSpace. Therefore, we have made some attempts with the original spatial reference from healthy condition and validated the results on the GeoMx DSP dataset.

24. Here again in line 335-336:” After reconstructing spatial relationships of cells, the pseudo-space of the Control and Covid-19 group was established (Fig. 6a)”, how the spatial relationship has been built without a spatial reference? Are the authors using the scSPACE here merely as a batch correction method between healthy and disease single-cell data?

Response: Thanks for your professional comments. We apologize for the misunderstanding. As mentioned above, the pseudo-space of the control and COVID-19 groups is reconstructed with the same spatial reference from a normal lung. As also

raised by other reviewers, we explain the analysis of COVID-19 in detail to address any concerns you may have. We fully agree that the control and COVID-19 group should be analyzed on paired spatial references, respectively. However, as explained above, we use the same spatial reference from the normal lung because of the lack of the paired COVID-19 dataset and the performance of scSpace on the same reference works well.

By projecting single cells in normal and diseased tissues into the same pseudo-space with scSpace, we can compare the cell type composition and proportion, the spatial distribution patterns, and the relative pairwise associations between cell subpopulations. Furthermore, we can thus describe the process of the occurrence and development of the disease more accurately, and find the key targets for the treatment of the disease.

Additionally, we have also attempted to validate the spatial variability between the Control and COVID-19 group identified by scSpace using the GeoMx DSP targeted spatial transcriptomics dataset generated by Rendeiro et al (Rendeiro et al., *Nature*, 2021, <https://www.nature.com/articles/s41586-021-03475-6>). As illustrated in Figure 35, compared with the Control group, we observed a significant increase in myeloid cell abundance in the region of interest (ROIs) of the alveolar and airway with COVID-19, which was consistent with the observed spatial variability by scSpace that the normalized distance between the myeloid cells and alveolar/airway epithelial cells was significantly reduced in the COVID-19 group, suggesting that severe immune infiltration may have occurred.

We then calculated the differentially expressed genes between the Control and COVID-19 group using this GeoMx DSP targeted spatial transcriptomics data, and defined the genes highly expressed in the COVID-19 group ($\log_2FC > 0.5$, $FDR < 0.05$) as the “COVID-19 signatures”, to further evaluate the accuracy of spatial variability identified by scSpace by comparing the expression levels of these signatures in all myeloid cells in the COVID-19 group. Based on the normalized distance to the alveolar/airway epithelial cell in pseudo space, the myeloid cells were further classified into “near to epithelial cell” and “not near to epithelial cell” populations. Unsurprisingly, the “near to epithelial cell” myeloid population showed higher COVID-19 signatures score (Figure 35e), and highly expressed multiple MHC-related genes associated with the INF response (B2M, HLA-A, HLA-B, HLA-C, HLA-E, and HLA-DRA) (Figure 35f), indicating this myeloid population was more likely recruited into the

alveolar and airway to produce infiltration due to the robust IFN response caused by SARS-CoV-2 viral infection (Mantlo et al., *Antiviral Res*, 2020, <https://doi.org/10.1016/j.antiviral.2020.104811>), and thus appeared closer to alveolar/airway epithelial cell in pseudo-space.

Figure 35. Validation of the spatial variability identified by scSpace on the GeoMx DSP targeted spatial transcriptomics. **a**, Enrichment score of myeloid cells for each ROI in the Control and COVID-19 group. **b**, Estimated the abundance of myeloid cells for each ROI in the Control and COVID-19 group using the CYBERSORT program. **c**, Volcano plot of differential gene expressions in COVID-19 group versus Control group. Red and blue points mark the genes with significantly increased or decreased expressions in COVID-19 group (FDR < 0.05 and log2FC > 0.5). **d**, The reconstructed pseudo space for COVID-19 group by scSpace, wherein myeloid cells were divided into “near to epithelial” and “not near to epithelial” myeloid cells according to the average distance to epithelial

cells. **e**, Boxplot comparing the COVID-19 signatures scores between two types of myeloid cells. **f**, Violin plots of expression levels of MHC-related genes in two types of myeloid cells.

Minor comments

1. Line 38-39: Unnecessary comma

Response: Thanks for your comments, we have revised the manuscript.

2. Line 391: Extra “the” after discussing.

Response: Thanks for your comments, we have revised the manuscript.

Reviewer #4 (Remarks to the Author):

Reviewer's Summary

In this article by Liao et al, the researchers developed a computational technique, scSpace, to identify the spatial variability within scRNAseq data. This is a worthwhile endeavor as scRNAseq requires the dissociation of the tissue which results in the loss of this information. Likewise spatial transcriptomic platforms regularly make use of scRNAseq data to deconvolve the spots to understand cellular heterogeneity, but this information is not often applied in the opposite direction. The authors make use of simulated data as well as described sc and spatial datasets with distinct anatomical regions (brain, heart, intestine, and liver) to validate their methodologies. They further apply scSpace to different datasets of disease including melanoma and COVID-19 patient samples. scSpace is a promising approach. Some further validation and expansion on derived biological significances gained with scSpace will strengthen the article.

Response: Thanks for your positive comments and insightful summary of our work. We appreciate your precise understanding of the purpose of scSpace, which is to use spatial transcriptomics data to enhance single-cell analysis, rather than reverse spatial deconvolution using single-cell information as other spatial integration methods do. Also, we attach great importance to your valuable suggestion and fully agree with you that further validation of scSpace is needed. Therefore, we have made immediate experiments from multiple perspectives, such as method benchmarking, biological significance explanation, and cross-validation by spatial deconvolution. Specifically, we have benchmarked the clustering algorithms with 140 simulated datasets. The outcomes of scSpace are fully discussed and explained in terms of the biological function and potential therapeutic effect. Besides, the pseudo-space reconstructed by scSpace is comprehensively validated by the enrichment of spatial niches, spatial cell-cell communications inference, comparison with spatial deconvolution results, and robust performance across tissues. The method has been much improved based on your advice.

As shown below in our detailed responses, we have carefully addressed the concerns raised and find that the constructive comments have helped to strengthen the manuscript in its revised version. The significant changes are incorporated in the updated manuscript and highlighted in blue.

Major comments

1. In the simulated datasets, it appears the ARI of scSpace continues to decrease whereas Louvain and Kmeans remain relatively stable as subclusters increase. At what number of subclusters, if any, does scSpace either match Louvain and Kmeans ARI score or have worse ARI scores than these 2 techniques?

Response: Thanks for your constructive comments. We have compared scSpace with Louvain, Kmeans, and Hclust on simulated datasets with more spatially varied subclusters. The number of subclusters ranged from 2 to 10. As illustrated in Figure 36, the ARI of scSpace was consistently highest as the number of subclusters increased, suggesting its superiority in spatially heterogeneous cell subpopulation identification. Besides, the performance of scSpace remained relatively stable when the number of subclusters increased to 8, whereas other methods continued to decrease.

Figure 36. ARI of only spatially varied subclusters (from 2 to 10) across all 140 simulated datasets for scSpace and other methods.

2. In the IT neuron data analysis, figure 4j uses LAMP5 and GRIK1 which were not identified in the differentially expressed marker genes from Extended data 4a. Please explain the rationale for choosing these genes as the current rationale for their choice does not match (line 254-255).

Response: Thanks for your insightful comments. Sorry that we had not clearly described the rationale for choosing these genes. Both LAMP5, GRIK1, and the differentially expressed marker genes from Extended data 4a were identified by the function “FindAllMarkers” in Seurat (Hao et al., *Cell*, 2021,

<https://doi.org/10.1016/j.cell.2021.04.048>) ($\log_2FC > 0.5$, $FDR < 0.05$). Extended data 4a only exhibited the top 10 markers of each IT subpopulation based on the \log_2FC due to space constraints. On the other hand, we hoped to choose those genes detected in the ISH data from the same brain donor to reflect their spatial expression patterns better. However, for the differentially expressed marker genes of C1 and C2 from Extended data 4a, there was no corresponding ISH data from the same brain donor on Allen Brain Atlas. We thus selected and downloaded the ISH data of LAMP5 and GRIK1 detected in the temporal cortex of the same brain donor (Experiment ID: 80631958).

The complete lists of marker genes of each IT subpopulation have been included in the Supplementary Data 4.

3. Figure 4j uses anatomically distinct regions of the brain. Although this information is found in the methods, it would be worthwhile informing the reader in the figure/text. Are both of these anatomic regions captured in the spatial dataset used as a reference?

Response: Thanks for your suggestion, we fully agree with you and have provided the detailed information in the revised version of figure legend. The *in situ* hybridization (ISH) images from temporal cortex (LAMP5, GRIK1) (Experiment ID: 80631958) and visual cortex (GABRG1, PCP4, and RXFP1) (Experiment ID: 79506078) of adult human brain were downloaded from Allen Brain Atlas: <http://human.brain-map.org/>.

Additionally, for scSpace, the human dorsolateral prefrontal cortex (DLPFC) spatial transcriptomics data (slice 151674) rather than these two ISH data was applied as the spatial reference. The purpose of using these ISH data was to further demonstrate the ability of space-informed clustering of scSpace, where the spatial expression patterns of target genes in ISH data were consistent with the distribution of the corresponding subpopulations in layers identified by scSpace. Also, we have annotated the layers in the revised figure.

4. In the melanoma dataset analysis, the authors reference a gene TPX2 (line 307, ref 40) but this gene is not shown in the analysis like the others (Fig 4h). Similarly, the authors discovered MKI67 is highly expressed in C5, but isn't shown in Fig 4h. Is anything known about MKI67 in the literature? Is this potentially a novel therapeutic

marker identified with scSpace? Recommend adding TPX2 and MKI67 in figure 4h.

Response: Thanks for your thought-provoking comments. In the previous version, we have performed the survival analysis of TPX2 and MKI67 and showed them in Extended Data Fig. 5c. As illustrated in Figure 37, the survival analysis of TPX2 and MKI67 also supported the findings in the melanoma analysis.

As you suggested, we have searched the function of MKI67 and retrieved no report on the treatment of melanoma. However, MKI67, like other scSpace identified genes, is a well-known cell-cycle related gene. We agree with you that it could be potential therapeutic marker. And in the revised manuscript, we have included the discussion on this according to your constructive suggestion.

Figure 37. Survival probability analysis of MKI67 and TPX2.

5. In the discussion, it is stated memory T cell features begin to be lost. Where is this data presented? It is unclear in the results section where the authors identify the loss of these features and the specific features/genes that are associated with memory T cells.

Response: Thanks for your professional comments. The statement of the T-exhaustive hypothesis is referenced from published literature and we apologize for the missing of citations. In the revised manuscript, we have cited the relevant references.

6. As is the case with any reference-based approach, the reference used will heavily affect the results. Addressing this in scSpace would be worthwhile. One way this can be done is with the melanoma dataset.

1) First, when using the melanoma dataset, it is unclear if all spatial sequencing

datasets were used as a reference or just a single slide/slice (reference paper appears to have performed spatial sequencing on 4 patient samples). In previous cases it appears a single slice is used, so I assume this is the case here. If not, how was the use of multiple slides/spatial data addressed in scSpace?

Response: Thanks for your insightful comments. As you said, we only used a single slice from biopsy 1 as the spatial reference in the original melanoma analysis.

2) To validate that the spatially variable information gained from scSpace is conserved, I recommend authors make use of the multiple spatial datasets from Thrane et al (ref 36). The authors should use all 4 spatial slices from Thrane et al. and show in their article how closely related the spatially variable information is within the sc data when using each slide as a reference. This could be an optimal dataset to help show that regardless of the reference spatial dataset, spatial variability is generally conserved. These spatial datasets are also optimal to use as the spatial sequencing reference is done from the same lab and article, which could help mitigate some batch effects in the spatial data.

Response: Thanks for your constructive suggestion, we fully agree with you that multiple spatial datasets from Thrane et al. can be used to validate the conservation of the spatially variable information gained from scSpace.

In the revised version, we selected biopsy 1 and 2 as the spatial reference, because only these two slices containing the manual annotation identified region of lymphoid tissue (Figure 38a). As illustrated in Figure 38b and Figure 38d, although the layouts of pseudo-spaces reconstructed by scSpace are quite different from each other when utilizing slices from different biopsies as the spatial reference, the relative spatial relationship between malignant cells and five T cell subpopulations are conserved.

Figure 38. Spatial analysis of T cell subpopulations in melanoma. **a**, Manual annotation by pathologist. The Hematoxylin and Eosin (H&E) stained tissue sections were manually annotated by trained pathologist. Areas of melanoma (black), stroma (red) and lymphoid tissue (yellow) were identified and marked. **b**, Utilizing biopsy 1 (top) and 2 (bottom) as the spatial reference to perform scSpace.

Besides, we have also evaluated the robustness of scSpace using other datasets. In the spatial reconstruction of human skin squamous cell carcinoma (SCC) scRNA-seq data by Ji et al (Ji et al., *Cell*, 2020, <https://doi.org/10.1016/j.cell.2020.05.039>) (Figure 39a), we used the three replicated ST data of patient 2 from the same experiment as the spatial reference (Figure 39b). As illustrated in Figure 39c, the TSKs, fibroblasts, and endothelial cells of scRNA-seq data showed specific patterns of colocalizations in

pseudo space, despite using the different spatial references to perform spatial reconstruction by scSpace. Consistent with the previous study results (Ji et al., *Cell*, 2020, <https://doi.org/10.1016/j.cell.2020.05.039>) and the deconvolution results of the spatial reference calculated by RCTD (Figure 39d), fibroblasts and endothelial cells were enriched at the TSK-high leading edge, further supporting a fibrovascular niche surrounding TSK cells. Besides, the pattern of the TSK score based on markers defined by Ji et al in the pseudo space was also in agreement with the spatial structure of the TSK-proximal fibrovascular niche (Figure 39e).

Figure 39. Spatial reconstruction of human SCC scRNA-seq data using scSpace. **a**, t-SNE of human SCC scRNA-seq data, the single-cell annotation was obtained from the original publication. **b**, The multiple spatial transcriptomics data utilized as the spatial reference. **c**, The reconstructed pseudo-space (left) and the fibrovascular niche (right) of scRNA-seq data by scSpace. **d**, The cell type

deconvolution result of spatial transcriptomics reference by RCTD. e, The expression patterns of TSK score in the pseudo space.

Moreover, in the spatial reconstruction of mouse kidney scRNA-seq data by Ransick et al (*Dev Cell*, 2019, <https://doi.org/10.1016/j.devcel.2019.10.005>) (Figure 40a), we further used two spatial transcriptomics data from different experiments and technologies as the spatial reference. The 10X Visium and Slide-seq v2 spatial transcriptomics data were downloaded from 10X Visium Spatial Gene Expression of “Adult Mouse Kidney (FFPE)” (<https://www.10xgenomics.com/resources/datasets>) and GEO (GSE129798), respectively. As illustrated, the “cortex – outer medulla – inner medulla” three-layer zonal distribution pattern of cells in the kidney (Figure 40b) is both restored by scSpace using the 10X Visium (Figure 40c) or Slide-seq v2 (Figure 40d) spatial transcriptomic data as the spatial reference. Moreover, the pairwise distances between cells from different zones to Z1 and Z3 in the pseudo-space also show a consistent pattern with the ground truth of the zone distribution.

Furthermore, scSpace also accurately reconstructed the more refined spatial architectures of the principal cells (PCs) of the ureteric epithelium, whose diversities relate at least in part to position along the cortical-medullary axis (Ransick et al., *Dev Cell*, 2019, <https://doi.org/10.1016/j.devcel.2019.10.005>) (Figure 40e). As illustrated in Figure 40f, the PCs exhibit sequential orders along the cortical-medullary axis in the pseudo-space, which was consistent with the anatomies. Concordantly, the accurate spatial reconstruction of PCs was also revealed in the pseudo space when utilizing a Slide-seq v2 spatial transcriptomics data as another spatial reference (Figure 40g).

In summary, these examples suggest the robustness and universality of scSpace in the pseudo-space reconstruction. The results have been included in the Extended Data.

Figure 40. Spatial reconstruction of mouse kidney scRNA-seq data using scSpace. **a**, t-SNE of mouse kidney scRNA-seq data, the single-cell annotation was obtained from the original publication, colored by cell types (left) and zones (right). **b**, Schematic of kidney zonation. **c**, Utilizing 10x Visium spatial transcriptomics data as the spatial reference to perform scSpace. **d**, Utilizing Slide-seq v2 spatial transcriptomics data as the spatial reference to perform scSpace. **e**,

Schematic map indicating anatomic position and ontology terms for principal cells (PCs) of the ureteric epithelium. **f**, The spatial distribution of different PCs in the pseudo space with the 10x Visium spatial reference. **g**, The spatial distribution of different PCs in the pseudo space with the Slide-seq v2 spatial reference.

7. For the COVID-19 dataset, several things should be addressed:

1) First, the only pseudospace shown is for epithelial and myeloid cells. Please present a pseudospace that shows all cell types either in the figure or in the supplementary figure.

Response: Thanks for your constructive comments. We have presented the pseudo-space that shows all cell types in Figure 41.

Figure 41. The reconstructed pseudo space for Control and COVID-19 group by scSpace.

2) Figure 6d, is there significance in the normalized distance between infected and control samples? Please add significance indicators.

Response: Thanks for your valuable comments. We have performed the Wilcoxon rank-sum test to evaluate the significance in the normalized distance between COVID-19 and Control samples and added significance indicators in the revised version.

Figure 42. Normalized pairwise distances between Alveolar/Airway Epithelial cells and Myeloid subtypes in Control and COVID-19 group. The p -value represents the difference of normalized distances between Control and COVID-19 group assessed with the Wilcoxon rank-sum test.

3) What non-mitochondrial genes were enriched in C4? Are there other non-mitochondrial genes that support the pathway in figure 5j? Also, C6 appears to have a relatively high enrichment of these mitochondrial genes too, proposed thoughts on this?

Response: Thanks for your comments. There were 35 non-mitochondrial genes among a total of 49 differentially expressed genes of C4 identified by Seurat (Hao et al., *Cell*, 2021, <https://doi.org/10.1016/j.cell.2021.04.048>) ($\log_2FC > 0.5$, $FDR < 0.05$) (Figure 43a). Notably, except for multiple mitochondria-related genes, we found C4 also highly expressed a set of MHC related genes associated with the INF response (B2M, HLA-DRA, CD74, CTSS), which were enriched in the “antigen processing and presentation of exogenous peptide antigen via MHC class II pathway” in Fig. 6j in the original manuscript. Besides, other non-mitochondrial genes also supported the different pathways in Fig. 6j, for example, APOE, GRN, TMSB4X, SBNO2, NUPR1, and PIK3AP1 were enriched in the “regulation of inflammatory response” pathway, C1QA, GRN, and BNO2 were enriched in the “immune effector process” pathway, and CD14, NFATC3, CARD8, and NUPR1 were enriched in the “inflammatory response” pathway (Figure 43c). All these results indicated C4 cell subpopulation may be highly and aberrantly activated (Melms et al., *Nature*, 2021, <https://www.nature.com/articles/s41586-021->

03569-1), playing a key role in the process of COVID-19.

Notably, we choose C4 subpopulation to conduct further analysis because C4 is the nearest subpopulation to epithelial cells in the pseudo-space. Subsequently, the differential gene expression analysis shows that mitochondrial-related genes are highly expressed in the C4 subpopulation. Even though C6 appears to have a relatively high enrichment of these mitochondrial genes, it is not the nearest subpopulation to the epithelial cells. We cannot draw conclusions about the relationship between C6-epithelial cell distance and the mitochondrial gene expression.

The results have been included in the Extended Data.

Figure 43. Spatial analysis of the invasion of MDM/TMDM subpopulations in COVID-19. a, Averaged expression level of 35 non-mitochondrial genes among differentially expressed genes of C4 in MDM/TMDM subpopulations. **b,** The expression level of four MHC-related genes in MDM/TMDM subpopulations. **c,** Significantly enriched gene ontology (GO) pathway terms for the differentially expressed genes of C4 subpopulation. The non-mitochondrial genes are highlighted in red.

4) Upon reading the methods, the only spatial reference used was an uninfected lung sample. However, I do not believe this is the best approach. In a viral infection, there will be major anatomical changes in response to the virus, such as inflammation and infiltrating cells. These responses will heavily influence the spatial variation and heterogeneity. It does not seem totally appropriate to me to use a healthy lung reference to make inferences about infected samples and spatially altered variation. Authors should revisit this analysis to address these impacts of infection. It would be worthwhile using an infection reference instead or to use one to validate observed spatial variability. Example spatial references include the GeoMX references generated by Rendeiro et al. 2021 (doi: 10.1038/s41586-021-03475-6) and/or Margaroli et al. 2021 (doi: 10.1016/j.xcrm.2021.100242).

Response: Thanks for your professional comments, we fully agree with you that the spatial reference for COVID-19 is required. However, there is currently no available spatial transcriptomics data of COVID-19 paired to the single-cell RNA-seq data. Even though, a spatial analysis has conducted with the GeoMx DSP technology, which profiled region of interests (ROIs) of the COVID-19 infected lung. Unfortunately, the dataset cannot be used as the spatial reference because of three reasons. First, the sample size is too small for mapping the entire single-cell data containing over 5,000 cells. Second, the spatial coordinates of the ROIs are meaningless since they are derived from discrete regions in different tissues. Third, the transcriptomics data for each ROI can be treated as a bulk sample containing hundreds or even thousands of cells. Moreover, though, the proportion of cell types changes, the cell-type composition is conserved between the control and COVID-19 groups. Therefore, despite the change in the state of cells, their identities remain. Meanwhile, the spatial reference only provides relative spatial relationship between cells, and does not represent the real spatial transcriptomics in different conditions. In fact, even paired single-cell and spatial transcriptomics data have different cell-type proportions. In conclusion, it is reasonable to reconstruct the pseudo-space of cells from same tissues in different states or conditions using the identical spatial reference, and our results support this hypothesis. Nevertheless, the GeoMX DSP dataset is an excellent candidate for the validation of scSpace. Therefore, we have made some attempts with the original spatial reference from healthy condition and validated the results on the GeoMx DSP dataset.

By projecting single cells in normal and diseased tissues into the same pseudo-

space with scSpace, we can compare the cell type composition and proportion, the spatial distribution patterns, and the relative pairwise associations between cell subpopulations. Furthermore, we can thus describe the process of the occurrence and development of the disease more accurately, and find the key targets for the treatment of the disease.

Figure 44. Validation of the spatial variability identified by scSpace on the GeoMx DSP targeted spatial transcriptomics. **a**, Enrichment score of myeloid cells for each ROI in the Control and COVID-19 group. **b**, Estimated the abundance of myeloid cells for each ROI in the Control and COVID-19 group using the CYBERSORT program. **c**, Volcano plot of differential gene expressions in COVID-19 group versus Control group. Red and blue points mark the genes with significantly increased or decreased expressions in COVID-19 group ($FDR < 0.05$ and $\log_2FC > 0.5$). **d**, The reconstructed pseudo space for COVID-19 group by scSpace, wherein myeloid cells were divided into “near to epithelial” and “not near to epithelial” myeloid cells according to the average distance to epithelial

cells. **e**, Boxplot comparing the COVID-19 signatures scores between two types of myeloid cells. **f**, Violin plots of expression levels of MHC-related genes in two types of myeloid cells.

Additionally, we have also attempted to validate the spatial variability between the Control and COVID-19 group identified by scSpace using the GeoMx DSP targeted spatial transcriptomics dataset generated by Rendeiro et al (Rendeiro et al., *Nature*, 2021, <https://www.nature.com/articles/s41586-021-03475-6>). As illustrated in Figure 44a, compared with the Control group, we observed a significant increase in myeloid cell abundance in the region of interest (ROIs) of the alveolar and airway with COVID-19, which was consistent with the observed spatial variability by scSpace that the normalized distance between the myeloid cells and alveolar/airway epithelial cells was significantly reduced in the COVID-19 group, suggesting that severe immune infiltration may have occurred.

We then calculated the differentially expressed genes between the Control and COVID-19 group using this GeoMx DSP targeted spatial transcriptomics data, and defined the genes highly expressed in the COVID-19 group ($\log_2FC > 0.5$, $FDR < 0.05$) as the “COVID-19 signatures” (Figure 44c), to further evaluate the accuracy of spatial variability identified by scSpace by comparing the expression levels of these signatures in all myeloid cells in the COVID-19 group. Based on the normalized distance to the alveolar/airway epithelial cell in the pseudo-space, the myeloid cells were further classified into “near to epithelial cell” and “not near to epithelial cell” populations (Figure 44d). Unsurprisingly, the “near to epithelial cell” myeloid population showed higher COVID-19 signatures score (Figure 44e), and highly expressed multiple MHC-related genes associated with the INF response (B2M, HLA-A, HLA-B, HLA-C, HLA-E, and HLA-DRA) (Figure 44f), indicating this myeloid population was more likely recruited into the alveolar and airway to produce infiltration due to the robust IFN response caused by SARS-CoV-2 viral infection (Mantlo et al., *Antiviral Res*, 2020, <https://doi.org/10.1016/j.antiviral.2020.104811>), and thus appeared closer to alveolar/airway epithelial cell in pseudo-space.

The results have been included in the Extended Data.

8. The discussion is a bit underwhelming as a reader. There seems to be little connection with other scientific literature and biological significance/insight. Recommended spending more time expounding on the last paragraph. Suggest splitting melanoma and COVID-19 discussion into 2 different paragraphs and highlighting what biological insights were gained specifically using scSpace.

Response: Thanks for your constructive suggestion. In the revised manuscript, we have fully discussed the results in the melanoma and COVID-19 analysis, including relevant reports from literature and the biological meaning of the findings, in the corresponding sections. In addition, the Discussion section is thoroughly reorganized with more content regarding the benchmarks and application.

The updated discussion is highlight in Line 606-680 in the revised manuscript.

Minor edits:

1. Some choppy wording/incomplete sentences are occasionally used and should be revised. For example, see lines 55-56 and 332-334

Response: Thanks for your comments, we have revised the sentence.

2. Covid-19 should be COVID-19. This version is the preferred/recommended style for multiple public health agencies such as the World Health Organization (WHO).

Response: Thanks for your comments, we have revised Covid-19 to COVID-19.

3. Figure 2i is difficult to read with the many black outlined cells which covers the expression level colors. Recommend revising.

Response: Thanks for your comments, we have revised figure 2i as follows.

Figure 45. Reconstructing hierarchical structure of mouse neocortex scRNA-seq data using scSpace. **a**, Pseudo space of cells in each layer of the scRNA-seq data predicted by scSpace. **b**, The distribution density of each layer in pseudo space. **c**, Normalized pairwise distances between cells from different layers and Layer 1. **d**, Marker gene expression patterns of cells from different layers in pseudo space.

4. Line 299-300 add reference

Response: Thanks for your comments, we have added the reference for the T-exhaustion hypothesis (Wherry, *Nat Immunol*, 2011, <https://www.nature.com/articles/ni.2035>).

5. The Github repo has the code for using scSpace but lacks the code used for additional analyses on the datasets, such as gene set enrichment analyses. For reproducibility, it would be best practice for authors to provide all code used for analyses, not just the code to run scSpace on a given dataset.

Response: Thanks for your comments, we have provided all code used for analysis on GitHub (<https://github.com/ZJUFanLab/scSpace/tree/master/AnalysisPaper>).

Reviewer #5 (Remarks to the Author):

Reviewer's summary

The authors propose scSpace, a new method to perform the spatial clustering of a tissue sample by embedding the high-resolution level of scRNA-seq experiments with the spatial information carried by spatial transcriptomics data.

The authors illustrate the advantages of using their method with a series of key studies.

Response: Thanks for your positive feedback and insightful comments to our work. In the revised manuscript, we have devoted great efforts to the explanation of the fundamental design concept of scSpace. Specifically, we emphasize that scSpace is to use spatial transcriptomics data to enhance single-cell analysis, rather than reverse spatial deconvolution using single-cell information as other spatial integration methods do. Besides, we value your suggestions very much and fully agree with you to comprehensively highlight the advantages of scSpace from a systematic design aspect. Therefore, we have made in-depth explorations from multiple perspectives, such as introducing a neighboring system in evaluation, considering the distance information in the pseudo-space for clustering, and comparison with spatial domain identification methods. Based on your creative ideas, the method has been improved qualitatively. In the revised manuscript, many experiments are included to fully demonstrate the performance of scSpace and the Method section is reorganized. Due to the robust and excellent performance of scSpace in various scenarios, we have full confidence in this version and look forward to your second review.

As shown below in our detailed responses, we have addressed the concerns raised point by point and find that the constructive comments have helped to strengthen the manuscript in its revised version. The significant changes are incorporated in the updated manuscript and highlighted in blue.

Major comments

1. Is it that common to have for a single tissue sample both the single-cell and the s.t. expression matrices? I presume that producing two kinds of experiments from the same tissue sample would be expensive and more complex than using just one

protocol. Please, discuss this aspect more in detail, especially in the introduction. This is something that should be clear from the beginning, in such a way that people will know if this method could be used or not for their problem.

Response: Thanks for your insightful comments. As you point out, it is uncommon to conduct both single-cell and spatial transcriptomics sequencing for the same sample because this is expensive and the sample may be too small to perform both experiments. Moreover, single-cell transcriptome data still dominates the cutting-edge of omics technology, with unmet biological and clinical needs. The reanalysis of scRNA-seq data by combining spatial information is meaningful. Besides, the limitations of spatial transcriptome techniques make computational methods such as scSpace to reconstruct spatial relationships between cells within scRNA-seq data more desirable than ever before. We have fully discussed this aspect in the introduction and discussion in the revised manuscript.

2. Even if Figure 2a shows that the pseudo space looks somehow similar to the original coordinate system, my concern is that the morphology of the data is not completely retained and so this might introduce some kind of bias in the analysis. I know that the original s.t. coordinates and the pseudo space coordinates cannot be easily compared because they represent in practice two different things, but I am wondering if there exists a way to evaluate if the neighbouring system from the s.t. data is somehow preserved in the new pseudo space.

Response: Thanks for your valuable suggestion. We have introduced a neighboring system inspired by the Spatial Correlation Analysis by Ji et al (<https://doi.org/10.1016/j.cell.2020.05.039>) to further compare the original coordinates of the spatial transcriptomics data and the pseudo-space coordinates constructed by scSpace. As illustrated in Figure 46a, the neighboring system of each spot was defined as its 6 nearest neighbors in human DLPFC 10x Visium samples. The quality of the pseudo space coordinates was next evaluated by the neighboring system, where the Pearson correlation coefficient (PCC) of layer type proportion and marker gene expression was calculated based on the ground truth (Figure 46b and Figure 46d). The median PCC of layer type proportion for 12 slices is 0.778 (Figure 46c), and the median PCC of marker gene expression for 12 slices is 0.936 (Figure 46e). These results suggested that the pseudo space constructed by scSpace indeed retained the

morphology of the data in both spot arrangement and gene expression. The results have been included in the Extended Data.

Figure 46. Evaluation the performance of scSpace on human DLPFC data by the neighbor system. **a**, Illustration of the neighboring system. **b**, Pearson correlation coefficient of layer type proportion for each slice. **c**, Pearson correlation coefficient of layer type proportion for 12 slices. **d**, Pearson correlation coefficient of marker gene expression for each slice. **e**, Pearson correlation coefficient of marker gene expression for 12 slices.

3. Some recent methods for the analysis of spatial transcriptomics data, such as SpatialDE, SPARK (Sun et al. 2020) and SpaRTaCo (Sottosanti and Risso 2021), exploit the distances across the spots in the space and not just the neighbours. I would like the authors to discuss if their idea of using a pseudo space would work also in a framework that works with the distances, or if instead there would be some critical issues that might affect the results.

Response: Thanks for your constructive comments. We have added a new optional function to define the spatial weight w in scSpace package inspired by SpatialDE (Svensson et al., *Nat Methods*, 2018, <https://www.nature.com/articles/nmeth.4636>), which works with the distances across the spots directly rather than just the neighbors. We also described it detailly in the Methods section, as reflected below:

For a gene expression graph $G_g(V, E_1)$, the spatial weight w of edge $E_{i,j}$ between cell S_i and cell S_j is negatively associated with their direct distance $d_{i,j}$ in

the pseudo space, which is defined:

$$w_{i,j} = \exp\left(-\frac{d_{i,j}^2}{2l^2}\right)$$

Since the hyperparameter l , also known as the characteristic length scale, determines how rapidly the weight decays as a function of distance, and would influence the performance of scSpace partly, we further discussed the effect of l on the performance of scSpace. As shown in Figure 47, scSpace performed best on 140 simulated data when l was set to 20. In addition, compared with the default strategy of defining the spatial weight w based on neighbors, scSpace performed equally well when $l = 20$ (Figure 47e). The results suggested that scSpace was also suitable for the framework that works with distances.

The results have been included in the supplementary materials.

Figure 47. Exploration of the effect of the hyperparameter in spatial weight construction on the performance of scSpace. a, b, ARI of all clusters for scSpace with different l values across all 140 simulated datasets. c, d, ARI of spatially varied subclusters for scSpace with different l values

across all 140 simulated datasets. e, Performance comparison of scSpace between the two strategies of calculating the spatial weight evaluated with the Wilcoxon test.

4. The authors should mention Zhao et al. (2021), who propose a space-informed clustering for spatial transcriptomics data called BayesSpace that artificially augments the data resolution. I believe that scSpace would be preferable over BayesSpace because it combines two different sources of information, the scRNA-seq data and the s.t. data. However, having the same tissue be processed with two different protocols is probably not that common, as it requires a specific experimental design and it would be more expensive than using just one of the two methods. I think that the authors should discuss the pro and contra (both in terms of computational and monetary cost) of using scSpace, which embeds the information coming from two different types of experiments, against BayesSpace, which is applied only to the spatial transcriptomics data and artificially augments the data resolution.

Response: Thanks for your professional comments. We fully agree with you that space-informed clustering methods should be discussed comprehensively. While there are similarities in the space-informed clustering between scSpace and other spatial domain identification methods such as SpaGCN, BayesSpace, DR-SC, etc., we highlight the fundamental difference between the methods. Specifically, scSpace focuses on the spatial analysis of scRNA-seq data whereas other methods target the cell-type decomposition of spatial transcriptomics data. By introducing the concept of “pseudo-space”, scSpace can efficiently recover the spatial association between single cells and identify spatially heterogeneous cell subpopulations within the scRNA-seq data. However, other spatial domain identification methods, which are designed more for spatial transcriptomics data, only detect rough spatial domains with coherent expression and histology but don't attach the importance of cell independence.

Indeed, scSpace needs two types of data to reconstruct the pseudo-space of cells while BayesSpace uses only spatial transcriptomics data. However, transcriptional features from single cells and spatial information can be simultaneously considered for the identification of spatially variable cell subclusters with biological significance.

In addition, once the latent biological feature representation extraction and the spatial reconstruction steps of scSpace are performed, it would be interesting to run the space-informed clustering phase but using BayesSpace instead of the method

described in the paper (from line 488). Based on these results, it will be possible to comprehend if the results presented in this article are mainly the fruit of the pseudo space idea, or if instead all the three phases described in the section Methods have a fundamental rule.

To wrap up, I would compare the results obtained from the following models:

1) scSpace

2) BayesSpace (on the original coordinate system, using only the Visium data)

3) scSpace + BayesSpace instead of what is shown from line 488

Response: Thanks for your thought-provoking comments. It's indeed interesting that attempt to combine BayesSpace and scSpace by performing the clustering phase using BayesSpace instead of the space-informed clustering part of scSpace. However, we should point out first that the design concept of these two methods is fundamentally different. As mentioned in the manuscript, scSpace is designed for identifying spatially heterogeneous cell subpopulations in scRNA-seq data, while BayesSpace is developed to conduct resolution enhancement and efficient spatial domain clustering for spatial transcriptomics data. Directly comparison of the clustering results of scSpace and BayesSpace on scRNA-seq or spatial transcriptomics data is not really appropriate for either. Besides, BayesSpace only works for spatial transcriptomics data with regular array coordinates (such as 10X Visium or ST), and will not perform spatial clustering when applied to scRNA-seq or other types of spatial transcriptomics data such as STARmap, MERFISH, or Slide-seq.

Therefore, we simulated a specific single-cell spatial transcriptomics data as a trade-off. As shown in Figure 49a, we simulated two cell types with transcriptional variation but spatial co-localization (Group2 and Group3), and two cell subtypes with transcriptional similarity but spatial heterogeneity (Group4 and Group5). As illustrated in Figure 49b, scSpace successfully distinguished both the spatially co-localized cell types and spatially heterogeneous cell subtypes, while BayesSpace failed to distinguish two spatially heterogeneous cell subtypes that are similar in transcriptomes, and clustered them together by encouraging neighboring spots/cells to belong to the same cluster via a spatial prior (Zhao et al., *Nat Biotechnol*, 2021, <https://www.nature.com/articles/s41587-021-00935-2>). On the other hand, since the

pseudo-space reconstructed by scSpace is not a regular array coordinate system, BayesSpace only performed initial clustering rather than spatial clustering step, and the clustering result was similar to the classical single-cell clustering methods such as Seurat.

This result further demonstrated the difference in fundamental design concept between scSpace and BayesSpace.

Figure 49. Comparison of the performance between scSpace and BayesSpace. **a**, Illustration of simulated single-cell spatial transcriptomics data, where the Group 2 and Group 3 are similar in space position while Group 4 and Group 5 are similar in transcriptomes. **b**, Clustering result of scSpace, BayesSpace on the original coordinate system, BayesSpace on the pseudo-space, and Seurat.

5. My opinion is that the authors should not present the results just by referring to the figures, but they should also explain and comment more in detail the content of the figures. This consideration applies to each and every section where some results are presented.

Response: Thanks for your kind suggestion. We have thoroughly revised the manuscript and commented more in detail the content of the figures. As you can see in Line 274-312 (embryonic human heart), Line 416-471 (human melanoma), Line 484-519 (human SCC), and Line 520-605 (COVID-19), in the updated version.

6. figure 2c: what does “distance to Layer 1” mean? Is there any sort of centroid that you use to compute the distance to Layer 1? The authors should explain better this part, as well as the purpose of the second graph in figure 2c.

Response: Thanks for your comments. The “distance to Layer 1” means the average of

all pairwise distances between spots from different layers and Layer 1 in the pseudo space. We have revised “distance to Layer 1” to “Normalized distance between Layer 1 and other layers” in the revised version. Furthermore, the purpose of figure 2c was to demonstrate the correlation between relative space position of spots in the original space and pseudo space. Following Reviewer 2’s comment, we have computed the all pairwise distances between spots in the pseudo space and original space rather than just the distances to the origin, respectively, and showed their correlations in a density plot to replace Fig. 2c (Figure 49).

Figure 49. Density plot of the correlation of pairwise distances between spots in the pseudo space and original space.

7. In some cases, the authors had to consider two different samples from different subjects in order to find an appropriate set of spatial coordinates to be applied to the scRNA-seq dataset of interest. However, it seems to me that this operation inevitably carries the effects due to the biological differences between the two samples in the analysis. Please, discuss this aspect.

Response: Thanks for your professional comments. We fully agree with you that the spatial distribution of cells may not be consistent between different tissue samples from different subjects. Indeed, in the analysis of 12 slices from human cortex and 12 slices from the human SCC (cancer), we observed a robust performance of scSpace on the well-organized cortex, yet inconsistent result of several samples in the SCC, where low spatial coherence is shared between patients or even replicates of slices. This is certainly an issue worth discussing, and therefore, users are encouraged to select paired datasets with comparable conditions, states, and origins before conducting the scSpace analysis.

However, we still feel positive to scSpace because in most scenarios, such as the cortex, the intestine, the liver lobule, the kidney, the embryonic heart, the melanoma, and the breast cancer, etc., scSpace can robustly reconstruct the spatial relationship between cells with spatial references from different individuals. These successes also indicate that even though the tissues vary in morphology, their intrinsic spatial relationship between cells is preserved across multiple slices. In this perspective, the reconstruction of the pseudo-space (relative cell distances) is as important as the mapping of cell types on the spatial reference (absolute coordinates). On the contrary, such unpaired issue still exists in the inverse operation by mapping cell types to the spatial reference. Typically, single-cell and spatial transcriptomics data are derived from separated experiments. When considering highly heterogeneous tumor tissues, the different expression level, cell-type composition, and cell states between single-cell and spatial transcriptomics data can affect the result.

As mentioned above, scRNA-seq still dominates the cutting-edge omics technology with so many cell atlases established. The reanalysis of scRNA-seq data by combining spatial information is desirable and meaningful.

8. lines 147-150: please clarify this sentence and discuss it by referring to the figure, not just citing the figure.

Response: Thanks for your comments. We have thoroughly revised the manuscript and clarified this sentence, as reflected in the revised manuscript:

“Moreover, we examined the spatial distribution of differential expression genes of each layer identified by Seurat, and found that the spatial expression patterns of these genes in the pseudo space and original space were consistent well.”

The followings are some comments to the section "Identification of spatially associated subpopulations by scSpace" (lines 432-498):

9. This section aims to describe the statistical theory at the base of scSpace. In my opinion, the level of writing is quite poor and the notation is at times inconsistent or is left unexplained. I would encourage the authors to carefully revise this part, explaining with more precision the operations that are performed within the three

steps of their method, also citing the proper literature.

Response: Thanks for your constructive comments and we are sorry for the lack of detail and explanation in the Method section. In the revised manuscript, we have devoted much more efforts on this important part for all readers. We are very glad to receive your professional comments, otherwise these unclear description and careless mistakes will bring great obstacles to readers' understanding of scSpace. We fully agree with you that all descriptions and formulas should be accurate and canonical. Therefore, we consulted several experts in mathematics and reorganized all the mathematical derivation and method description. All related sections are re-written and highlighted in blue in the updated manuscript. We look forward to having your second review.

Understanding the math behind a method is crucial to determine its advantages and limitations, thus this section must be thorough.

Just to give some examples:

lines 441-442: marginal distribution of what? Say clearly what are the rows and the columns of your data matrix;

Response: Thanks for your suggestion. We have clearly described the rows and the columns of data matrix, as reflected in the manuscript:

“We denote the scRNA-seq data as $X_S = \{\mathbf{x}_{S_1}, \mathbf{x}_{S_2}, \dots, \mathbf{x}_{S_{n_1}}\}$ and the spatial transcriptomics reference as $X_T = \{\mathbf{x}_{T_1}, \mathbf{x}_{T_2}, \dots, \mathbf{x}_{T_{n_2}}\}$. Here, $\mathbf{x}_{S_i}, \mathbf{x}_{T_i} \in \mathbb{R}^N$ represent the gene expression vectors of the cell S_i in scRNA-seq data and the spot (or cell) T_i in spatial transcriptomics reference respectively, n_1 and n_2 represent the number of cells and spots, and N is the number of genes. Let $P(X_S)$ and $Q(X_T)$ be the distributions of X_S and X_T .”

line 448: what are n_1 and n_2 ? what does the summation index i stand for?

Response: Thanks for your suggestion. We have clearly described it, as reflected in the

manuscript:

*“We denote the scRNA-seq data as $X_S = \{\mathbf{x}_{S_1}, \mathbf{x}_{S_2}, \dots, \mathbf{x}_{S_{n_1}}\}$ and the spatial transcriptomics reference as $X_T = \{\mathbf{x}_{T_1}, \mathbf{x}_{T_2}, \dots, \mathbf{x}_{T_{n_2}}\}$. Here, $\mathbf{x}_{S_i}, \mathbf{x}_{T_i} \in \mathbb{R}^N$ represent the gene expression vectors of the cell S_i in scRNA-seq data and the spot (or cell) T_i in spatial transcriptomics reference respectively, **n_1 and n_2 represent the number of cells and spots**, and N is the number of genes. Let $P(X_S)$ and $Q(X_T)$ be the distributions of X_S and X_T .”*

In the original text, we made a typo about summation index i . The revised formula is as follows.

$$\text{Dist}(X'_S, X'_T) = \left\| \frac{1}{n_1} \sum_{i=1}^{n_1} \phi(\mathbf{x}_{S_i}) - \frac{1}{n_2} \sum_{i=1}^{n_2} \phi(\mathbf{x}_{T_i}) \right\|_{\mathcal{H}}^2$$

line 452: what are the four elements in the K matrix?

Response: Thanks for your suggestion. We have clearly described it, as reflected in the manuscript:

“where $K = \begin{bmatrix} K_{S,S} & K_{S,T} \\ K_{T,S} & K_{T,T} \end{bmatrix} \in \mathbb{R}^{(n_1+n_2) \times (n_1+n_2)}$ is a kernel matrix, $K_{S,S}$, $K_{T,T}$ and $K_{S,T}$ are the kernel matrices defined by $\phi(\mathbf{x}_S)$ and $\phi(\mathbf{x}_T)$.”

Lines 459-460: say clearly how to determine m and \tilde{W} ;

Response: Thanks for your suggestion. We have clearly described it, as reflected in the manuscript:

“the kernel matrix K can be decomposed as $K = (KK^{-1/2})(K^{-1/2}K)$. A more lightweight solution is to conduct dimensionality reduction to transform the empirical kernel map features vectors to an m -dimensional space with a matrix $\tilde{W} \in \mathbb{R}^{(n_1+n_2) \times m}$, where $m \ll n_1 + n_2$. The m is 50 by default.”

line 479: say clearly how to determine X'_S and X'_T (if I got it correctly, they are the output of the previous step. Thus, it must be clear how to compute them);

Response: Thanks for your suggestion. We have clearly described it, as reflected in the manuscript:

“After transposing and dividing $W^T K$, we obtain the latent feature representation $X'_S \in \mathbb{R}^{n_1 \times m}$ for scRNA-seq data and $X'_T \in \mathbb{R}^{n_2 \times m}$ for spatial transcriptomics data with true biological characteristics.”

lines 484-486: I am confused about this sentence. Probably it should be “scSpace first trains the model on s.t. data using mean squared error, ...”, but maybe I misunderstood what the authors wanted to say.

Response: Thanks for your valuable comments. We apologize for the mistakes. You are right about the true meaning here. We intended to state that “scSpace first trains the model on spatial transcriptomics data” in the original manuscript. We have revised the sentence according to your advice and thus thoroughly checked all statements.

10. In Section "spatial reconstruction" (lines 474-487), I'm wondering what could be the advantage of running the multi-layer fully connected neural network over the entire set of coordinates from the s.t. dataset without testing its prediction ability. To me, the idea of creating a pseudo space sounds like a prediction problem which answers the question “What would be the coordinate system of a scRNA-seq dataset X'_S , given that there exists a certain relation between the coordinates of a s.t. dataset and its expression matrix X'_T captured by the neural network?”

If the neural network is trained on a given s.t. dataset and tested over a separate s.t. dataset, it would be possible to evaluate if such a model is reliable to create the pseudo space.

Please, discuss your choice.

Response: Thanks for your constructive and insightful comments. In the original

manuscript, we validated the prediction of scSpace with the spatial transcriptomics data of two tissue sections, one of which was used as the spatial reference and the other as an input whose coordinates were removed in advance. We fully agree with you that training with neural networks on a given spatial transcriptomics dataset with and testing on a separate spatial transcriptomics dataset will comprehensively demonstrate the spatial prediction capability of scSpace. The results derived from 12 slices of the human dorsolateral prefrontal cortex (DLPFC) demonstrated the high accuracy of scSpace in spatial reconstruction to some extent. Notably, instead of splitting a given spatial transcriptomics dataset into a training set and a test set, we employ different replicates of tissues to verify each other. We think this is more convincing because tissue morphology and transcriptomics can be variable among replicates. Therefore, three examples including 12 slices from the human dorsolateral prefrontal cortex (DLPFC), 12 slices from the human skin squamous cell carcinoma (SCC), and 36 slices from the human breast cancer (BC), are introduced to comprehensively discuss the performance of scSpace in processing heterogeneous tissue sections.

As shown in Figure 50, when scSpace is used in tissue sections with inherent spatial patterns, such as cortical layers of brains whose structure is conservative even between multiple individuals, the resulting Pearson's correlation coefficient (PCC) scores remain at a high level robustly across many consecutive or discrete slices. For instance, the PCC scores range from 0.448 to 0.696 in the analysis of the DLPFC dataset (Figure 50a). Moreover, when more complex conditions are encountered, such as cancers, the molecular signature of the tissues is highly heterogeneous among different patients and even within tumors. Specifically, for the SCC (Figure 50b) and BC (Figure 50c) datasets, PCC scores obtained by scSpace analysis on the same tissue from several donors are under 0.4 (Patient 5 and Patient 9 in the SCC dataset, and Patient B and Patient F in the BC dataset), indicating a striking inconsistency between tissue sections, because the biological replicates from these patients share low spatial coherences. Nevertheless, in most cases, the PCC scores were consistent across tissue replicates from the same donors, and some results even demonstrate that PCC scores were reproducible in distinct donors by scSpace analysis. For instance, the high PCC scores for three replicates of Patient 2 in SCC dataset are also supported by the PASTE analysis, where Patient 2 is separated from Patients 5, 9, and 10 based on a higher spatial coherence score.

Figure 50. Performance evaluation of scSpace using multiple tissue sections from different organs, donors, and replicates. a, Pearson’s correlation coefficient (PCC) of pairwise distance between different tissue sections (Left) and different donors (Right) from the brain cortex in the DLPFC dataset. **b**, PCC of pairwise distance between different tissue sections (Left) and different patients (Right) in the SCC dataset. **c**, PCC of pairwise distance between different tissue sections (Left) and different patients (Right) in the BC dataset.

As shown in Figure 51, the molecular structure of the human brain cortex is highly organized, therefore, the intrinsic gene expression distribution also exhibits a layered spatial pattern, which results in the robust performance of scSpace across different

donors and tissue replicates.

Figure 51. The original distribution of transcriptomics, the reconstructed pseudo-space by scSpace, and the pairwise distance correlation between cells among different donors and tissue replicates in the DLPCF dataset.

In the SCC dataset (Figure 52), the composition and spatial distribution of spot types were highly heterogeneous across patients and even replicates from same patients because the tumor involves a very complex pathological process.

Figure 52. The original distribution of transcriptomics, the reconstructed pseudo-space by scSpace, and the pairwise distance correlation between cells among different donors and tissue replicates in the SCC dataset.

Similar results are also derived from the BC dataset (Figure 53). For example, tissue sections from Patient E show obvious spatial architecture in transcriptomics. However, little spatial coherence is shared in three replicates of Patient F.

Figure 53. The original distribution of transcriptomics, the reconstructed pseudo-space by scSpace, and the pairwise distance correlation between cells among different donors and tissue replicates in the SCC dataset.

In conclusion, the ST-on-ST validations illustrate a robust performance of scSpace on the pseudo-space reconstruction. However, there is still limitation regarding variable morphology of the tissue microenvironment, the heterogeneity across multiple donors, and the intrinsic intra-disease difference in distinct tissue sections. From this perspective, our findings are in line with the published work, such as PASTE (*Nat Methods*, 2022), which attributed the difficulty in the alignment and integration of spatial transcriptomics data from Patient 5 and Patient 9 to the intrinsic differences

in the spatial homogeneity of tumors. We have included the ST-on-ST experiment in the Extended Data.

11. In the Section "space-informed clustering", please describe more in detail the idea of using the space graph G_S as a weight function of the gene expression graph G_g (especially line 496).

Response: Thanks for your comments, we fully agree with you and have provided the detailed information in the revised version, as reflected in the manuscript:

Space-informed clustering

scSpace applies space-informed clustering to identify spatially heterogeneous single-cell subpopulations based on the gene expression and the generated pseudo space information of cells in scRNA-seq data. In detail, a gene expression graph $G_g(V, E_1)$ is first constructed on the reduced principal components derived from normalized gene expression using k -nearest neighbor (KNN) algorithm (top 50 PCs are selected by default). Since our goal is to find spatially heterogeneous subpopulations that may be similar in gene expression, the pseudo space information of cells is expected to be transformed to the spatial weight w of each edge in gene expression graph $G_g(V, E_1)$:

$$w = W(E_1)$$

So that the spatial relationship of cells could be considered in the later unsupervised clustering step. We provide two optional strategies to define the spatial weight w .

(1) The spatial weight of edge $E_{i,j}$ between cell S_i and cell S_j is negatively associated with their direct distance $d_{i,j}$ in the pseudo space, which is defined:

$$w_{i,j} = \exp\left(-\frac{d_{i,j}^2}{2l^2}\right)$$

The hyperparameter l , also known as the characteristic length scale, determines how rapidly the covariance decays as a function of distance. This framework that works with the distances has also been employed in SpatialDE (Svensson et al., *Nat Methods*, 2018, <https://www.nature.com/articles/nmeth.4636>).

(2) A space graph $G_S(V, E_2)$ is constructed on the pseudo space of cells using k -nearest neighbor (KNN) algorithm firstly, and the spatial weight of edge $E_{i,j}$ between cell S_i and cell S_j is negatively associated with their distance $d_{i,j}$ on the space graph $G_S(V, E_2)$, which is defined:

$$w_{i,j} = \frac{1}{\alpha + d_{i,j}} + \beta$$

Where α and β are pseudocounts to guard against excessively large and small weights, respectively (Miller et al., *Genome Res*, 2021, <https://doi.org/10.1101/gr.271288.120>). By default, $\alpha = \beta = 1$. The distance $d_{i,j}$ is calculated based on the adjacency matrix A of the space graph. Specifically, for given cell S_i and cell S_j , the distance $d_{i,j} = 1$ if S_j is the neighbor of S_i on the space graph or $d_{i,j} = 2$ if S_j is the neighbor of the neighbor of S_i and so forth.

Finally, scSpace applies unsupervised clustering on space-weighted gene expression graph using Leiden algorithm (Traag et al., *Scientific reports*, 2019, <https://www.nature.com/articles/s41598-019-41695-z>).

Minor comments

1. figure 1b: what are the x and the y axes?

Response: Thanks for your comments. The x and y axes represent the first two dimensions of t-SNE. We have added the axes annotation in the figure. The results have been included in the supplementary materials.

Figure 54. Revised figure 1b with axes annotation.

2. lines 38-40: the sentence seems to be incomplete.

Response: Thanks for your comments. We have checked the sentence.

3. lines 57-60: the sentence seems to be incomplete.

Response: Thanks for your comments. We have checked the sentence.

4. To me, lines 377-395 belong more to the Introduction than to the Discussion.

Response: Thanks for your comments. We have thoroughly revised the Discussion section of the manuscript. As highlighted in Line 606-680 in the updated version.

REVIEWERS' COMMENTS

Reviewer #1 (Remarks to the Author):

The authors have addressed my comments and made significant improvements to the manuscript. I have only a few minor comments:

1. Please spell out and briefly explain the definition of PCC the first time it is used in line 207.
2. I may have missed it, but it seems like Extended Data Fig. 6 was not referenced in the results section.
3. Extended Data 20-21 are referred to after 22-23 in the text. The authors may consider switching the order of these to be consistent with the flow of the text in the results.
4. For clarity, the authors should take care to refer to data sets as single cell or single nucleus as appropriate.

Reviewer #3 (Remarks to the Author):

All my comments are addressed in the revision.

Reviewer #4 (Remarks to the Author):

I applaud the authors on this revised manuscript. The extensive addition of analyses and datasets significantly strengthen the paper. Additionally as a reviewer with less formal background on the mathematical modeling, I appreciated the new figure additions and text expounding on this. I only have a few minor comments and suggestions.

1. Personal preference, but in the abstract please remove the etc following the tissue types (line 41) and just list all tissues or say something like "and others". Given the breadth of datasets used, this just felt a bit out of place since many of the datasets are not inherently connected.
2. Line 297 change 0,91 to 0.91
3. Lines 448-454 if you increase K will Seurat eventually identify the C5 population? Please state whether or not this occurs, and if it does, at which K value.
4. Line 575 this should be IFN not INF. Also please define IFN first before using acronym(interferon).
5. I may have overlooked this but what was the cutoff value in distance for labeling "near to epithelial cells" and "not near to epithelial cells"? If not explicit, please state and rationale for that cutoff.
5. Addition of MHC data was nice to see in COVID-19 data.
6. In general, figures with the grey dots labeled as "others" in the legend is still a bit confusing. I understand reading through your responses what "others" means, but a reader may still wonder what this means. In the figure descriptions, please just add a short description. For example, figure 6c it states "...average distance to alveolar/airway epithelial cells." so just add something like "...average distance to alveolar/airway epithelial cells. All other cell types are labeled as "others"." This occur at least with Figures 4 and 6 and Supp Fig 30.

7. Supp Fig 30c. Would be nice to label the signature genes on the volcano plot as I didn't see those genes in a supplementary table.

Reviewer #7 (Remarks to the Author):

The authors extensively addressed all the comments raised by reviewer #2 revising the manuscript accordingly.

Response to reviewers

Overview of Changes

We sincerely appreciate the reviewers' constructive comments and positive feedback. Our work has been much improved based on their valuable suggestions. We have tried our best to address the concerns raised by the reviewers point by point. Compared to the original version, the revised manuscript corrects errors in the description, and adds results of further comparisons with classical clustering methods.

We hope this edition will satisfy the reviewers and address all raised concerns to win their approval for the publication of our manuscript. Please find detailed responses below.

Reviewer #1 (Remarks to the Author):

Reviewer's summary

The authors have addressed my comments and made significant improvements to the manuscript. I have only a few minor comments:

Response: Thanks for your positive comments.

1. Please spell out and briefly explain the definition of PCC the first time it is used in line 207.

Response: Thanks for your constructive comments, we have spelt out and briefly explained the definition of PCC in the revised manuscript, as reflected below:

“Interestingly, for the STARmap data, scSpace can successfully restore the spatial arrangement of cells along the X-axis, with a Pearson’s correlation coefficient (PCC) of 0.782 (Fig. 2f), ...”

2. I may have missed it, but it seems like Extended Data Fig. 6 was not referenced in the results section.

Response: Thanks for your professional comments. We apologize for missing Extended Data Fig. 6 in the results section. In the revised manuscript, we have included the instruction, as described below:

“As illustrated in Fig. 2a and Extended Data Fig. 5-6, scSpace can successfully reconstruct the hierarchical structure of different human DLPFC layers in the pseudo-space, with the relative position between the layers as well as the neighbor system of spots preserved (Fig. 2b).”

3. Extended Data 20-21 are referred to after 22-23 in the text. The authors may consider switching the order of these to be consistent with the flow of the text in the results.

Response: Thanks for your valuable comments. We have switched the order of

Extended Data 20-21 and Extended Data 22-23.

4. For clarity, the authors should take care to refer to data sets as single cell or single nucleus as appropriate.

Response: Thanks for your valuable comments. We have revised the descriptions in the update manuscript.

Reviewer #3 (Remarks to the Author):

Reviewer's summary

All my comments are addressed in the revision.

Response: Thanks for your positive comments.

Reviewer #4 (Remarks to the Author):

Reviewer's summary

I applaud the authors on this revised manuscript. The extensive addition of analyses and datasets significantly strengthen the paper. Additionally as a reviewer with less formal background on the mathematical modeling, I appreciated the new figure additions and text expounding on this. I only have a few minor comments and suggestions.

Response: Thanks for your positive comments.

1. Personal preference, but in the abstract please remove the etc following the tissue types (line 41) and just list all tissues or say something like “and others”. Given the breadth of datasets used, this just felt a bit out of place since many of the datasets are

not inherently connected.

Response: Thanks for your professional comments. We have revised the sentence in the update manuscript, as reflected below:

“When employed to reconstruct the spatial architectures of complex tissue such as the brain cortex, the small intestinal villus, the liver lobule, the kidney, the embryonic heart, and others, ...”

2. Line 297 change 0,91 to 0.91

Response: Thanks for pointing it out. We have corrected it in the update manuscript.

3. Lines 448-454 if you increase K will Seurat eventually identify the C5 population? Please state whether or not this occurs, and if it does, at which K value.

Response: Thanks for your insightful comments. We have attempted to increase the cluster number (K) of Seurat, and as shown in (Figure 1), the C5 population identified by scSpace is finally distinguished by Seurat when K increases to 7. Once again, the results confirmed that the spatial information of each cell is crucial for the characterization of its cellular identity, where classic single-cell clustering methods such as Seurat missed the spatial variation within the single-cell data at the same clustering resolution compared with scSpace. The results have been included in the supplementary materials.

Figure 1. Comparison the clustering performance of scSpace and Seurat on human melanoma scRNA-seq data. a, t-SNE of 2064 T cells in human melanoma scRNA-seq data, colored by the clusters of scSpace. **b**, t-SNE of 2064 T cells in human melanoma scRNA-seq data colored by the clusters of Seurat under different targeted clustering number setting (from 5 to 7).

4. Line 575 this should be IFN not INF. Also please define IFN first before using acronym(interferon).

Response: Thanks for your professional comments. We have revised INF to IFN and defined it first, as reflected in the update manuscript:

“..., such as B2M, HLA-A, HLA-B, HLA-C, HLA-E, and HLA-DRA, which are associated with the interferon (IFN) response (Extended Data Fig. 30f).”

5. I may have overlooked this but what was the cutoff value in distance for labeling “near to epithelial cells” and “not near to epithelial cells”? If not explicit, please state and rationale for that cutoff.

Response: Thanks for your professional comments. We apologize for missing the description of the cutoff value. In the revised manuscript, we have added it, as reflected below:

“Based on the normalized distance to the alveolar/airway epithelial cell in the pseudo-space, myeloid cells are further classified into two populations, where the cells with a close average distance to the alveolar/airway epithelial cell (the top 50%) are defined as the “near to epithelial cell” populations and the rest as “not near to epithelial cell” populations (Fig. 6c and Extended Data Fig. 30d).”

6. Addition of MHC data was nice to see in COVID-19 data.

Response: Thanks for your positive comment.

7. In general, figures with the grey dots labeled as “others” in the legend is still a bit confusing. I understand reading through your responses what “others” means, but a reader may still wonder what this means. In the figure descriptions, please just add a short description. For example, figure 6c it states “...average distance to alveolar/airway epithelial cells.” so just add something like “...average distance to alveolar/airway epithelial cells. All other cell types are labeled as “others”.” This occur at least with Figures 4 and 6 and Supp Fig 30.

Response: Thanks for your professional comments. We have added the description of “others” to all the relevant figure legends.

8. Supp Fig 30c. Would be nice to label the signature genes on the volcano plot as I didn't see those genes in a supplementary table.

Response: Thanks for your professional comments. We have labeled the top-10 signature genes (based on the \log_2FC) on the volcano plot (Figure 2). Additionally, the complete lists of signature genes in the COVID-19 group have been included in the Supplementary Data 6.

Figure 2. Volcano plot of differential gene expressions in COVID-19 group versus Control group. Red and blue points mark the genes with significantly increased or decreased expressions in COVID-19 group ($FDR < 0.05$ and $\log_2FC > 0.5$).

Reviewer #7 (Remarks to the Author):

Reviewer's Summary

The authors extensively addressed all the comments raised by reviewer #2 revising the manuscript accordingly.

Response: Thanks for your positive comments.